# Intranasal mask for protecting the respiratory tract against viral aerosols

Xiaoming Hu[1,2,14], Shuang Wang[1,2,14], Shaotong Fu[2,3,14], Meng Qin[4], Chengliang Lyu[1], Zhaowen Ding[1], Yan Wang[1,2], Yishu Wang[1,2], Dongshu Wang[5], Li Zhu[5], Tao Jiang[6], Jing Sun[7], Hui Ding[8], Jie Wu[1,2], Lingqian Chang[9], Yimin Cui[10,11], Xiaocong Pang[10,11], Youchun Wang[12], Weijin Huang[12], Peidong Yang[13], Limin Wang[2,3] ✉, Guanghui Ma[1,2] ✉ & Wei Wei[1,2] ✉

The spread of many infectious diseases relies on aerosol transmission to the respiratory tract. Here we design an intranasal mask comprising a positively-charged thermosensitive hydrogel and cell-derived micro-sized vesicles with a specific viral receptor. We show that the positively charged hydrogel intercepts negatively charged viral aerosols, while the viral receptor on vesicles mediates the entrapment of viruses for inactivation. We demonstrate that when displaying matched viral receptors, the intranasal masks protect the nasal cavity and lung of mice from either severe acute respiratory syndrome coronavirus 2 or influenza A virus. With computerized tomography images of human nasal cavity, we further conduct computational fluid dynamics simulation and three-dimensional printing of an anatomically accurate human nasal cavity, which is connected to human lung organoids to generate a human respiratory tract model. Both simulative and experimental results support the suitability of intranasal masks in humans, as the likelihood of viral respiratory infections induced by different variant strains is dramatically reduced.

Infectious diseases have massive global impacts in terms of public health[1–4]. According to a report by the World Health Organization, coronavirus disease 2019 (COVID-19) has killed 6.9 million people thus far, and influenza causes over 290,000 deaths each year (data from the World Health Organization). Both of these infectious diseases share the following common feature: they are spread by virus-laden aerosols in both short and long ranges[5–7]. Typically, the size of viral aerosols is ~1–10 μm[8]. Infectious viral aerosols are produced by the expiratory activities of infected patients (*e.g.*, breathing, talking, coughing, and sneezing)[9–11] and travel with the airflow. These viral aerosols can remain infective for up to several days[12,13]; once humans inhale those floating viral aerosols, infection may occur.

Utilizing face masks has been among the important elements of public health efforts to reduce the rates of respiratory infections[14,15], especially during the COVID-19 pandemic. However, the protective effect of face masks against severe acute respiratory syndrome

coronavirus 2 (SARS-CoV-2) was only 67%[16,17]. This rate was even lower for SARS-CoV-2 variants, such as Omicron, because the variants exhibited a stronger infection ability[18–21]. The above situation suggests that providing a new strategy to prevent viral aerosol infection is necessary. Given that the entry point for viral aerosols is the human nasal cavity[22], we envisioned that applying a layer here might enable efficient interception and even potentially inactivate viruses that might be entrapped.

Here, we engineer an intranasal mask (Fig. 1a), which is composed of an irreversibly thermosensitive hydrogel with positive charges, into which we introduce engineered cell-derived microsized vesicles (MV). These MV contain known receptors for specific viruses that are overexpressed on the vesicle surface. The resulting "MV@GEL" can be sprayed into the nasal cavity at room temperature and quickly transforms from the liquid state to the gel state at body temperature, which is favorable for prolonging the retention

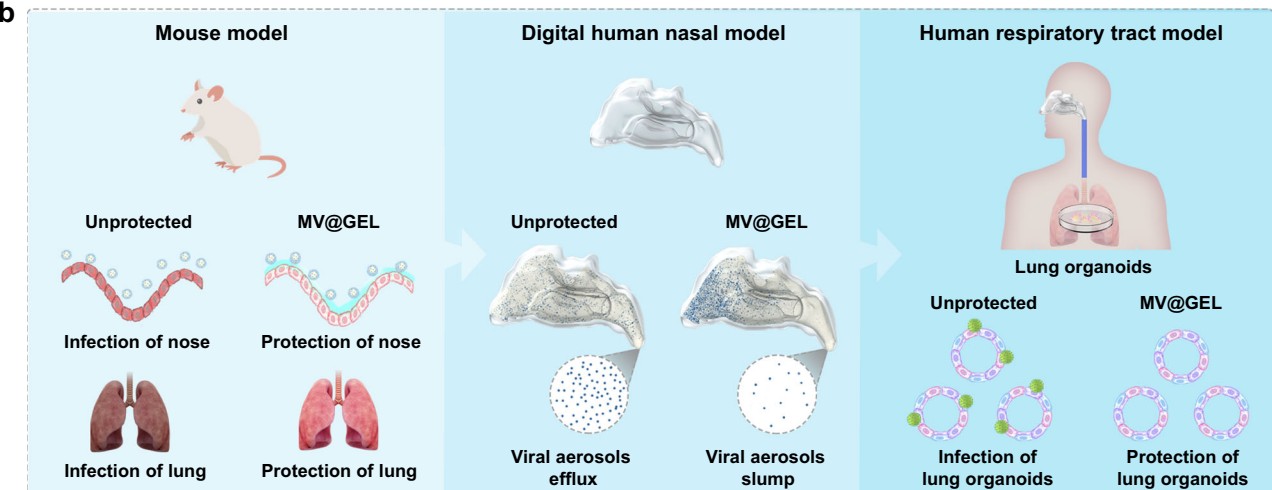

**Fig. 1 | Schematic and experimental design for MV@GEL as an intranasal mask to intercept viral aerosols and entrap virus. a** The intranasal mask (MV@GEL) was composed of engineered cell-derived microsized vesicles (MV) with viral receptor and thermosensitive hydrogel with positive charges. It could be sprayed into the nasal cavity at room temperature and quickly transformed from the liquid state to the gel state at body temperature. The viral receptor of vesicles could help vesicles entrap the virus, and the thermosensitive hydrogel could prolong the retention time of vesicles in the nasal cavity. Once the negative viral aerosols were inhaled, the intranasal mask could perform the protective effect in the following steps: Step 1, the positively charged hydrogel could intercept the negatively charged viral aerosols presenting in airflow; Step 2, those viral aerosols could fuse with MV@GEL and release viruses into MV@GEL; Step 3, the embedded MV in

MV@GEL could entrap those released viruses. **b** The protective effect of the intranasal mask was investigated from the following three aspects. 1. Mouse model: MV@GEL conferred strong protection against viral aerosol infection in the mouse nose and downstream lung; 2. Digital human nasal model: based on computerized tomography (CT) images of the human nasal cavity, computational fluid dynamics (CFD) simulation supported that viral aerosols could be intercepted in the human nasal cavity under MV@GEL protection; 3. Human respiratory tract model: connecting a realistic human nasal apparatus with human lung organoids and providing respiratory airflow by the pump, the human respiratory tract model was constructed and utilized to demonstrate the good performance of MV@GEL in protecting the lung organoids from viral aerosols.

time on the intranasal wall of the nose. The positively charged hydrogel can intercept the negatively charged viral aerosols presenting in airflow, while the receptor on the vesicles can interact with the virus that is released from viral aerosols to MV@GEL, thereafter mediating the entrapment of virus for inactivation. Working with the mouse model, we show that MV@GEL systems confer strong protection to the nasal cavity and downstream lung against viral aerosol infection (Fig. 1b). We also acquire computerized tomography (CT) data of the human nasal cavity to support computational fluid dynamics (CFD) simulation; with the CT image data and three-dimensional (3D) printing technology, we finally

fabricate a realistic apparatus of the human nasal cavity, which is further connected to a human lung organoids culture and provided with respiratory airflow by a pump, resulting in an integrated human respiratory tract (HRT) model. Through this integrated model, we confirm the potent protection of MV@GEL against viral aerosols of different variant serotype, showing the great suitability in humans. The unique merits of broad protection against different viral mutation/serotype, flexible receptors engineering for matching different types of viruses, and translational potential support our intranasal mask as a promising shield modality for improving public health in the pandemic.

## Results

### Preparation and characterization of MV

Inspired by receptor-mediated viral infections in host cells[23–27], studies using viral receptors as virus decoys[28–34], and the size of most viruses ranging from 60 nm to 140 nm[35], we envisioned that cell-derived vesicle with abundant viral receptors and a large cavity could facilitate viral entrapment[36]. Owing to the lack of organelles, DNA, etc., in the MV, the entrapped virus would fail to replicate inside the vesicle[37], thus excluding the possibility of infecting the host cells. To this end, we initially prepared MV displaying overexpressed angiotensin-converting enzyme 2 (ACE2) for SARS-CoV-2 (aMV). Briefly, we first transfected 293 T cells with ACE2-plasmid and then screened out a cell line that stably overexpressed ACE2 on the cell membrane (ACE2-293T cells) (Fig. 2a). After immunostaining with ACE2 antibody and corresponding secondary fluorescent antibody, confocal laser scanning microscopy (CLSM) images revealed a substantial ACE2 signal (white) on the membrane of ACE2-293T cells rather than 293T cells treated with empty plasmid (WT-293T; Fig. 2b). Such overexpression was further confirmed by flow cytometry analysis (Fig. S1), showing that almost all ACE2-293T cells highly expressed ACE2.

Next, we used the cell-permeable mycotoxin cytochalasin B to promote the formation and release of aMV from ACE2-293T cells (Fig. 2c). Upon vortexing and centrifugation, both CLSM and transmission electron microscopy (TEM) imaging showed that the obtained vesicles exhibited a typical vesicle structure with an average particle size of approximately 1.26 μm (Figs. 2d and S2). These aMV were further stained with 1,1'-dioctadecyl-3,3,3',3'-tetramethylindodicarbocyanine,4-chlorobenzenesulfonate salt (DiD, membrane dye, red) and ACE2 antibody with the corresponding secondary fluorescent antibody, while the MV from WT-293T (nMV) was used as the control. As expected, there was no obvious ACE2 signal (white) on nMV, but aMV contained a substantial accumulation of ACE2 on the surface (Fig. 2e), with a loading amount of 50 μg ACE2 per mg aMV. Corresponding flow cytometry analysis also showed that the proportion of ACE2-positive vesicles in the aMV group was over 95% (Fig. 2f). Additionally, aMV exhibited consistent characteristics in terms of average size and ACE2 expression (Fig. S3), thus ensuring the production repeatability and paving the way for subsequent interaction and entrapment of SARS-CoV-2.

### In vitro entrapment of SARS-CoV-2 pseudovirus by aMV

With these aMV in hand, we tested whether the aMV worked as designed for interacting with and entrapping SARS-CoV-2. We exposed DiD-stained aMV to fluorescein isothiocyanate (FITC)-stained SARS-CoV-2 pseudovirus (SPV, wild-type) at 37 °C and then conducted stimulated emission depletion (STED) microscopy imaging. Initially, we observed the SPV signal (green) at the aMV surface (Fig. S4a), indicating a favorable interaction between SPV and ACE2 on the membrane. Over time, the SPV signal was evident at both the surface and the interior of aMV (Fig. 2g), hinting at the occurrence of entrapment. Moreover, we also prepared ultrathin section of the SPV-challenged aMV, and TEM imaging revealed the structure of the SPV within aMV (Fig. 2h), thereby providing stronger evidence for the entrapment of SPV by aMV. In contrast, no obvious SPV signal appeared on or in nMV (Fig. S4b–d), thus excluding nonspecific binding of SPV on MV and again suggesting that ACE2 specifically mediated the entrapment of SPV into aMV.

The above results prompted us to test whether aMV could reduce the extent of ACE2-293T cell infection after being challenged with SPV. Prior to challenge with SPV, ACE2-293T cells were treated with one of the following six agents: PBS, recombinant human ACE2 protein, nMV, nanoscale vesicles prepared from ACE2-293T cells (aNV, a comparison group for vesicle size, Fig. S5), aMV, and aMV pre-blocked by ACE2 antibody (aMV/anti-ACE2). Then, we monitored the expression of green fluorescent protein (GFP), which SPV can express upon entry

into the host cell. Compared with the slightly to medially inhibited SPV infection (green) in the ACE2 protein, nMV, aMV/anti-ACE2, and aNV groups, a substantially reduced SPV infection was observed in the aMV group (Fig. 2i), with a dose-dependent manner of protection (Fig. S6). This should be attributed to the rational design of aMV by integrating the indispensable features of the cellular vesicle, micro-size, and ACE2 display for efficient entrapment of SPV, which together result in complete SPV shielding and avoid the infection risk induced by residual inactivated S proteins on the surface of SPV (Fig. S7). Moreover, aMV also persistently showed potent inhibition of host cell infection against the variants of SPV, including D614G, the variant of interest (B.1.617.1, P.2, B.1.429), and the variant of concern (B.1.1.7, B.1.351, P.1, B.1.617.2 (Delta), B.1.1.529 (Omicron)) (Fig. 2j), highlighting the unique merit that aMV's prominent performance was independent of SARS-CoV-2 mutation. This merit could be attributed to the SARS-CoV-2's naturally binding to the ACE2 protein on the surface of MV, which remained unaffected by viral mutations. Of note, no detectable change of cell protective effect was observed even after keeping aMV in 4 °C for 3 months, indicating a good long-term storage stability (Fig. S8).

### Preparation of chitosan/β-sodium glycerophosphate hydrogel loaded with aMV

Having confirmed the virus-entrapping capacity and infection-mitigating ability of aMV, we decided to use a gel-based strategy for the in vivo application of aMV. Given that many viruses are transmitted via aerosols, we envisioned that delivering and retaining aMV in the nasal cavity should help suppress the transmission of virus to infection-sensitive cells in the nasal cavity and downstream lung. We had experience working with thermosensitive hydrogels, and in the present study, we explored the use of chitosan and β-sodium glycerophosphate to prepare irreversibly thermosensitive hydrogels with good wettability and adhesion property[38,39]. These hydrogels were liquid at room temperature (25 °C) but gelified at the body temperature of ~37 °C, providing an ideal medium for embedding aMV and prolonging retention in the nasal cavity. Additionally, we realized that selecting this positively charged gel material should enable the following antiviral process: the positively charged gel could intercept negatively charged viral aerosols by electrostatic interactions to cut off the spread of viral aerosols[40,41]. In this case, restricting the released virus in the gel could further facilitate subsequent virus entrapment by the embedded aMV.

After extensively testing three chitosan gel recipes in terms of their thermosensitive gelling property (Fig. S9a), spray characteristic (Fig. S9b), interception ability to SPV aerosols (Fig. S9c), interaction force with SPV (Fig. S9d), effect on cell tight junction (Fig. S9e), and cytotoxicity (Fig. S9f), we selected one chitosan gel recipe (12 mg/ml chitosan, 200 mg/ml β-sodium glycerophosphate, GEL) that exhibited a good performance in all aspects (Fig. S9g). Then, GEL was mixed with aMV to generate aMV@GEL (Fig. 3a), wherein, as shown in the CLSM image (Figs. S10, S11, and 3b), DiD-stained aMV was well dispersed in FITC-stained GEL and remained the micro-sized vesicle structure. Compared with the GEL (Fig. S12), aMV@GEL maintained many similar features in terms of its porous structure (pore size ~10–20 μm, Fig. 3c), thermosensitive gelling property (360 s at 37 °C, Figs. 3d and S13), spray characteristic (spray area 2.3 cm², 10 cm away from the nozzle, Fig. 3e), and positive charge (10.1 mV). Additionally, the integrity and in vitro anti-viral efficacy of aMV embedded in GEL was not influenced after experiencing spray-induced shear rates (Figs. S14 and S15), paving the way for exerting entrapment function upon spray in the nasal cavity.

### In vitro interception of SPV aerosols and further entrapment of SPV by aMV@GEL

To evaluate the ability to intercept viral aerosols, we utilized an aerosol generator that was capable of producing aerosols with a size

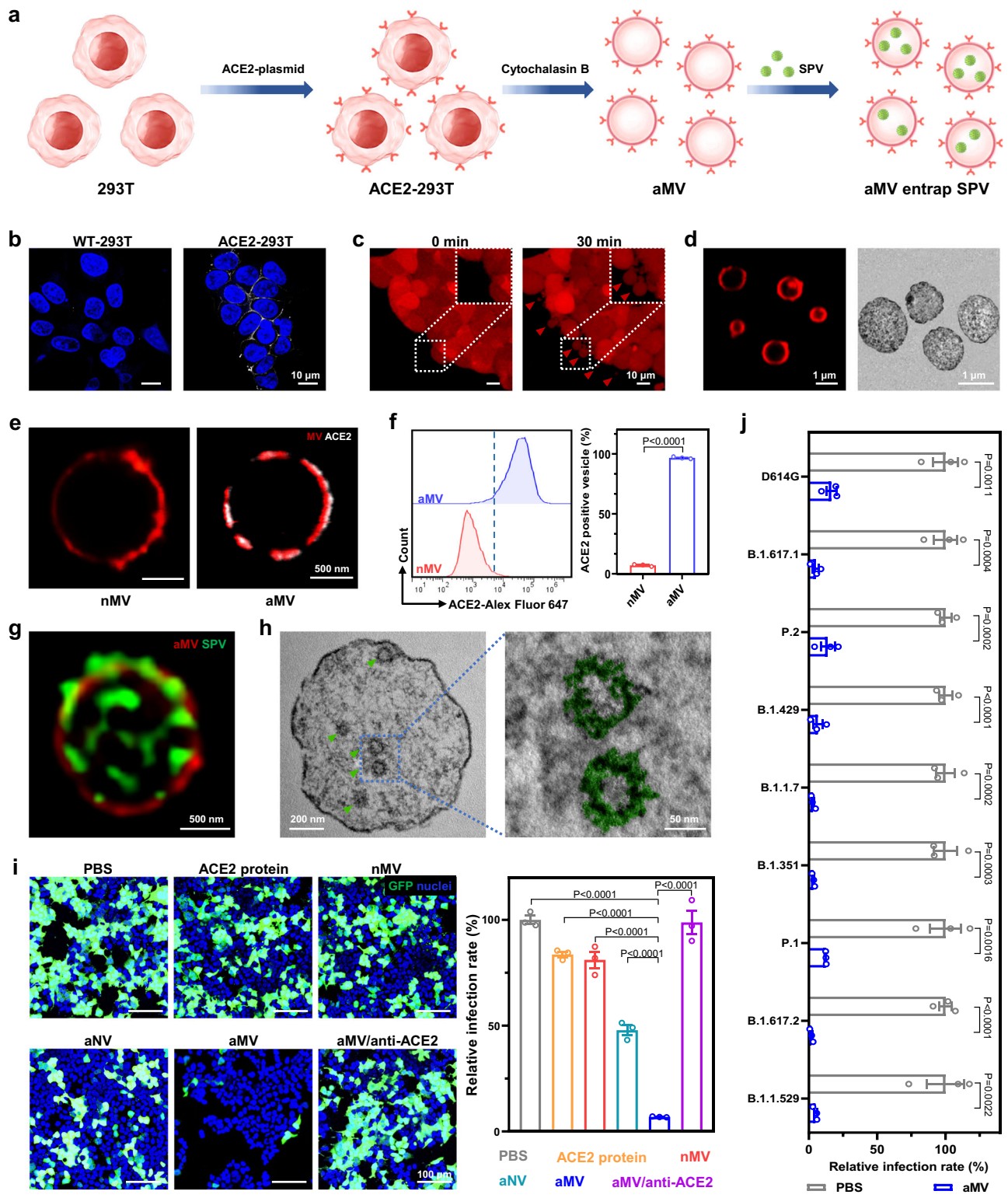

distribution of 1–10 μm (Fig. S16) and designed an in vitro experiment. As shown in the left panel of Fig. 3f, the inner wall of a tube was precoated with PBS, aMV, GEL, or aMV@GEL, which was followed by incubation at 37 °C. Subsequently, the aerosols containing cyanine 5 *N*-hydroxysuccinimide ester (Cy5)-labeled SPV (wild-type) were nebulized into the tube from the left side, and the fluorescence intensity inside the tube was detected by an imaging system. Compared to the PBS and aMV groups, the GEL and aMV@GEL groups showed much stronger interception of the viral aerosols, with a greater extent of

spatial observation in the left area due to the high concentration of viral aerosols near the tube entrance (Fig. 3f, g). In addition, we collected the outflow of viral aerosols from the right side of the tube by immersing this side into the water. The fluorescence intensity of water in the GEL and aMV@GEL groups showed about 3-fold reduction compared to that of the PBS group (Fig. 3h), further supporting the superior interception ability of GEL and aMV@GEL.

Next, we were interested in whether the virus could be released from the viral aerosols after interception (Fig. 3i). To this end, we

**Fig. 2 | Preparation of MV with high angiotensin-converting enzyme II (ACE2) expression and their entrapment effect on SARS-CoV-2 pseudovirus (SPV).**
**a** Schematic for MV with high ACE2 expression (aMV) preparation and its mechanism of entrapping the SPV. **b** Confocal laser scanning microscopy (CLSM) images showing the ACE2 protein expression on wild-type 293 T cells transfected with control empty plasmid (WT-293T) or ACE2-plasmid (ACE2-293T). White: ACE2 antibody labeled ACE2 protein. Blue: 4′,6-diamidino-2-phenylindole (DAPI) labeled cell nuclei. **c** CLSM images of ACE2-293T cells after cytochalasin B treatment. Red: carboxyfluorescein succinimidyl ester labeled cytoplasm (CSFE). **d** CLSM and transmission electron microscope (TEM) images of aMV. Red: 1,1′-dioctadecyl-3,3,3′,3′-tetramethylindodicarbocyanine,4-chlorobenzenesulfonate salt (DiD) labeled cell membrane. **e** CLSM images of ACE2 protein expression on vesicles that were obtained from WT-293T cells (nMV) and aMV. Red: DiD labeled cell membrane. White: ACE2 antibody labeled ACE2 protein. **f** Flow cytometry analysis of ACE2 protein expression on nMV and aMV. The ACE2 protein were labeled with the

ACE2 antibody. **g** Stimulated emission depletion (STED) microscopy image of SPV (wild-type) internalized within aMV. Green: fluorescein isothiocyanate (FITC) labeled SPV. Red: DiD labeled cell membrane. **h** TEM images of ultrathin section of aMV after 24 h of challenge with SPV (wild-type). The green arrow indicated SPV. In the enlarged image (right), SPV was marked with false color (green). **i** CLSM images of ACE2-293T cells that were treated with the indicated agents and then challenged with SPV (wild-type), accompanied with the corresponding quantification of relative infection rate. Green: GFP was expressed in infected cells. Blue: DAPI labeled cell nuclei. **j** Relative infection rates of ACE2-293T cells that were treated with PBS or aMV and then challenged with the indicated SPVs with S protein mutation. Quantitative data in **f**, **i**, and **j** represent as means ± S.E.M., $n = 3$ biologically independent experiments. Statistical significance was calculated using two-tailed unpaired t-test (**f**, **j**) and one-way ANOVA with multiple comparison tests (**i**). All P-values are indicated. Source data are provided in the Source data file.

diluted Cy5-SPV into water containing fluorescein sodium, which was nebulized to form viral aerosols. The colocalization condition of SPV and water in viral aerosols was then investigated by CLSM before and after interception by aMV@GEL. As shown in Fig. 3i(i), the Cy5-SPV signal (green) was restricted to each aerosol (purple) before arriving at the surface of aMV@GEL. Once the viral aerosols contacted the GEL, the signals of both the water containing fluorescein sodium and Cy5-SPV became dispersed in the GEL (Figs. 3i(ii and iii) and S17), indicating the aerosols were deconstructed and SPV was released after interception. These free SPVs then diffused in the GEL (Fig. S18), and colocalized with DiD-aMV (red) in the FITC-stained GEL (cyan) (Fig. 3i(iv)), indicating that the released SPV was entrapped by aMV and excluding the possibility of viral infection of the underlying host cells (Fig. S19) in vitro and the underlying epithelial cells in the nasal cavity in vivo.

### In vivo distribution of aMV@GEL and its protective effect against SPV aerosols

Encouraged by the strong performance of aMV@GEL in vitro, we next carried out a series of in vivo experiments. First, we assessed the impacts of aMV@GEL on the basic respiratory function of mice. Briefly, we applied aMV@GEL to the nasal cavity of mice through intranasal administration. We detected no significant differences in respiratory rate, tidal volume, and ventilate volume before and after aMV@GEL administration (Fig. S20a). Moreover, CLSM images of the nasal epithelium layer and hematoxylin-eosin (H&E) staining data of the nasal mucosal cells together indicated that aMV@GEL did not cause obvious damage to the structure of the nasal mucosa (Fig. S20b, c). In addition, the comprehensive evaluations, including blood routine analysis, serum biochemistry index analysis, nasal inflammatory cytokine detection, and histological analysis of major tissues, confirmed the safety upon multiple aMV@GEL administrations (Fig. S21).

Next, we stained the aMV with Cy5 and compared the retention times of free aMV and aMV@GEL after intranasal administration (Fig. 4a). As shown in Fig. 4b, free aMV rapidly disappeared from the nasal cavity within 4 h. In sharp contrast, the gel prominently prolonged the retention of aMV in the nasal cavity, as the signal was still detectable at 8 h in the aMV@GEL group. After quantification and calculation, the half-life of aMV@GEL was 5.23-fold longer than that of the free aMV group (Fig. 4c). In another parallel experiment, we sacrificed mice at 8 h and prepared frozen sections of the nasal cavity. CLSM imaging showed a bright fluorescent layer (red) on the surface of the nasal epithelium in the aMV@GEL group rather than the free aMV group (Fig. 4d). In addition, we detected the fluorescence signal in the major tissues. For the free aMV group, the lung showed obvious accumulation of aMV (Fig. 4e), which could likely be attributed to the aMV's translocation via respiration. In contrast, no obvious signal was detected outside the nasal cavity for the aMV@GEL group, again supporting the GEL's contribution for enhancing the retention of aMV

in the nasal cavity. Notably, neither aMV nor aMV@GEL entered the brain (Fig. S22), excluding the possibility of aMV entering the brain from the olfactory nerve upon intranasal administration[42].

Finally, we tested the protection performance in vivo by using a transgenic mouse model that was previously shown to express human ACE2 in the nasal cavity and lung[43]. We divided the mice into four groups, which were administered PBS, aMV, GEL, or aMV@GEL and then challenged with SPV (wild-type) aerosols (Fig. 4f). On Day 3, infection statuses in the nasal cavity and lung were investigated by monitoring the signal of GFP that was expressed by SPV. As expected, mice in the PBS group that received SPV challenge indeed exhibited a large area of infected GFP signal (green) in the nasal mucosa and around the alveoli of the lung (Fig. 4g, h). The infected signal appeared slightly in the nasal cavity and lung of the aMV group, owing to the virus entrapment ability of aMV in the nasal cavity and even the lung in which aMV translocated. In contrast, in the GEL group, a large area of infected signal appeared in the nasal cavity, but a small area appeared in the lung, indicating that GEL could intercept viral aerosols in the nasal cavity but was helpless to inactivate the virus. By combining aMV and GEL, the infected GFP signals neither appeared in nasal cavity nor lung of aMV@GEL group, which should be attributed to the viral aerosol interception of GEL (Fig. S23) and virus inactivation of aMV in nasal cavity. In another experiment in which mice were challenged with SPV variant (B.1.1.529, Omicron) aerosols, corresponding GFP expression analysis in the nasal cavity and lung also showed the similar protective effect of aMV@GEL (Figs. 4i and S24).

### Preparation of sMV@GEL and its performance against influenza A

Inspired by the excellent capacity of aMV@GEL to protect against SPV and its variant, we next explored whether this general approach using an intranasal mask could work for another aerosol-transmitted virus. To this end, we next focused on influenza A virus, which is known to target sialic acid (SA)-modified proteins on the membranes of human cells in the respiratory tract for infection[26]. More specifically, we utilized the H1N1 virus as the test bed and our strategy to generate MV to entrap H1N1 virus, which was based on enhancing the level of SA on the vesicle membrane. Considering the tropism of influenza virus to α-2,6 SA in human[44,45], we herein chose α-2,6 SA for overexpression on MV. We achieved this in 293 T cells by overexpressing a Golgi-localized Type II membrane protein (β-galactoside α-2,6-sialyltransferase 1, gene: *ST6GAL1*; those cells were denoted as ST-293T cells; Fig. 5a). This protein could catalyze the transfer of SA onto galactose-containing substrate proteins[46,47], thus enhancing the level of SA on the cell membrane.

As shown in Fig. 5b, cyanine 3 *N*-hydroxysuccinimide ester (Cy3)-conjugated Sambucus nigra lectin (SNL, a lectin that binds specifically to SA) showed strong binding on the surface of ST-293T cells rather than WT-293 cells, demonstrating the successful overexpression of SA

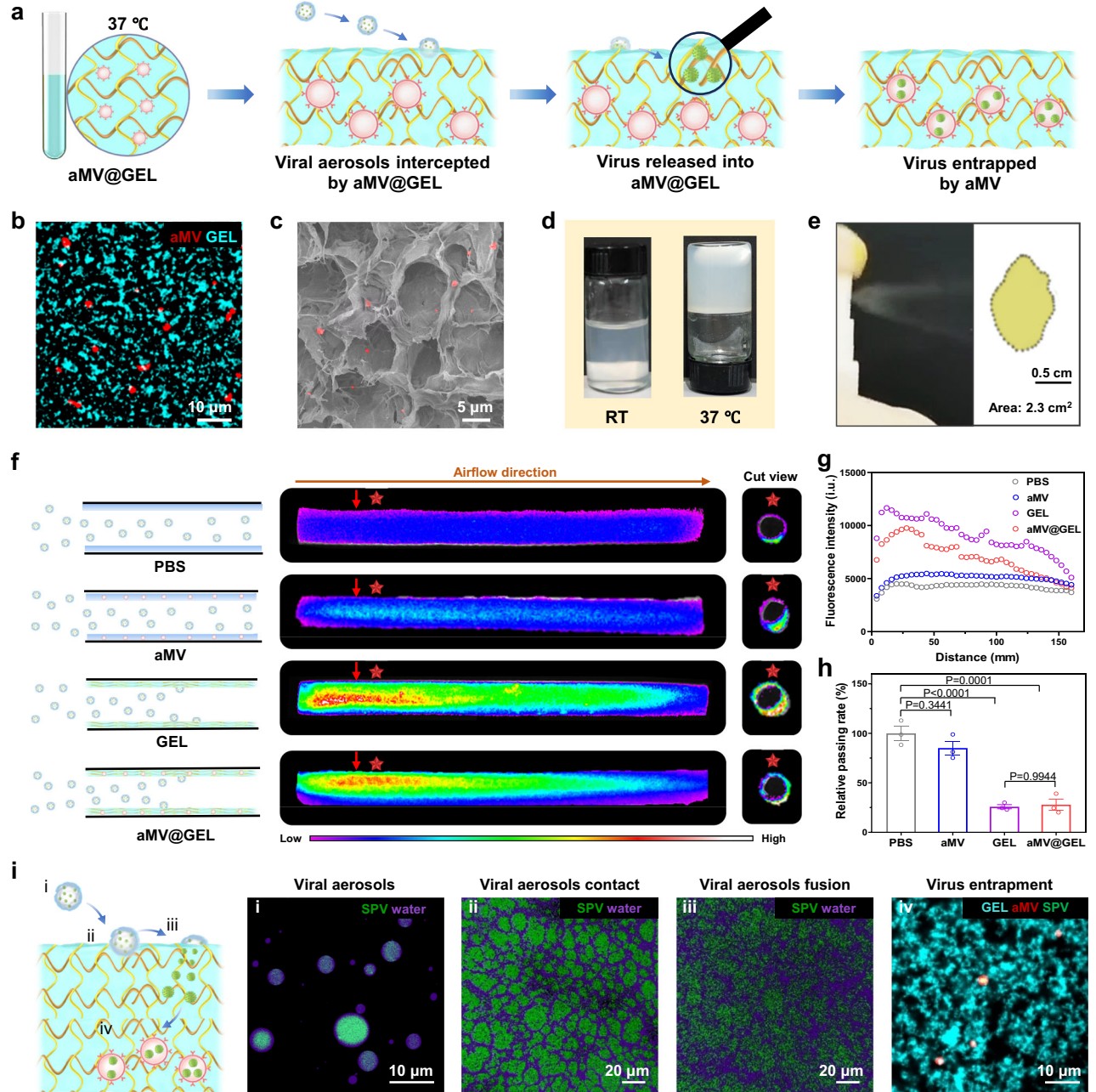

**Fig. 3 | Characterization and protective effect of chitosan/β-sodium glycerophosphate hydrogel loaded with aMV. a** Schematic for the viral aerosol interception mechanism of chitosan/β-sodium glycerophosphate hydrogel (GEL) loaded with aMV (aMV@GEL). **b** CLSM image of aMV@GEL showing aMVs were embedded in the network of GEL. Red: DiD labeled aMV; Cyan: FITC labeled GEL. **c** Scanning electron microscope (SEM) image of aMV@GEL. aMVs were marked with false color (red). **d** Photographic images of aMV@GEL showing aMV@GEL was in the liquid state at room temperature (RT, 25 °C) and transformed to the gel state at 37 °C. **e** Photographic image of spray characteristic of aMV@GEL and corresponding spray area formed on the paper 10 cm away from the sprayer. **f** Schematic design for evaluating the interception effect (left), fluorescence images of the tube precoated with PBS, aMV, GEL, or aMV@GEL after nebulizing cyanine 5 N-hydroxysuccinimide ester (Cy5) stained SPV (wild-type) aerosols into the left entrance end (middle), and corresponding fluorescence images of cut view site (indicated by

the red arrow, right). **g, h** Fluorescence quantitative analysis of different sites in tubes of **f** (**g**) and relative passing rate of SPV aerosols through these tubes (**h**). **i** Schematic and corresponding CLSM images illustrating the detailed process of SPV (wild-type) aerosol interception and virus entrapment by aMV@GEL. (i) The drops of viral aerosols consisting of water and SPV. (ii) The top view of viral aerosols contacting the aMV@GEL (0 min). (iii) The top view of viral aerosols fusing with aMV@GEL (10 min). (iv) The entrapment of the released SPV by aMV inside GEL (60 min), which image was captured at a distance of 15 μm from the surface of the GEL. Purple: fluorescein sodium-stained water in viral aerosols; Green: cyanine 3 N-hydroxysuccinimide ester (Cy3) stained virus in viral aerosols; Red: Cy5 stained aMV; Cyan: FITC-stained gel. Quantitative data in **h** represents as mean ± S.E.M., n = 3 biologically independent experiments. Statistical significance was calculated using one-way ANOVA with multiple comparison tests. All significant P-values are indicated. Source data are provided in the Source data file.

on the cell membrane. Upon treating the cells with cytochalasin B, we observed the expected formation of MV on the cell surface (Fig. 5c), which we separated to generate sMV in a reproducible way (Figs. 5d, S25, and S26). After experimentally confirming that sMV could entrap

H1N1-CA07 (Fig. 5e), we conducted a classical plaque assay with classical host cells (Madin-Darby Canine Kidney cells, MDCK cells)[47]. Owing to the low expression of SA, nMV showed a general performance in protecting the cell from the H1N1 virus (Fig. 5f, g). In sharp contrast,

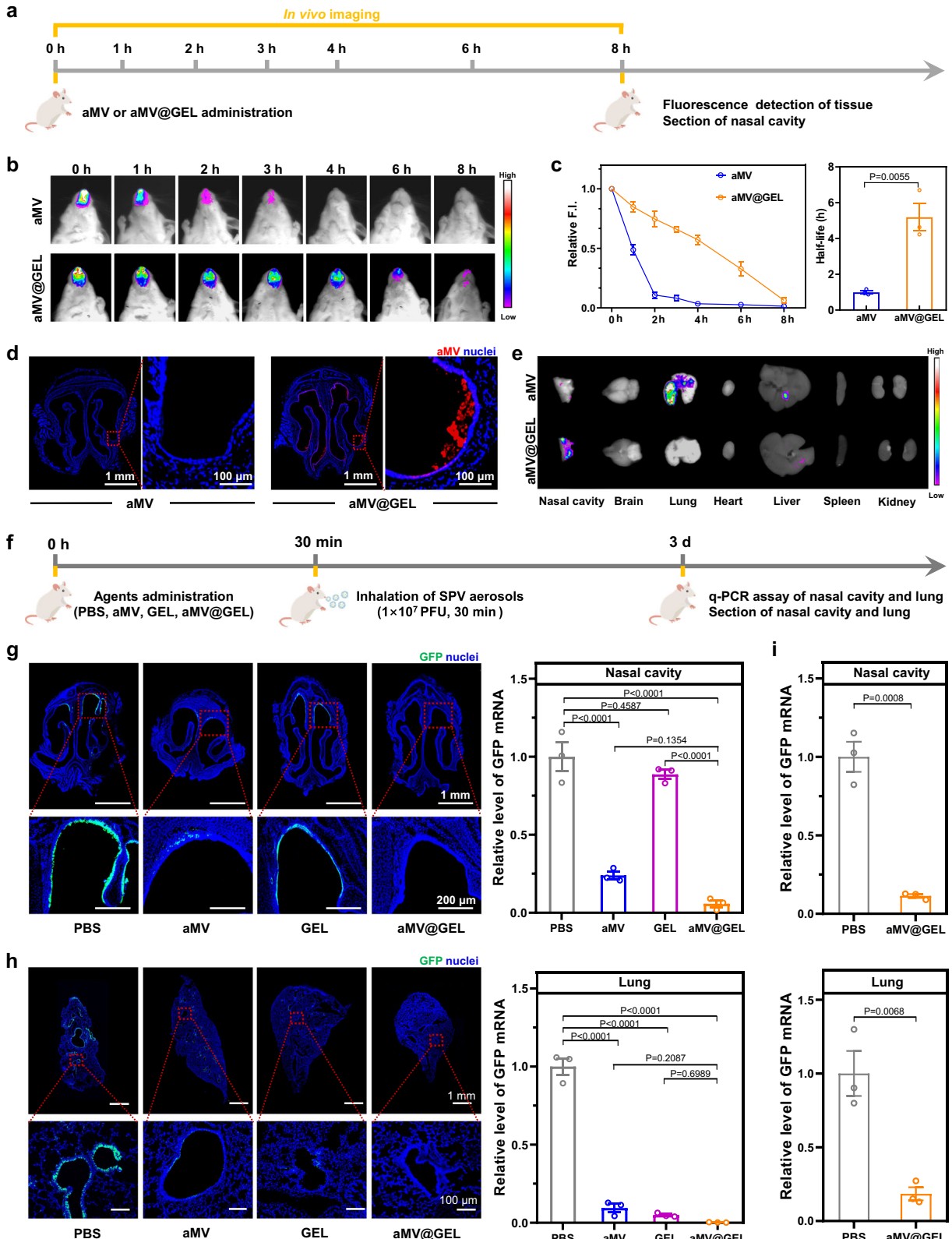

sMV significantly decreased the H1N1-CA07 infection-induced plaque area in a dose-dependent manner. Furthermore, sMV also achieved a similar protective effect against the H1N1-PR8 virus (Fig. S27), indicating the applicability of sMV@GEL for another typical H1N1 serotype.

Having demonstrated that sMV worked as designed in vitro, we continued to prepare sMV@GEL (Fig. 5h) and test its protective performance in vivo. Briefly, PBS or sMV@GEL was applied to the nasal cavity of mice, which were then challenged with low-dose H1N1-CA07 aerosols or high-dose (5-fold low dose) H1N1-CA07 aerosols, separately (Figs. 5i and S28a). In the low-dose challenged mice, we measured the mRNA of H1N1-CA07 in the nasal cavity and lung on Day 7 and found that viral mRNA in the sMV@GEL group was significantly lower than that in the PBS group (Fig. 5j). The immunohistochemical (IHC) analysis showed that the N protein of H1N1-CA07 (brown) obviously

**Fig. 4 | Distribution of aMV@GEL after intranasal administration and its in vivo protective effect against SPV aerosols. a** Schematic diagram for evaluating the distribution of free aMV and aMV@GEL after intranasal administration. **b** Representative in vivo fluorescence images of mice at the indicated time points after intranasal administration of free aMV or aMV@GEL. aMV in both groups were stained with Cy5. **c** Relative fluorescence intensity (F.I.) of free aMV or aMV@GEL in mouse nasal cavity in **b** (left) and corresponding half-life (right). **d** Representative frozen section images and enlarged views of the mouse nasal cavity at 8 h after intranasal administration of free aMV or aMV@GEL. Red: Cy5 stained aMV; Blue: DAPI stained cell nuclei. **e** Representative ex vivo fluorescence images of various tissues (nasal cavity, brain, lung, heart, liver, spleen, and kidney) at 8 h after intranasal administration of free aMV or aMV@GEL. aMV in both groups were stained with Cy5. **f** Schematic diagram for evaluating the in vivo protective effect of aMV@GEL against SPV aerosols. **g** Representative frozen section images and

enlarged views of mouse nasal cavity (left) and relative level of GFP mRNA in the nasal cavity by using q-PCR detection (right) after 3 days of challenge with SPV (wild-type) aerosols. GFP was produced by SPV-infected cells; Blue: DAPI stained cell nuclei. **h** Representative frozen section images and enlarged views of mouse lung (left) and relative level of GFP mRNA in the lung tissue by using q-PCR detection (right) after 3 days of challenge with SPV (wild-type) aerosols. GFP was produced by SPV-infected cells; Blue: DAPI stained cell nuclei. **i** Relative level of GFP mRNA that was produced by SPV-infected cell in nasal cavity and lung using q-PCR detection after 3 days of challenge with SPV (S protein B.1.1.529 mutation (Omicron)) aerosols. Quantitative data in **c**, **g**, **h**, and **i** represent as means ± S.E.M., $n = 3$ biologically independent mice. Statistical significance was calculated using two-tailed unpaired t-test (**c**, **i**) and one-way ANOVA with multiple comparison tests (**g**, **h**). All significant $P$-values are indicated. Source data are provided in the Source data file.

appeared in the nasal mucosa and around the alveoli in the PBS group, whereas very few N protein signals were present in both sites of the sMV@GEL-treated mice (Fig. 5k, l). Furthermore, H&E staining analysis showed nasal mucosa epithelial damage and pulmonary inflammatory infiltration in the PBS group, while neither were detected in the sMV@GEL group (Fig. 5k, l). In addition, compared with the PBS group, the inflammatory cytokines in the lung of the sMV@GEL group were significantly decreased (Fig. S29), indicating that the lung inflammation caused by viral infection was significantly relieved under sMV@GEL protection. Regarding high-dose challenged mice, although the weight of both groups decreased, compared to the PBS group, mice in the sMV@GEL group started losing weight later and recovered more quickly (Fig. S28b), which should be attributed to the substantially reduced viral loading upon sMV@GEL protection. Moreover, four of the six mice in the PBS group died from Day 6, whereas no deaths occurred in the sMV@GEL group over 14 days (Fig. S28c). Similar protective effects of sMV@GEL were also observed in mice challenged with H1N1-PR8, demonstrating that the protective effect of sMV@GEL could be extended to another typical H1N1 serotype (Fig. S30).

Encouraged by the above excellent protective effect of sMV@GEL against virus infection, we were also interested in the performance of sMV@GEL for preventing host-host transmission. To this end, we designed a cage divider, which allowed airflow but no direct contact or fomite (including diet and bedding) transmission (Fig. 5m). Donor mice were infected with H1N1-CA07 virus on Day 0 by intranasal administration. On Day 3, two donors were introduced into one side of the cage divider. Meanwhile, two sMV@GEL treated intermediate recipients (IRs) and two PBS-treated IRs were placed on the other side of the divider (downflow from the donor mice). On Day 6, the half of IRs were sacrificed for viral-loading assay, and the rest of IRs were introduced into a new cage, wherein the cage divider separated two sMV@GEL IRs or two PBS IRs from two terminal recipients (TRs, naïve mice, downflow from the IR mice). On Day 9, the TRs were sacrificed for infection rate analysis. As shown in Fig. 5n, the mRNA of H1N1-CA07 in sMV@GEL protected IRs significantly lower than that in PBS IRs, indicating the strong protection provided by sMV@GEL against viral aerosol transmission. In addition, we observed that 5 TRs (5/6) exposed to PBS IRs were infected, while none of TRs exposed to sMV@GEL IRs were infected (Fig. 5o). Above findings supported that our sMV@GEL had the ability to effectively block the host-host transmission of viral aerosols.

**Prediction of MV@GEL performance in the human nasal cavity**
Thus, our in vitro and in vivo results clearly showed that the layer of aMV@GEL and sMV@GEL present in the nasal cavity of mice could protect against SPV aerosols or H1N1 viral aerosols by intercepting viral aerosols and entrapping virus. Furthermore, considering both the known physical complexity of the viral aerosol infection process and the anatomical differences between mice and humans, we performed computer simulation to assess the prospective performance of our

MV@GEL. To ensure that the digital human nasal cavity model for these simulations was representative of a real human nasal cavity structure, we initially used CT to scan the nasal cavity of a healthy adult volunteer. Then, 3D Slicer software was used to reconstruct the adult nasal cavity CT images into a 3D digital model[48] (Fig. 6a).

Next, we divided the model into the computational mesh and added it into open-source software OpenFOAM for the computational fluid dynamics-discrete particle simulation (CFD-DPS)[49–51]. The airflow through the nasal cavity was realized by the PIMPLE algorithm[52], which was a merged PISO-SIMPLE (pressure implicit split operator-semi-implicit method for pressure-linked equations) algorithm[53,54], and the positions of discrete aerosols were updated by Newton's second law. The respiratory rate followed a sinusoidal form with time, with the peak inhalation flow rate set as 0.64 m/s, equivalent to 10 L/min. The discrete viral aerosols were regarded as monodisperse and spherical particles with the density of liquid water (1000 kg/m³), which were added into the nasal cavity at 6,000 per second in a period of inhalation with the same velocity as that of the airflow. In addition to the drag force exerted by fluid, the buoyancy, and the gravity of the viral aerosols, the interaction between viral aerosols and MV@GEL was calculated. Electric force could be regarded as the vector sum of all small Coulomb forces between aerosol and its neighboring MV@GEL elements that were uniformly embedded on the nasal wall and was implemented by the C++ program (Fig. 6b). Upon obtaining the abovementioned forces, the velocity and position of the viral aerosols were updated immediately by Newton's second law, and viral aerosols were considered to be captured when they reached the nasal wall. These calculations thus enabled us to compare the unprotected case and situation with applied MV@GEL during a given viral aerosol.

We calculated the distribution of viral aerosols in the nasal cavity for a total respiratory cycle of 3 s, including an inhalation process for the first 1.5 s and an exhalation process from 1.5 s to 3 s. After 1.5 s of inhalation, the unprotected situation showed that approximately 55.2% of the viral aerosols (blue dots) entered the downstream trachea from the nostril with the airflow. In contrast, a large number of aerosols accumulated in the anterior part of the nasal cavity, and only 6.8% of the viral aerosols entered the downstream trachea from the nostril (Fig. 6c). For a high-precision analysis of the simulated viral aerosols in the nasal cavity, we extracted and observed five different cross sections of the nasal cavity after 1.5 s of inhalation (Fig. 6d). In an unprotected situation, the viral aerosols (blue dots) were evenly distributed in the lumen and the wall. Once MV@GEL was applied, almost all viral aerosols attached to the nasal wall, especially in the anterior section.

Next, we focused on the exhalation process. Viral aerosols that had flowed into the downstream trachea in the inhalation process flowed back into the nasal cavity with the airflow of exhalation. We found that almost all exhalant viral aerosols (red dots) in the nasal cavity had already been intercepted by MV@GEL at 2.1 s (Fig. 6e). In contrast, the exhaled viral aerosols remained fully within the whole nasal cavity in the unprotected situation. After one respiratory cycle, the unprotected

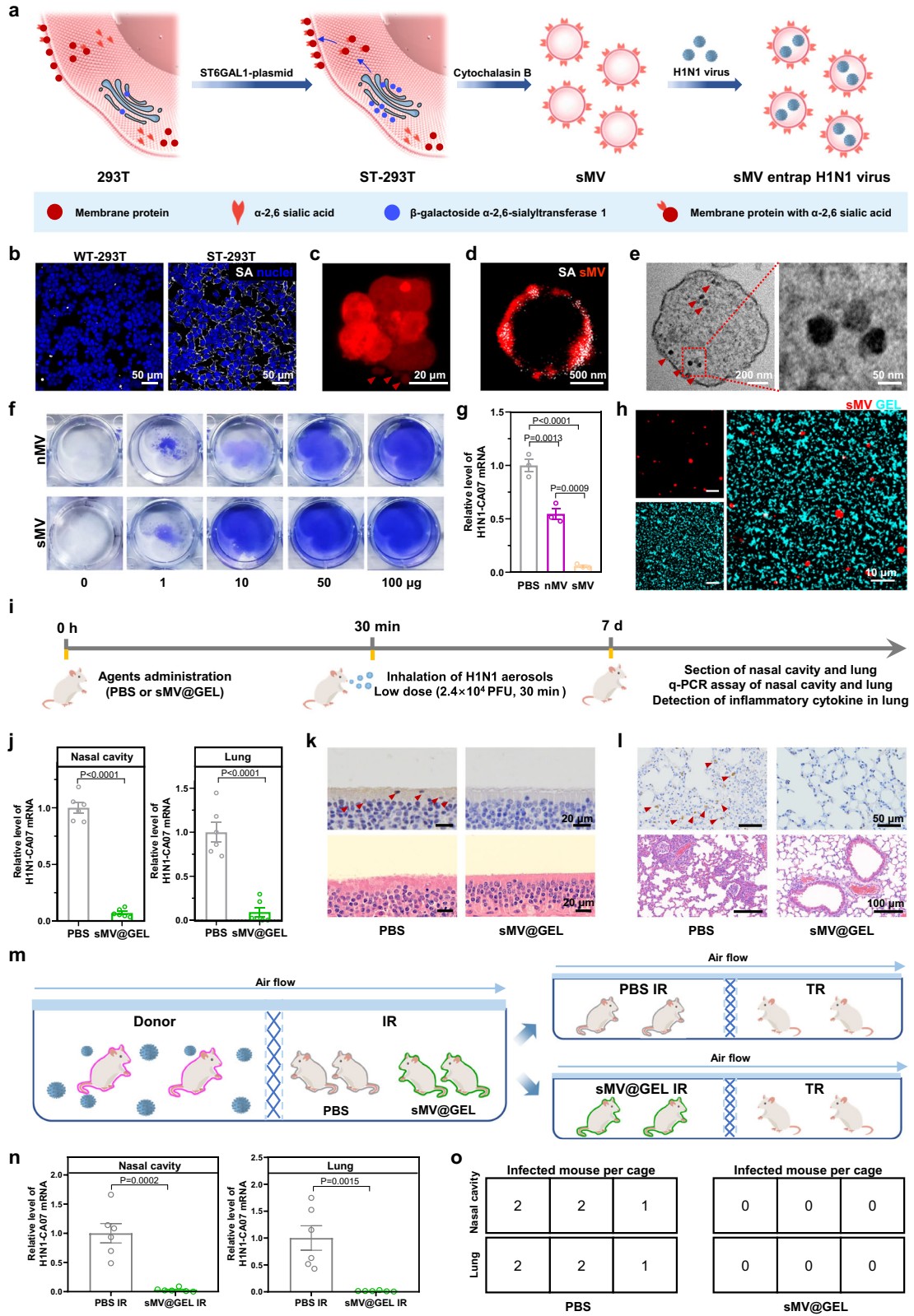

situation showed that 23.6% of the exhalant viral aerosols remained in the nasal cavity (Fig. S31), leading to the increased possibility of nasal infection. More importantly, the majority of the exhalant viral aerosols were expelled into the atmosphere and could cause a secondary transmission of viral aerosols. However, almost no viral aerosol flowed out to the atmosphere at the end of the breath when MV@GEL was applied. Note that we extended above simulation to a child's 3D digital

model and observed much similar result (Fig. S32), showing the applicability for protecting children from the viral aerosols infection.

## Establishment of an integrated HRT model and confirmation of sMV@GEL's protective effect

Although our simulation provided a line of evidence supporting the benefit of applying MV@GEL materials to the human nasal cavity to

**Fig. 5 | Preparation of GEL loaded with α-2,6 sialic acid-overexpressing MV (sMV@GEL) and its protective effect against H1N1-CA07 aerosols. a** Schematic for the preparation and H1N1 virus entrapment mechanism of MV with α-2,6 sialic acid-overexpression (sMV). **b** CLSM images showing SA expression on different cells. White: sialic acid (SA); Blue: nuclei. **c** CLSM image of CFSE stained ST-293T cells (red) after 30 min of cytochalasin B treatment. **d** CLSM image showing SA expression on sMV. White: SA; Red: sMV. **e** TEM images of ultrathin section of sMV after 24 h of challenge H1N1-CA07. The red arrow indicated H1N1-CA07. **f** Representative photographs of Madin-Darby canine kidney (MDCK) cells after 3 days of challenge with H1N1-CA07 under different dose of nMV or sMV protection. **g** Relative level of H1N1-CA07 mRNA in MDCK cells of different groups (PBS, 50 μg nMV, and 50 μg sMV). **h** CLSM images of sMV@GEL showing sMV were embedded in the network of gel. Cyan: GEL in sMV@GEL; Red: sMV in sMV@GEL. **i** Schematic diagram for evaluating the in vivo protective effect of sMV@GEL against H1N1-CA07 aerosols in mice. **j** Relative level of H1N1-CA07 mRNA in the nasal cavity (left) and lung (right) in the PBS group and sMV@GEL group after 7 days of challenge with low dose of H1N1-CA07 aerosols. **k, l** Representative IHC (upper) and H&E (lower) sections of the nasal cavity (**k**) and lung (**l**) in PBS group (left) and sMV@GEL group (right) of **j**. The red arrows: N protein of H1N1-CA07 (brown). **m** Schematic of H1N1-CA07 transmission model. **n** Relative level of H1N1-CA07 mRNA in the IR nasal cavity (left) and lung (right) after 3 days exposed to donor. **o** Number of infected TR in each cage after 3 days exposed to PBS IR (left) or sMV@GEL IR (right). Data in **g, j** and **n** represent as means ± S.E.M., *n* = 3 biologically independent experiments (**g**), *n* = 6 biologically independent mice (**j, n**). Statistical significance was tested with two-tailed unpaired *t*-test (**j, n**) and one-way ANOVA with multiple comparison tests (**g**). All significant *P*-values are indicated. Source data are provided in the Source data file.

confer protection against viral aerosols, we still wanted to confirm this result in a human-related experiment. Pursuing this, we used 3D printing technology and the aforementioned CT data from the adult volunteer to fabricate a realistic human nasal apparatus as the "nasal module". Meanwhile, we also prepared lung organoids from the paracancerous tissue of a lung cancer patient as the "lung module", which were reported with similar cell types, cellular organization and function to the lung tissue and could be used for evaluating the lung infection in several studies[55–57]. To mimic human respiratory, we used a pump to provide respiratory flow as the "respiratory module" (Fig. 7a, b). With the three modules in hand, we obtained the integrated HRT model by placing the lung module into a container with the following vents: one vent was connected to the nasal module via a sterile tube, and the other was connected to the respiratory module. Utilizing this integrated HRT model, we were thus able to (i) to spray MV@GEL into the nasal cavity, (ii) to recapitulate the flow process of viral aerosols from the nose to the lung, and (iii) to monitor the infection in human lung organoids (Fig. 7b).

We firstly determined that our lung organoids exhibited the same markers (FOXJ1, ciliated cell marker; SCGB1A1, club cell marker; P63, basal cell marker; and SA, H1N1 receptor) as paracancerous lung tissue[55–57] (Figs. 7c and S33), which indicated that our lung organoids composed of similar cell types with those in lung tissue and could be used for evaluating the protection performance against H1N1 viral aerosol. Then we applied PBS or sMV@GEL as a coating in the inner wall of the nasal module and used an aerosol generator to nebulize H1N1-CA07 viral aerosols around the nostril. Meanwhile, the respiratory module started to inhale and exhale those viral aerosols with the respiratory rate, upon which the viral aerosols through the nasal module could reach the "lung module", enter the culture medium and subsequently infect lung organoids. After 24 h and 48 h of culture, the protection effects of the two groups on lung organoids were detected.

As the 3D reconstructed images of the organoids showed in Fig. 7d, the lung organoids in the PBS group exhibited an obvious signal of N protein of H1N1-CA07 virus (green) at 24 h, and a greater extent observed at 48 h, indicating that the virus had infected lung organoids and copied themselves. In sharp contrast, the lung organoids in the sMV@GEL group did not show any N protein signal even for 48 h. The expression of H1N1-CA07 mRNA in lung organoids of the sMV@GEL group was also significantly lower than that in the PBS group at 24 h and 48 h (Fig. 7e). Consequently, lung organoid in the PBS group showed substantially reduced cell viability, while lung organoid in the sMV@GEL group had no sign of either apoptosis or death (Fig. S34). Similar results were obtained when we repeated this experiment with H1N1-PR8 viral aerosols (Fig. 7f) or with child-derived HRT model (nasal module updated with a realistic child nasal apparatus, Fig. S35). Thus, beyond supporting our conclusions from the CFD simulation, these results from our integrated HRT model provided strong human-related evidence that MV@GEL conferred effective protection against infection by viral aerosols.

## Discussion

We have shown that with our MV@GEL system, the thermosensitive and positively charged hydrogel could be easily applied into the nasal cavity and intercept negatively charged viral aerosols; the MV with the viral receptor that was embedded into the hydrogel could be retained in the nasal cavity for 8 h and effectively entrapped the viruses that were released from aerosols. Utilizing the mouse model, CFD simulation, and integrated HRT model, we demonstrated that the application of our MV@GEL system dramatically reduced the likelihood of respiratory viral infection. This concept of an "intranasal mask" could meet the protection requirement against viral aerosols in the daily life of the general population and could provide effective protection for some individuals who cannot conveniently wear face masks, such as patients with asthma. Moreover, aiming at high-risk individuals, such as doctors and nurses, our intranasal mask could also be combined with face masks to further reduce the risk of infection from aerosols containing threatening viruses (Fig. S36), such as SARS-CoV-2, H1N1, and SARS.

Some researchers have developed multiple antibodies to fight against various types of viruses, while most antibodies may lose the ability to bind viruses as viruses exhibit high variability, especially Omicron[58,59]. Inspired by viral receptor-mediated host cell infection, nanovesicles and cell membrane-coated particles displaying viral receptors have recently emerged as new tools to bind viruses and further suppress viral infection[29–34,60]. It should be emphasized that the ultrasmall size of the nanovesicles (e.g., ~100 nm for exosome) or the solid core of the membrane-coated particles (e.g., polymeric particle coated with ACE2-membrane) were unfavorable for harnessing the potential of entrapping virus inside, which failed to completely isolate the virus from host cells. In the present study, we rationally prepared MV displaying viral receptors and demonstrated their virus entrapment ability, which could complete shielding the virus and could further reduce the infection risk.

Some researchers had also utilized mucoadhesive polymers (such as carrageenan and gellan gum) to develop an intranasal barrier against virus[61–64]. Typically, those polymers could wrap around the cell to form a physical barrier on the surface of epithelium cells in the nasal cavity, thus reducing the contact between the cell and virus. However, these studies focusing on the virus within the nasal cavity failed to prevent virus entering the lung through airflow. In our study, we introduced a positively charged hydrogel that not only prevented infection in the nasal cavity but also intercepted viral aerosols, thus effectively preventing their entry into the lung and substantially reducing the risk of lung infection. Further incorporation of MV displaying viral receptors in our study resulted in a notable reduction of infection rates within both the nasal cavity and lung.

Beyond SARS-CoV-2 and influenza A virus, the vesicles in our research have the potential to be applied to extended viruses. For example, we can prepare vesicles from host cells that express dipeptidyl peptidase 4 receptor, which could be used for entrapping Middle

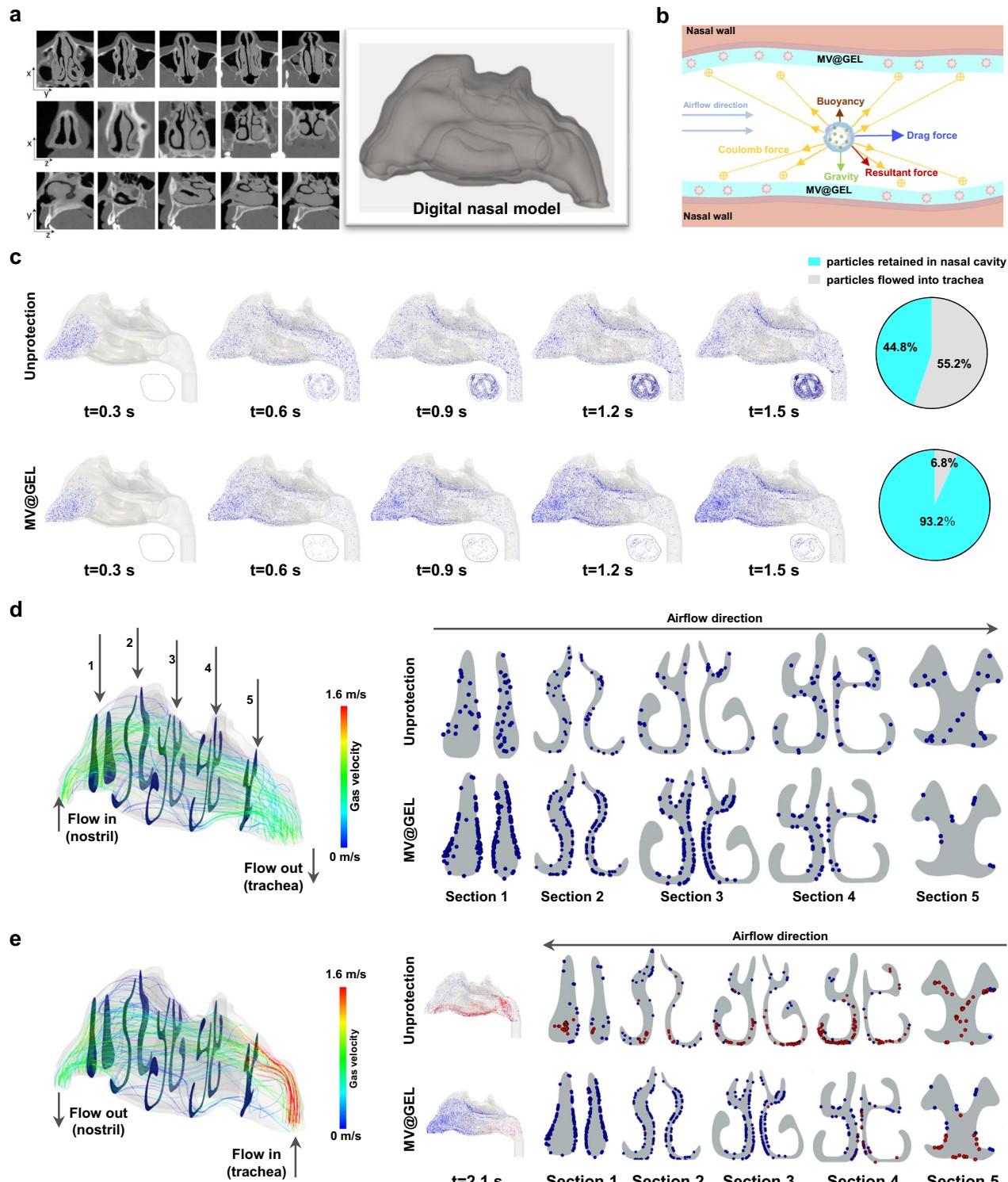

**Fig. 6 | Prediction of viral aerosol interception effect of MV@GEL in human digital nasal cavity by using computational fluid dynamics-discrete particle simulation (CFD-DPS). a** Representative CT images of the human nasal cavity and its corresponding reconstructed 3D digital model. **b** Schematic diagram of the charge interaction between MV@GEL and aerosols in the nasal cavity during simulation calculation. Each tiny part on the nasal wall could generate a tiny Coulomb force to the specific viral aerosol. Meanwhile, the drag force of fluid, buoyancy, and gravity were considered, and the vector sum was the resultant force it received. **c** Distribution of inhaled viral aerosols (blue dot) at different time points (0.3 s, 0.6 s, 0.9 s, 1.2 s, and 1.5 s) of unprotection situation (upper) or MV@GEL

situation (lower) after the beginning of inhalation (left), and the corresponding percentage of viral aerosols that retained in the nasal cavity or flowed into trachea at 1.5 s (right). **d** Flow field state of inhalation airflow (left) and the distribution of inhaled viral aerosols (blue dot) in different cross sections under unprotection situation or MV@GEL situation after 1.5 s inhalation (right). **e** Flow field state of exhalation airflow (left), the distribution of viral aerosols at 2.1 s (1.5 s inhalation followed by 0.6 s exhalation) under unprotection situation or MV@GEL situation (middle), accompanied with corresponding viral aerosols distribution in different cross sections (right). The inhaled viral aerosols were blue dots, and the exhalant viral aerosols were red dots.

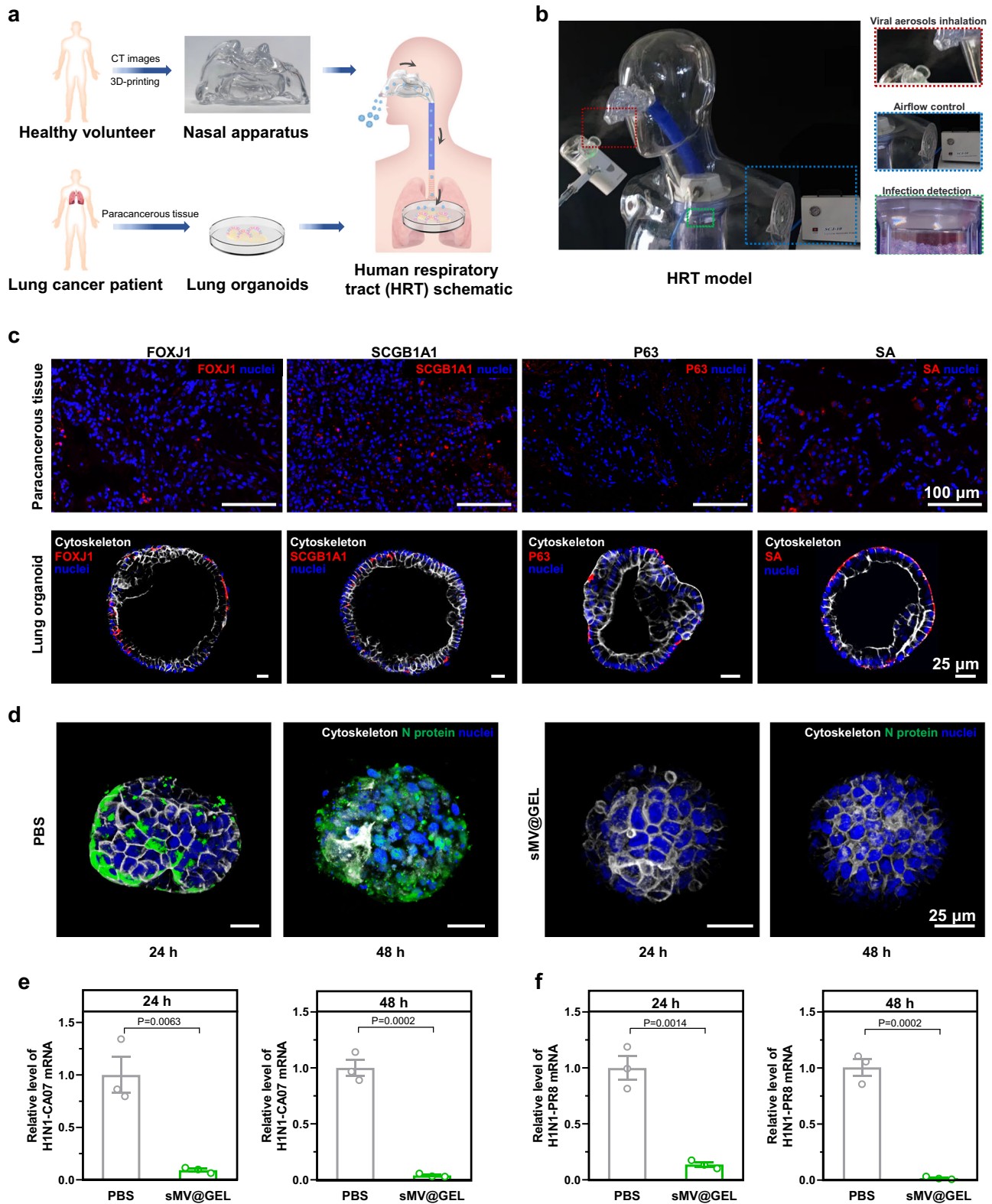

East respiratory syndrome (MERS) virus[65]. In addition, gene editing can potentially be used to induce the expression of multiple types of viral receptors in cells to obtain a "super vesicle" that could confer protection against multiple viruses at the same time. Recent studies have reported that the influenza A virus may circulate simultaneously with SARS-CoV-2, leading to more serious respiratory diseases and causing significant concerns in clinical treatment[4]. Aiming at the above problem, we can prepare a super vesicle that co-overexpress ACE2 and SA,

which holds the potential to entrap both influenza A virus and SARS-CoV-2, thus avoiding the severe respiratory diseases induced by coinfection of the above viruses.

As for the hydrogel, we screened out an ideal thermosensitive hydrogel that exhibited outstanding performance in terms of spraying ability, retention in the nasal cavity, viral aerosol interception capacity, and undetectable cytotoxicity. Considering the thermosensitive property, some points should be noted. First, our hydrogel should be

**Fig. 7 | Confirmation of the protective effect of sMV@GEL against H1N1 viral aerosols by using the human respiratory tract model. a** Schematic diagram of the experimental design. We used the CT data from the volunteer and 3D printing technology to fabricate a realistic human nasal apparatus. Meanwhile, we cultured human lung organoids from the paracancerous tissue of a lung cancer patient. To imitate the human respiratory tract (HRT), we constructed an integrated HRT model by placing the lung organoids into a container with the following vents: one vent was connected to the nasal apparatus via a sterile tube, and the other was connected to a pump for providing respiratory airflow. **b** Full-view photograph of the integrated HRT model and magnified photographs of three important modules, showing the inhalation of the viral aerosols near the nasal apparatus (red frame), the airflow control by a pump (blue frame), and the lung organoids container for infection detection (green frame). **c** CLSM images showing FOXJ1 (ciliated cell marker), SCGB1A1 (club cell marker), P63 (basal cell marker), and SA (H1N1 receptor) expression in paracancerous tissue (upper) and lung organoids (lower). FOXJ1, SCGB1A1, and P63 were stained with corresponding antibodies and secondary Alexa Fluor® 647 fluorescent antibody, SA was stained with Cy3-SNL (red, false color). White: FITC-phalloidine stained cytoskeleton; Blue: DAPI stained cell nuclei. **d** Representative 3D reconstructed CLSM images of lung organoids in different groups after 24 h or 48 h of challenge with H1N1-CA07 aerosols. White: FITC-phalloidine stained cytoskeleton; Blue: DAPI stained cell nuclei. Green: the fluorescent antibody-stained N protein of H1N1-CA07. **e** Relative level of H1N1-CA07 mRNA in lung organoids of different groups after 24 h or 48 h of challenge with H1N1-CA07 aerosols. **f** Relative level of H1N1-PR8 mRNA in lung organoids of different groups after 24 h or 48 h of challenge with H1N1-PR8 aerosols. Data in **e** and **f** represent the means ± S.E.M., $n = 3$ biologically independent experiments. Statistical significance in **e** and **f** was tested with two-tailed unpaired t-test. All significant $P$-values are indicated. Source data are provided in the Source data file.

stored at temperature below 25 °C to avoid unexpected gelation. Moreover, people should avoid extremely cold environments or torrid environments when using our intranasal mask, because the excessively low ambient temperature (below 8 °C) may delay the gelation time in the nose and the long-time exposure to excessively high ambient temperature (over 38 °C) may lead to unexpected gelation in the container. To further upgrade our hydrogel formulation, we can use some special hydrogels with shear-thinning properties, in which gelation is independent of the temperature. For example, peptide-based hydrogels, protein-based hydrogels, and hydrogels from blends can be sprayed into the nasal cavity under shear stress and immediately recover their mechanical properties[66,67], thus remaining fixed at the inner wall of the nasal cavity. In addition, considering the 8 h intranasal retention time of our hydrogel, effort can be made to upgrade the gel component, such as gellan gum with higher mucoadhesion[68,69], for individuals with long-term exposure in the high-risk environment.

Moreover, there are some notable points regarding the translational potential of the MV@GEL system. On one hand, the system is used for topical application, which do not require HLA matching or individualization of MV. On the other hand, chitosan and β-sodium glycerophosphate are materials approved by the Food and Drug Administration, exhibiting high biocompatibility for intranasal use. Given that many studies have demonstrated the tolerance of cellular vesicles during lyophilization and that both chitosan and β-sodium glycerophosphate are very stable, we propose that the constituent materials of the system can be stored in separate containers for a long time and should simply be mixed before use. Therefore, our MV@GEL exhibits good potential from the bench to the public, providing a mild and effective shield for healthy and high-risk people against viral aerosols.

Despite the valuable findings presented in our study, it is important to acknowledge certain limitations that should be addressed in future research. Although the ACE2 transgenic mouse model was wildly used, easily acquired, and cost-effective model for SARS-CoV-2 infection, it still has some limitations[43,70-72], including the virus tropism, susceptibility to deadly diseases, and immune system differences. Syrian hamster, ferret, and non-human-primate with natural expression of ACE2 can be considered as preferable models in the studies focusing on virus transmission, pathology, and host immune responses. Moreover, due to the constraints of our cooperative P3 laboratory, we employed SPV aerosols instead of authentic virus aerosols in our SARS-CoV-2 infection experiment, which might not fully replicate real-world scenarios. Alternatively, we employed authentic H1N1 viral aerosols to provide convincing evidences for MV@GEL's protection effect in real-world scenario. In addition, our sMV@GEL focused solely on the α-2,6 SA receptors, and future studies could engineer vesicles overexpressed α-2,3 SA receptors or overexpressed both α-2,6 and α-2,3 SA receptors to investigate the antiviral effects across different species, thus providing guidance for the choice of receptors in different application scenarios.

## Methods
### Study approval
The mouse experiments were approved by the Institutional Animal Care and Use Committee at the Institute of Process Engineering, Chinese Academy of Sciences (Number: IPEAECA2020042).

Human research involved in this study was approved and performed in accordance with the institutional guidelines of the Peking University First Hospital (Number: 2021-S-486), and informed consents were obtained from the participants (male volunteer who provided CT data of human nasal cavity and female volunteer who donated lung tissue).

### Reagents and materials
Anti-human ACE2 (ab108252; 1:100 dilution; clone: EPR4435(2)), Anti-N protein of H1N1 (ab104870; 1:100 dilution; clone: Polyclonal), and Goat Anti-Mouse IgG H&L (Alexa Fluor® 647) (ab150115; 1:1000 dilution; clone: Polyclonal) were purchased from Abcam. Anti-SCGB1A1 (DF3950; 1:100 dilution; clone: Polyclonal), Anti-P63 (DF6860; 1:100 dilution; clone: Polyclonal), and Anti-FOXJ1 (AF0372; 1:100 dilution; clone: Polyclonal) antibodies were purchased from Affinity Biosciences.

Cy3-SNL was purchased from Vector Laboratories. Cytochalasin B, puromycin, FITC, sodium fluorescein, FITC labeled Phalloidin, β-sodium glycerophosphate, chitosan, and enzyme-linked immunosorbent assay (ELISA) kits for cytokines (IL-2, IL-6, and TNF) were purchased from Solarbio Life Sciences. RNAeasy™ Animal RNA Isolation Kit with Spin Column, ACE2 Activity Fluorometric Assay Kit, and BeyoFast™ SYBR Green One-Step qRT-PCR Kit were purchased from Beyotime Biotechnology. DiD and Cy5 were purchased from FANBO biochemicals. CSFE and TPCK-Trypsin were purchased from Thermo Fisher Scientific. Penicillin-Streptomycin, Trypsin-EDTA, and Dulbecco's modified Eagle medium (DMEM) were purchased from VivaCell, Shanghai, China. Fetal bovine serum was purchased from Gibco. Cell counting kit-8 (CCK-8) was purchased from Biosharp. pLV.CMV.hA-CE2.PGK.Puro.WPRE (human ACE2-plasmid) was purchased from PackGene Biotech. pLV-CMV-MCS-EF1-ZsGreen1-T2A-Puro (ST6GAL1-plasmid) was purchased from Fenghui Biotech.

SPV and SPV with S protein B.1.1.529 mutation (Omicron) was purchased from Langmiao Biotechnology. The primer of SPV and H1N1 virus were purchased from Sangon Biotech. Other serval variants of SARS-CoV-2 pseudovirus were gifted from Professor Youchun Wang and Professor Weijin Huang of Institute for Biological Product Control, National Institutes for Food and Drug Control, and WHO Collaborating Center for Standardization and Evaluation of Biologicals. The H1N1-CA07 strain of influenza A virus was gifted from Professor Tao Jiang of State Key Laboratory of Pathogen and Biosecurity, Beijing Institute of Microbiology and Epidemiology. The H1N1-PR8 strain of influenza A virus was gifted from Professor Meng Qin of Beijing Advanced Innovation Center for Soft Matter Science and Engineering, College of Life Science and Technology, Beijing University of Chemical Technology

and Professor Jing Sun of Institute of Chinese Materia Medica, China Academy of Chinese Medical Sciences.

### Animals

BALB/c mice (6-8 weeks, female) and C57/BL6 mice (6-8 weeks, female) were purchased from Beijing Vital River Laboratories. Transgenic mice (6-8 weeks, female) with human ACE2 expression were purchased from GemPharmatech. The mice were housed in an environmentally controlled room (23 °C, with $55 \pm 5\%$ humidity and under a 12 h–12 h light–dark cycle). All mice were randomly divided into various groups for subsequent experiments. This study was performed in strict accordance with the Regulations for the Care and Use of Laboratory Animals and Guideline for Ethical Review of Animal (China, GB/T 35892-2018).

### Cell lines and cell culture

293T cell (catalog no. CL-0005) and MDCK cell (catalog no. CL-0154) was purchased from Procell Life Science&Technology Co.,Ltd (Wuhan, China). Hnepc (catalog no. C303) was purchased from Honsun Biological Technology Co., Ltd (Shanghai, China).

293T, ACE2-293T, Hnepc, MDCK, and ST-293T cells were cultured in DMEM with 1% Penicillin-Streptomycin and 10% fetal bovine serum under a 5% $CO_2$ environment at 37 °C. Trypsin-EDTA (0.25%) was used to detach cells for subculturing every 2–3 days.

### Preparation and characterization of ACE2-293T cells

Human ACE2-plasmid was added to 293 T cells culture medium. 3 days later, 10 µg/ml puromycin was added to 293T cell culture medium for screening. After 1 day, the remaining cells were harvested. Subsequently, those ACE2-293T cells were cultured with the medium containing 10 µg/ml puromycin.

For detecting the ACE2 expression, ACE2-293T and WT-293T cells were fixed and treated with anti-human ACE2 antibody at 4 °C overnight, and then incubated with Alexa Fluor® 647 labelled secondary antibody for 1 hour at room temperature. The cell nuclei were stained by DAPI. Subsequently, above cell samples were imaged by CLSM (A1/SIM/STORM, Nikon) and detected by flow cytometry (CytoFLEX LX, Beckman Coulter).

### Preparation and characterization of MV

ACE2-293T cells were treated with cytochalasin B (10 µg/ml) and placed in a 5% $CO_2$ environment at 37 °C. After 30 min, above cells were harvested with PBS and then underwent vigorous vortex for 10 min. The suspension was centrifuged at $800 \times g$ for 30 min to remove cells, and the resultant supernatant was centrifuged at $5000 \times g$ for 30 min to harvest MV sediment. The resultant supernatant this time was further centrifuged at $100,000 \times g$ for 30 min to harvest NV sediment. Using the same way, nMV was harvested from WT-293T cells.

The particle sizes of above MV were analyzed by Dynamic Light Scattering (ZetaSizer NANO ZS90, Malvern). The morphology of above MV was characterized by TEM (Jeol, JEM-1400). For detecting the ACE2 expression, aMV and nMV were treated with anti-human ACE2 antibody at 4 °C overnight, and then incubated with Alexa Fluor® 647 labelled secondary antibody for 1 h at room temperature. The membrane of MV was stained by DiD. Subsequently, the above MV samples were imaged by CLSM (A1/SIM/STORM, Nikon) and detected by flow cytometry (CytoFLEX LX, Beckman Coulter). For quantifying the amount of ACE2, the ACE2 activity of commercial ACE2 protein, aMV and aNV was tested by ACE2 Activity Fluorometric Assay Kit (Beyotime Biotechnology).

### Entrapment of SPV by aMV

aMV was treated with SPV at 37 °C for 24 h and then was harvested by centrifugation ($5000 \times g$, 5 min). The collected sediment was fixed with 2.5% glutaraldehyde, prepared in ultra-thin section, and detected by TEM (Jeol, JEM-1400). For fluorescence detection experiment, the SPV was pre-stained with FTIC, and the collected MV sediment was further stained with DiD and detected by STED (SP8, Leica).

### The in vitro protective effect of aMV against SPV

Firstly, PBS, ACE2 protein, nMV, aNV, aMV, and aMV/anti-ACE2 (aMV was pretreated with excess anti-human ACE2 antibody overnight) were added to ACE2-293T cells ($1 \times 10^4$), separately. Then ACE2-293T cells were challenged with 10 µl of $1 \times 10^7$ PFU/ml wild-type SPV for 3 days, and then imaged by CLSM (A1/SIM/STORM, Nikon) and analyzed by flow cytometry (CytoFLEX LX, Beckman Coulter). In the similar experiments that using SPV with S protein mutation (D614G, B.1.617.1, P.2, B.1.1.7, B.1.351, P.1, B.1.617.2 (Delta), B.1.429, B.1.1.529 (Omicron)), PBS and aMV (100 µg) were added to ACE2-293T cells ($1 \times 10^4$), which were then challenged with above SPV variants (10 µl of $1 \times 10^7$ PFU/ ml).

### Preparation and characterization of GEL

According to the method in a previous work[73], 0.10 g, 0.12 g, and 0.14 g chitosan was respectively added to 7 ml ultrapure water, followed by the addition of 40 µl glacial acetic acid. After stirring overnight, 3 ml β-sodium glycerophosphate (500 mg/ml) was slowly added to above solutions under the ice bath condition. After further stirring overnight, $CS_{0.10}$, $CS_{0.12}$, and $CS_{0.14}$ were prepared and stored at 4 °C. The gelation time of $CS_{0.10}$, $CS_{0.12}$, and $CS_{0.14}$ was detected by Rheometer (MCR302, Anton-Paar) using a stainless-steel plate-plate with smooth surface (50 mm diameter) as clamp. The rheological experiment was carried out under the condition of linear regime of deformation with the constant stress (1 Pa) and frequency (1 Hz) at 37 °C. Three repeated tests for each gel were conducted. The spraying state of gel was recorded by phone (Mi10).

For investigating the SPV aerosols interception effect of $CS_{0.10}$, $CS_{0.12}$, and $CS_{0.14}$, the gel layer was firstly formed on the inner wall of a tube (length: 16 cm, diameter: 10 mm). During this process, the tube was slowly rotated to ensure uniform coverage of the coating (coating area 50.1 $cm^2$, coating thickness 60-80 µm). Then we immersed the right side of the tube in a water container and then nebulized Cy5 stained SPV aerosols ($1 \times 10^5$ PFU, 10 min) into the left side of the tube by using an aerosol generator (AER-S-NS12, TOW). We further used microplate reader (Infinite M200, TECAN) to read out the Cy5 fluorescence intensity of the water, and thus calculated the interception efficiency of $CS_{0.10}$, $CS_{0.12}$, and $CS_{0.14}$.

The interaction force between the $CS_{0.10}$, $CS_{0.12}$, or $CS_{0.14}$ and SPV was determined by quartz crystal microbalance (Q-SENSE E4, Biolin) according to a typical protocol[74]. First, spin-coating of different hydrogel solutions (10 µl) onto a silicon chip was conducted to form a membrane. The chip was then incubated at 37 °C for 1 hour to allow the hydrogel's gelation. Subsequently, the chip was installed into the instrument, and pure water was continuously pumped at the speed of 200 µl/min to remove non-immobilized hydrogel. When the frequency curve reached a plateau, the SPV solution was pumped at the speed of 50 µl/min to interact with the hydrogel, and the frequency curve was monitored until it reached a new plateau. Finally, pure water was pumped again to remove dissociated SPV until the frequency curve reached a plateau.

After adding 10 µl gel to cells for 1 hour, the cell viability was measured using CCK-8 kit. The effect of $CS_{0.10}$, $CS_{0.12}$, and $CS_{0.14}$ on the tight junction of the cytoskeleton was tested by the following experiment: we formed the gel layer on the Hnepc cells for 1 h, and then stained the cells with FITC-phalloidin and DAPI, following with the CLSM (A1/SIM/STORM, Nikon) detection.

### The simulation of virus diffusion in aMV@GEL

In simulation of virus diffusion in aqueous solution and GEL, GEL was regarded as gel-framework filled with aqueous solution and virus was considered diffusing freely in pores of GEL. Then, the virus solution

was added on the top surface of aqueous solution or GEL. Following the assumption of continuous media, virus diffusion could be governed by convection-diffusion equation:

$$\frac{\partial c}{\partial t} + \mathbf{u} \cdot \nabla c = D\nabla^2 c \qquad (1)$$

Here convection effect was neglected and the virus velocity $\mathbf{u}$ was zero. $c$ denoted dimensionless concentration, and $D$ represented diffusion coefficient that could be described by Stokes-Einstein equation:

$$D = \frac{kT}{6\pi\mu r} \qquad (2)$$

where $k$ was Boltzmann constant, $T$ was temperature and set as 300 K, $\mu$ was viscosity of water, and $r$ denoted radius of virus and was set as 0.1 μm. Here lattice Boltzmann method with D3Q19 model[73,74] was used to solve the convection-diffusion equation, and the simulation was implemented by GPU-based in-house code, which has been comprehensively verified and used in our previous work[75–79]. In initial condition, virus was added on the top of aqueous solution or GEL, and $c$ was set as 1, while in other regions $c$ was 0. No-slip boundary condition was set on the gel-matrix, as well as top and bottom of simulation domain, with half bounce-back scheme used, while periodic boundary condition was set on other boundaries.

### The biosafety evaluations of the long-term intranasal administration of aMV@GEL

The mice were intranasally administered with aMV@GEL every other day in a period of two weeks. The blood and the major tissues (including the nasal cavity, brain, lung, heart, liver, spleen, and kidney) were collected at both day 7 and day 14. The blood was used for blood routine (WBC, RBC, and PLT) and biochemical indicators (BUN, ALT, AST, LDH, and ALP) analysis by using automatic haemacytometer analyzer and biochemical analyzer, respectively. The inflammatory cytokines (IFN-γ, TNF, IL-2, and IL-6) of the nasal homogenate were detected by the corresponding ELISA kit. The major tissues were fixed and prepared into paraffin sections (3–4 μm) for H&E staining.

### Preparation, characterization, and SPV aerosols interception of aMV@GEL

0.12 g chitosan was added to 6 ml ultrapure water, followed by the addition of 40 μl glacial acetic acid. After stirring overnight, 3 ml β-sodium glycerophosphate (500 mg/ml) and 1 ml aMV suspension (5 mg/ml) was slowly added to above solution under the ice bath condition. After stirring overnight, the aMV@GEL was prepared. The morphology and structure of aMV@GEL were characterized by SEM (JSM-6700, JEOL) and CLSM (A1/SIM/STORM, Nikon). The surface Zeta potential of aMV@GEL was tested by Solid Surface Zeta Potential Analyzer (Attract2.0). The gelation time of aMV@GEL was determined by Rheometer (MCR302, Anton-Paar).

To detect the interception effect of PBS, aMV, GEL, and aMV@GEL, the inner wall of the paper tube (length: 16 cm, diameter 10 mm) was coated by a layer of those agents, respectively. During this process, the tube was slowly rotated to ensure uniform coverage of the coating (coating area 50.1 cm², gel coating thickness 60-80 μm). Then we immersed the right side of the tube in a water container and then sprayed Cy5 stained SPV aerosols ($1 \times 10^5$ PFU, 10 min) into the left side of the tube by using an aerosol generator (AER-S-NS12, TOW). The fluorescent images of above paper tubes were obtained by an in vivo imaging system (FX Pro, 624 Kodak) with the Carestream MI software (v.5.0.7 version). The Cy5 fluorescence intensity of the water were detected by microplate reader (infinite M200, TECAN).

To capture the SPV behavior after being intercepted by aMV@GEL, the Cy5 stained SPV ($1 \times 10^5$ PFU) was added to 1 ml fluorescein sodium-stained water, then the mixture was nebulized into aMV@GEL. The images of the mixture after contacting aMV@GEL were captured by CLSM (A1/SIM/STORM, Nikon) at 0 min and 10 min. And the image of aMV@GEL that entrapping SPV was captured by CLSM (A1/SIM/STORM, Nikon) after SPV aerosols was nebulized to fluorescent aMV@GEL for 24 h.

To assess the protective effect of aMV@GEL in vitro, a Transwell™ model (Corning, 3384) was used. ACE2-293T cells ($1 \times 10^4$) were cultured in the bottom chamber, and PBS or aMV@GEL was added in the upper chamber. The SPV aerosols ($1 \times 10^5$ PFU, 10 min) were nebulized over the Transwell™ model, and then the cells were placed into a 5% $CO_2$ environment at 37 °C. After 3 days, the infection rate was detected by CLSM (A1/SIM/STORM, Nikon) and flow cytometry (CytoFLEX LX, Beckman Coulter).

To evaluated the anti-viral efficacy of aMV@GEL experiencing dropping or spraying, we added the GEL (drop), GEL (spray), aMV@GEL (drop) and aMV@GEL (spray) in the upper chamber of Transwell™, respectively. Then the infection experiment and the infection rate detection of the ACE2-293T cells in lower chamber was conducted following the aforementioned method.

### The in vivo and ex vivo distribution of aMV@GEL and aMV

First, Cy5 stained aMV and corresponding aMV@GEL were prepared. The C57/BL6 mouse was treated with fluorescent aMV or aMV@GEL by intranasal administration (nasal drip), respectively. In detail, the cold solution (aMV or aMV@GEL) with identical volume was administered slowly into the nasal cavity of the mouse to prevent leakage. The coating thickness of aMV@GEL ranged from 60 μm to 120 μm. Then the fluorescent images of fluorescent aMV distribution were captured in vivo at 0, 1, 2, 3, 4, 6, and 8 h by an in vivo imaging system (FX Pro, 624 Kodak) with the Carestream MI software (v.5.0.7 version). For ex vivo detection, mice were euthanized and dissected at 8 h, the nasal cavity, brain, lung, heart, liver, spleen, and kidney were also imaged by above imaging system. Moreover, the nasal cavity and brain were further fixed with 4% paraformaldehyde and stored in the optimum cutting temperature compound at -20 °C. The frozen samples were further cut into 10 μm tissue sections using a frozen microtome (LM1950, Leica). The sections were stained with DAPI and imaged using CLSM (A1/SIM/STORM, Nikon).

### The protective effect of aMV@GEL against SPV aerosols in mouse

The transgenic mice with high human ACE2 expression were treated with PBS, aMV, GEL, and aMV@GEL by intranasal administration (nasal drip), respectively. After 30 min, above mice were challenged with SPV aerosols ($1 \times 10^7$ PFU, 30 min) or SPV with S protein B.1.1.529 mutation (Omicron) aerosols ($1 \times 10^7$ PFU, 30 min) by using the equipment aerosol inhalation of 12 noses (AER-S-NS12, TOW). After 3 days, the nasal cavity and lung were prepared into 10 μm tissue sections using a frozen microtome (LM1950, Leica). The sections were stained with DAPI and imaged by CLSM (A1/SIM/STORM, Nikon). The GFP mRNA in the nasal cavity and lung of mice was extracted by using RNAeasy™ Animal RNA Isolation Kit with Spin Column and measured by One Step PrimeScript RT-PCR Kit on the real-time PCR detection system (CFX96, Bio-Rad). The GAPDH was used as reference. The correlated primer sequences were as follows: GFP-R: *CATGTACCACGAGTCCAAGTTC-TACG*; GFP-F: *CTCCCAGTTGTCGGTCATCTTCTTC*.

### Preparation and characterization of ST-293T cells

*ST6GAL1*-plasmid was used to prepare ST-293T cells following the similar method as ACE2-293T cells preparation. The SA in ST-293T and

WT-293T cells were stained with Cy3-SNL, the nuclei were stained with DAPI, and then cells were imaged by CLSM (A1/SIM/STORM, Nikon) and detected by flow cytometry (CytoFLEX LX, Beckman Coulter).

## Entrapment of H1N1 by sMV

The sMV was treated with H1N1 virus at 37 °C for 24 h and then was harvested by centrifugation (5000 g, 5 min). The collected sediment was fixed with 2.5% glutaraldehyde, prepared in ultra-thin section, and detected by TEM (Jeol, JEM-1400).

## Plaque assay

The medium of MDCK was discarded and washed 3 times by PBS. Then 0, 1, 10, 50, 100 μg nMV or sMV was added, separately. After that, $1 \times 10^5$ PFU HN1-CA07 or H1N1-PR8 were added to cells, separately, and placed cells into 5% $CO_2$ environment at 37 °C. Two hours later, the cells were washed 3 times with PBS, and 0.3% agarose dissolved in DMEM was added to cover the cells. After 3 days, the cells were stained with 1% crystal violet. The mRNA of H1N1-CA07 or H1N1-PR8 in cells was detected by using above-described method. The correlated primer sequences were as follows:

H1N1-CA07-R: *AGGGCATTYTGGACAAAKCGTCTA*;
H1N1-CA07-F: *GACCRATCCTGTCACCTCTGAC*;
H1N1-PR8-R: *GGTGACAGGATTGGTCTTGTCTTTA*;
H1N1-PR8-F *CTTCTAACCGAGGTCGAAACGTA*.

## The protective effect of sMV@GEL against H1N1 aerosols in mouse

The BALB/c mice were treated with PBS or sMV@GEL by aforementioned intranasal administration (nasal drip), respectively. After 30 min, the above mice were challenged with low dose H1N1-CA07 aerosols ($2.4 \times 10^4$ PFU, 30 min) or high-dose H1N1-CA07 aerosols ($1.2 \times 10^5$ PFU, 30 min) or low-dose H1N1-PR8 aerosols ($2.4 \times 10^4$ PFU, 30 min) by using the equipment of aerosol inhalation of 12 noses (AER-S-NS12, TOW). For low dose challenged mice, the nasal cavity and lung were fixed and prepared into paraffin sections (3–4 μm) for H&E staining or IHC staining (N protein of H1N1) at Day 7. The mRNA of H1N1-CA07 or H1N1-PR8 in the nasal cavity and lung was detected by using above-described method. The inflammatory cytokines (IL-2, IL-6, and TNF) in the lung were detected by the corresponding ELISA kit. For high-dose challenged mice, the weight and survival status were recorded during 14 days.

## The protection effect of sMV@GEL in H1N1 transmission model

We designed a cage divider, which allowed airflow but no direct contact or fomite (including diet and bedding) transmission. Donor mice were infected with H1N1-CA07 virus on day 0 by intranasal administration. On day 3, two donors were introduced into one side of the cage divider. Meanwhile, two sMV@GEL-treated intermediate recipients (IRs) and two PBS-treated IRs were placed on the other side of the divider (downflow from the donor mice). On day 6, the half of IRs were sacrificed for viral-loading assay, and the rest of IRs were introduced into a new cage, wherein the cage divider separated two sMV@GEL IRs or two PBS IRs from two terminal recipients (TRs, naïve mice, downflow from the IR mice). On day 9, the TRs were sacrificed for infection rate analysis. The mRNA of H1N1-CA07 in the nasal cavity and lung was detected by using above-described method. In q-PCR analysis, the mouse was considered uninfected when relative Cq value was higher than 35.

## Preparation of digital human nasal model

Based on the nasal CT data from a healthy adult volunteer, the digital human nasal model was reconstructed by 3D-slicer software. The child nasal model was obtained by scaling down the adult nasal cavity model, using a scaling factor of 0.65[80,81].

## Modeling for CFD-DPS

In simulation of CFD-DPS, the gas phase was governed by Navier-Stokes equation for mass and momentum conservation:

$$\nabla \cdot \mathbf{u}_f = 0 \tag{3}$$

$$\frac{\partial \mathbf{u}_f}{\partial t} + (\mathbf{u}_f \cdot \nabla)\mathbf{u}_f = -\frac{1}{\rho_f}\nabla p + \nu \nabla^2 \mathbf{u}_f \tag{4}$$

where $\mathbf{u}_f$ was fluid velocity, $t$ was time, $p$ was pressure, $\rho_f$ was fluid density, and $\nu$ was kinematic viscosity. Taking the maximum inlet velocity ($\mathbf{u}_{inlet,\ max} = 0.64$ m/s, $t = 0.75$ s) into account, the Reynolds number was 734 at inlet and 1261 at outlet. Considering the relatively low Reynolds number and the focus of nasal cavity, we proposed that the situation could be approximately regarded as in laminar region. The individual viral aerosol was regarded as spherical particle and its position and velocity were updated by the Newtown's second law:

$$m_p \frac{d^2\mathbf{x}_p}{dt^2} = m_p \frac{d\mathbf{u}_p}{dt} = \mathbf{F}_d + \mathbf{F}_e + m_p\mathbf{g}\frac{\rho_p - \rho_f}{\rho_p} \tag{5}$$

where $m_p$ was mass of particle, $\mathbf{g}$ was gravitational acceleration, $\rho_p$ was the particle density, $\rho_f$ was fluid density, $\mathbf{x}_p$ was particle position, and $\mathbf{u}_p$ represented particle velocity. $\mathbf{F}_d$ was drag force exerted by gas and calculated by Wen-Yu drag model. $\mathbf{F}_e$ was electric force produced by MV@GEL on the surface of nasal wall. Considering small size and concentration of aerosol particles, the collision force between aerosol particles and reaction force of particle on fluid were all neglected. The human nasal model had been discretized to computational mesh as well as nasal wall surface before the simulation, and the $\mathbf{F}_e$ was calculated by summing all element forces between aerosol and nearby wall elements, with an assumption that MV@GEL was evenly embedded on the surface of nasal wall. The element force was determined by electric-field intensity test and was fitted to algebraic form about distance: $\mathbf{F}_{element} = (-8 \times 10^{-11}d + 5.6 \times 10^{-12})\ \mathbf{n}$, where $d$ was distance and $\mathbf{n}$ was normalized vector from particle to wall element.

## Numerical implementation for simulation

The governing equation of gas phase and aerosol particles solution were carried out with open-source finite-volume based program, OpenFOAM-6. The nasal model was discretized as computational mesh by SnappyHexMesh (mesh generation module in OpenFOAM-6), and mesh topology was determined by refining until that grid independence was achieved. The final mesh was characterized by 670,506 elements and 720,496 nodes. All variables of fluids and particles were stored in the centroid of element cells. The PIMPLE algorithm was used to solve the partial differential equation, with second-order scheme for spatial discretization and first-order Euler scheme in time integration. The electric force was redeveloped based on C++ (programing language) and implanted into the particle governing equation. The same boundary conditions were used in the inhalation and exhalation processes. The transient velocity boundary was imposed at the nostrils to simulate a complete respiratory cycle, with $\mathbf{u}_{inlet} = 0.64\sin(2\pi/3 \times t)$. *pressureInletOutletVelocity*, which was a kind of zero gradient boundary in OpenFOAM, was used at the throat for gas flow into and out of nasal cavity freely. The aerosol particle size was 5 μm, at a emit rate of 3000 per second in each nostril. Aerosol particles were injected only in first 1.5 s, and particles out of nasal from throat were stored, which would flow back to nasal with gas entrainment at expiratory process (1.5–3 s). When aerosol particle touched the nasal wall, it was considered that aerosol was captured by MV@GEL and its renewal was stopped. All computation was performed on a 2.2 GHz processer, with 4 threads for parallel computing. It normally took 96 h to reach the end.

## Construction of HRT model

For the preparation of lung organoids, we firstly obtained para-cancerous tissues from surgically resected sample of a lung cancer patient. The tissues were washed with PBS which containing antibiotics and chopped into approximately 5 mm pieces. And tissues were further washed with 10 ml advanced DMEM/F12 which containing 1× Glutamax, 10 mM HEPES, and antibiotics, then digested in 10 ml advanced DMEM/F12 with 2% fetal calf serum and 2 mg/ml collagenase, shaking at 37 °C for 1–2 h. Dissociated cells were collected at 400 g, suspended in complete human organoid medium with 10% growth factor reduced Matrigel[82–84]. The lung organoids were harvested until those cells formed spheroids. For identification, the lung organoids were stained with FOXJ1, SCGB1A1, or P63 antibodies and corresponding secondary Alexa Fluor® 647 fluorescent antibodies, respectively. The SA in the lung organoid was stained with Cy3-SNL. The cytoskeleton was stained by FITC-phalloidine, cell nuclei were stained with DAPI.

The 3D printing technology was used to construct a real apparatus of the human nasal module following the digital human nasal model. To mimic the HRT, the lung organoids were placed into a sealed container. One side of the sealed container connected to the nasal apparatus, and the other side connected to the pump.

## The protective effect of sMV@GEL against H1N1 aerosols in HRT model

The cold solution (PBS or sMV@GEL) with identical volume was spray in to nasal cavity with a fixed position (0.5 cm inside the nasal cavity) and fixed direction (45°), which could ensure the consistence of the coating layer (thickness 60-150 μm) for each test. Then the HRT model was placed into 37 °C, the H1N1-CA07 ($2.4 \times 10^4$ PFU, 30 min) or H1N1-PR8 ($2.4 \times 10^4$ PFU, 30 min) aerosols were nebulized around the nares of HRT model. Meanwhile, the pump was started to provide the respiratory flow (10 L/min). After 30 min, the lung organoids were placed into 5% $CO_2$ environment at 37 °C. After further 24 h and 48 h culture, the mRNA of H1N1-CA07 or H1N1-PR8 in the lung organoids was detected following above-described method. To analyze the cell apoptosis and cell viability of lung organoids, the Annexin V-mCherry Apoptosis Detection Kit and the Calcein/PI Cell Viability/Cytotoxicity Assay Kit were used following the instructions, respectively.

To evaluate the performance of using the intranasal mask plus the regular mask, four groups including PBS, mask, intranasal mask (sMV@GEL), and combined masks (mask+sMV@GEL) were included. Simulating real-life scenarios where masks may not be worn correctly, we introduced a "loose wearing" condition by creating a gap (1-3 mm) between the mask and the nasal module. For the combination group of "mask+sMV@GEL", the sMV@GEL was applied in nasal apparatus, and then the surgical mask was worn on the nasal apparatus in "loose wearing" way. The infection experiment was conducted following the previous mentioned HRT infection method ($2.4 \times 10^5$ PFU, 30 min).

The child-derived HRT model was integrated by following the construction method of HRT model. The breath air flow rate was adjusted to 8 L/min from the 10 L/min of HRT model. The protection effect of sMV@GEL in child-derived HRT model was evaluated following the method in HRT model.

## Statistics and reproducibility

Statistical analyses were performed using GraphPad Prism 8.4.3. All data was repeated in biology and all graphs present mean ± S.E.M. unless otherwise specified. A two-tailed unpaired Student's $t$-test was used to compare two groups, one-way ANOVA was used to compare multiple groups, and log-rank test was used to analyze survival time. The threshold for statistical significance was set as $P < 0.05$.

Experiments were replicated multiple times and obtained the similar results. For in vitro and ex vivo studies, experiments were replicated three times. For viral infection inhibition studies, experiments were replicated twice.

## Reporting summary

Further information on research design is available in the Nature Portfolio Reporting Summary linked to this article.

## Data availability

All data supporting the findings of this study are available within the article and its supplementary files. Any additional requests for information can be directed to, and will be fulfilled by, the corresponding authors. Source data are provided with this paper and are also available in Figshare (https://doi.org/10.6084/m9.figshare.24590910.v2). Source data are provided with this paper.

## Code availability

Code for simulation section in this study are available in GitHub (https://github.com/ScottFu123/simulation-files/tree/master) and (https://doi.org/10.5281/zenodo.10212442)[85].

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

## Acknowledgements

We are grateful to the volunteers for participation in the study. We are grateful to Professor Youchun Wang and Professor Weijin Huang for kindly providing serval variants of SARS-CoV-2 pseudovirus. We are grateful to Professor Meng Qin, Professor Jing Sun and Professor Tao Jiang for kindly providing H1N1-PR8 and H1N1-CA07 strains of Influenza A virus. We are grateful to Professor Li Zhu for kindly providing biosafety level 2 laboratory. This work was supported by the National Natural Science Foundation of China (T2225021 to W.W., 21821005 to G.M., 32030062 to G.M., U2001224 to W.W., 22108284 to S.W., and 82102205 to H.D.), CAS Project for Young Scientists in Basic Research (YSBR-083 to W.W.), National Key Research and Development Program of China (2021YFC2302600 to G.M.), IPE Project for Frontier Basic Research (QYJC-2022-006 to S.W.), Beijing Natural Science Foundation (JQ21027 to W.W.), Beijing Nova Program of Science and Technology (20230484335 to S.W.) and the Strategic Priority Research Program of the Chinese Academy of Sciences (XDB29040303 to G.M.).

## Author contributions

W.W., G.M., and L.W. conceived and designed the study. X.H. and S.W. performed most of the experiments and analyzed data. S.F. performed the computer simulation. M.Q., C.L., Z.D., D.W., L.Z., T.J., and J.S. assisted with the H1N1 infection experiments. J.W. assisted with the preparation of hydrogel. C.L., Y.C., and X.P. assisted with the lung organoids culture. M.Q., Y.W., Y.S.W., and P.Y. assisted with the cell experiments. L.C. and H.D. assisted with computer simulation. Y.C.W. and W.H. assisted with SPV infection experiments. M.Q, Z.D., and C.L. provided suggestions on data presentation. X.H., S.W., and S.F. wrote the original draft manuscript, W.W. and G.M. revised the manuscript. All authors reviewed the manuscript and approved the final version.

## Competing interests

The authors declare no competing interests.

## Additional information

[1]State Key Laboratory of Biochemical Engineering, Institute of Process Engineering, Chinese Academy of Sciences, 100190 Beijing, China. [2]School of Chemical Engineering, University of Chinese Academy of Sciences, 100049 Beijing, China. [3]State Key Laboratory of Multiphase Complex Systems, Institute of Process Engineering, Chinese Academy of Sciences, 100190 Beijing, China. [4]Beijing Advanced Innovation Center for Soft Matter Science and Engineering, College of Life Science and Technology, Beijing University of Chemical Technology, 100029 Beijing, China. [5]State Key Laboratory of Pathogen and Biosecurity, Beijing Institute of Biotechnology, 100071 Beijing, China. [6]State Key Laboratory of Pathogen and Biosecurity, Beijing Institute of Microbiology and Epidemiology, 100071 Beijing, China. [7]Institute of Chinese Materia Medica, China Academy of Chinese Medical Sciences, 100029 Beijing, China. [8]Shenzhen Key Laboratory of Nanozymes and Translational Cancer Research, Department of Otolaryngology, Shenzhen Institute of Translational Medicine, The First Affiliated Hospital of Shenzhen University, Shenzhen Second People's Hospital, 518035 Shenzhen, China. [9]Key Laboratory of Biomechanics and Mechanobiology, Ministry of Education Beijing Advanced Innovation Center for Biomedical Engineering School of Biological Science and Medical Engineering, Beihang University, 100083 Beijing, China. [10]Department of Pharmacy, Peking University First Hospital, 100034 Beijing, China. [11]Institute of Clinical Pharmacology, Peking University, 100191 Beijing, China. [12]Division of HIV/AIDS and Sex-Transmitted Virus Vaccines, Institute for Biological Product Control, National Institutes for Food and Drug Control (NIFDC) and WHO Collaborating Center for Standardization and Evaluation of Biologicals, 102629 Beijing, China. [13]Department of Breast Surgery, Affiliated Quanzhou First Hospital of Fujian Medical University, 362000 Quanzhou, China. [14]These authors contributed equally: Xiaoming Hu, Shuang Wang, Shaotong Fu. ✉e-mail: lmwang@ipe.ac.cn; ghma@ipe.ac.cn; weiwei@ipe.ac.cn

