## [Peer review file · Nature Communications]

REVIEWER COMMENTS

Reviewer #1 (Remarks to the Author):

Hu et al.

Intranasal mask for protecting the respiratory tract against viral aerosols

This manuscript by Hu et al. presents a very interesting series of studies designed to create an intranasal mixture that combines several features in order to capture and entrap incoming respiratory viruses, and inactivate them.

A series of models, some innovative, are set up in the course of developing this strategy. The well-written and detailed paper lays out a proof of concept for several components of this nasal trapping approach, using the examples of a SARS pseudovirus and influenza, and will spur further study of this strategy. The setup to assess release of virus from aerosols, and other elements of the assays, are ingenious.

There are several points that require attention.

1. A central concept in this paper is the use of a receptor-bearing vesicle as a "decoy" of sorts, and this is developed throughout the paper. Yet the topic is introduced (lines 109-111) as an original idea of the authors ("inspired by receptor-mediated viral infections in host cells, ref 23") when the concept of using virus receptor-bearing decoys of various sorts is a topic that has been developed by others. While there is no need to review that field, this strategy should be introduced with at least the notion that using viral receptors on particles as decoys is not a new idea (it is not referenced at all). For example (among others) Porotto PLoS One 2011, where synthetic particles bearing viral receptors inactivate viral particles.
2. The authors aim for, and achieve, what appears to be entrapment of viral particles within the receptor-bearing vesicles. However, is entrapment necessary, or would binding to the decoy be sufficient? Also, have the authors considered viral inactivation by receptor engagement on these vesicles (ie, the S protein is inactivated upon engagement with its target)?
3. Figure legends are far too long and need to be made concise (minor point).
4. While suggestions about whether or not the mixture also blocks transmission of virus between hosts (not within one host) can be drawn based on the data, the authors could be more explicit about this. It would be good to know whether the viruses after contact with the nasal cavity of one host cannot be transmitted to a new host.
5. The transgenic mouse model (introduced line 241) for SARS-CoV-2 has been determined over the last years to be suboptimal in many ways. It serves its purpose in this paper but the authors should provide appropriate caveats and mention what model might be preferable in the future.
6. Line 265. The authors overexpress $\alpha 2,6$ -type receptors for use with influenza virus, but earlier had correctly mentioned the importance of both $\alpha 2,3$ - and $\alpha 2,6$ -type influenza receptors. Influenza viruses target airway epithelial cells via both $\alpha 2,3$ - and $\alpha 2,6$ -type receptors, but the distribution of these receptors in species is uncertain and this influences infection, tropism etc. The authors should justify the choice of using only $\alpha 2,6$ -type and while this is fine as a POC, other linkages may be necessary in the future.
7. Line 298. Please comment on why the high-dose mice lost weight.
8. Line 367 and ongoing paragraph. The organoid section lacks necessary context; the infected cells presumably are (should be) the cells lining the center of the organoid, not the outer surface. So, what

do the authors take by infectivity at the outside of the organoid? What does this mean? There needs to be a little more understanding here of how the organoid is infected in this scenario and whether or how this represents the lung.

9. Along these lines, there is inadequate description either here or in the methods about the generation and validation of these human lung organoids. In order to view them as representing human lung, the method of differentiation and validation needs to be clear. The section starting on line 677 (construction of human respiratory tract model) is inadequate – and has no references. This is a significant issue.

10. Line 381 and elsewhere. The mention of the strategy being “independent of virus mutation” is too unspecific and is premature. The authors may mean “serotype” here, not mutation. Or perhaps CA07 vs. PR8 is considered a different of “mutation” but this would not be not a standard way to express this. The statement that there is indifference to strain or sequence (if this is as meant) is premature and overstated and should be re-worded and tempered.

Overall this is a truly interesting and stimulating paper that will benefit from attention to these points.

Reviewer #2 (Remarks to the Author):

Hu et al reported an interesting intranasal mask to entrap virus for the protecting the respiratory tract against viral aerosols. While engineering cell-derived vesicles to express viral binding receptors for entrapping viruses is not a new idea, using hydrogels as an ‘intranasal mask’ may provide some additional respiratory protection.

Main concerns:

1. MV may not be a viable treatment option in clinics. As each patient’s human leukocyte antigen (HLA) is different (with the exception of identical twins), to avoid unwanted immune responses, the authors have to make a personalized MV for each patient, which would be both expensive and impractical. HLA-matched cells must be engineered for individual patients, which can result in exorbitant costs and is thus unattainable by many patients.
2. MVs are composed of random unknown components from 293T cells which can result in significant variability and safety concerns and can induce unwanted immune responses.

I also have the following concerns:

1. The thermosensitive hydrogel is one of the main components of MV@GEL. However, the actual thermosensitivity regarding the critical temperature and time duration leads to transition from liquid to gel was not characterized. This needs to be carefully studied considering an actual case when a patient is holding a spray of the MV@GEL using warm hands. What if it turns into gel before even the spray? Can it transit from gel to liquid freely and does it change the performance of the MVs loaded? The thermosensitive gel also adds instability and risks for delivery during severe weather conditions like high environmental temperature.
2. The authors mentioned the hydrogel with positive charges could trap some intercepts the negatively charged viral aerosols. Such an interaction seems not specific to viruses. Could it also bind some negatively charged cell fragments or bacteria? If so, would it block some interaction of MVs and actual viruses?
3. The authors showed some viruses were entrapped by MVs, while there are still some viruses on the surface of the MVs (Step3, Fig1A). The viruses trapped on the cell surface may have the possibility to infect cells, particularly for authentic viruses.
4. On page 5, the authors first showed that SPV signal on the aMV surface (Fig. S3A) and then showed that overtime, the SPV signal would be seen in the interior of aMV (Fig. 2G-H). The authors should provide the control of using nMV in Fig. 2G-H to exclude some nonspecific binding in this prolonged

time point.

5. On page 7, the authors mentioned the size distribution of 1 to 10 μm (Fig. S10). However, the a-axis is nm instead of μm . This needs to be corrected and proper citations need to be added.

6. On page 9, the quantification of relative expression of GFP does not seem to be consistent with the fluorescent images in Fig. 4G. The GFP intensity of Gel group seems much lower than that of PBS in the images, while the quantification showed similar intensities (n.s.). The authors should double check the quantification results and also clearly mention the method to do the quantification. Also, only 3 (N=3) seemed insufficient for such image analysis, considering the region-of-interest and varieties of such imaging experiments.

7. The advantage of protecting mice weight loss in influenza infection is marginal. In Fig. 5M, the weight losses in day 10 day are comparable between sMV@Gel and PBS groups.

8. Although the authors used transgenic ACE2 mice to test the protection effect of aMV@GEL, however, they only used pseudovirus and cannot represent the authentic virus and claim the protection effects. What if they challenged the aMV@GEL group using real SARS-CoV2 viruses? Also, would the protection be the same or different across different variants?

9. The reviewer appreciates the attempts of using computational fluid dynamics-discrete particle method (CFD-DPM) and integrated human respiratory tract (HRT) model to investigate the protection of respiratory system. Could the authors describe how many patients they used for reconstructing the CT data considering the possible differences between individuals?

Minor point:

1. The author should properly cite existing literature that already showed the cell-derived vesicles can trap viruses. There are several have been published. On page 4, the authors mentioned 'we envisioned that the cell-derived vesicles that contained overexpressed receptor on the surface could potentially entrap the virus' without proper citations.

Overall, this is manuscript has done significant work to characterize the MV@GEL and its protection. However, the novelty and biosafety are the primary concerns in the current format.

Reviewer #3 (Remarks to the Author):

Xiaoming Hu et al describes a novel method to prevent viral infections by retaining/entrapping viral aerosols thru the use of an intranasal mask made of a gel containing cell-derived vesicles that display specific viral receptors. The overall design of the experiments and data display are well done. The manuscript includes several innovative approaches to evaluate the safety, efficacy and efficiency of the proposed mask. Nevertheless, some important aspects of this device need to be evaluate to convey its real applicability to prevent infections in humans.

Major points:

1. During the in vivo evaluation of the aMV@GEL safety, there were not significant differences on the overall cellular structure and respiration of the mice. However, the exposure occurred only once for a short time. Since this product is meant to be used several times during an extended period, it will be important to demonstrate that several applications of the aMV@GEL during different days/weeks would not lead to adverse effects like, for instance, damage of the epithelium, recruitment of inflammatory mediators, etc. In this line, the accumulation of the particles in vital organs was only assessed 8h after application of the aMV@GEL, could the authors present later time points like 24/48h post initial exposure?

2. Regarding the HRT model, while it is an innovative and interesting set up, several pitfalls need to the addressed to carry the author's message. The cellular characterization of the organoids needs to be re-done. The stainings presented are not convincing and appear to show the same cells (no cilia in the ciliated cell marker), instead of using 2D pictures would be better to present whole mount 3D pictures as done in the NP panel. Regarding the organoid infection, I would recommend to do more readouts than just the staining NP or viral RNA expression. The authors could include a cell viability marker, apoptosis measurements, analysis of other relevant inflammatory mediators, plaque assay

analysis.

3. In addition, the organoid model described reflects the airway epithelium of the proximal airways. Therefore, it will make more sense to have the cultures closer to the nasal apparatus. Another option will be to replace the current model for one that will better reflect the distal epithelium of the lung, this will be particular more relevant since influenza infection of the distal epithelium is a major

4. Regarding the product applicability, one major point raised by the authors in the conclusions is that health workers could use the intranasal mask as an additional method for protection against viral infections. Could the authors evaluate if the use of the intranasal mask plus the regular mask compared to only use one or the other really adds up to a higher protection against viral infections?

Minor points:

5. The use of color in the background of some of the graphs is a bit distracting and do not follow a pattern. Maybe be better to remove.

Reviewer #4 (Remarks to the Author):

For this paper I was asked to review the following: "we would specifically require input regarding the computational aspects (Figure 6) from you, as we have the other aspects featured in this study covered by other referees".

After reading the manuscript I find the computational methodology and the numerical simulations appropriate and useful. However, I would like to ask the authors extend a little bit the discussion and clarify the following points/comments.

1. According to Koullapis et al. (2018) "the flow in the upper airways is mostly turbulent and/or transitional in nature, even at low inhalation rates". However, it seems that the performed simulations have been for laminar conditions. Therefore, the authors should convincingly argue why they decided to perform laminar flow simulations. Which was the flow Reynolds number at inlet and outlet of the computational domain?

2. Repeatability of the numerical results. Authors include only 18000 particles for the total simulation time, which are very few. I would like to know if the simulations have been repeated several times under identical conditions (except for a random location of the tracked particles) and if the same results were obtained.

3. Authors should explain why some phenomena as Brownian motion, Saffman lift forces or buoyancy effects have not been considered in the simulations.

4. Which were the exact boundary conditions in the exhalation phase?

5. "one order Euler scheme" should be "first order Euler scheme".

6. Which was the density of aerosol particles?

Authors should correctly address the raised questions prior to acceptance.

References

P. Koullapis, S.C. Kassinos, J. Muela, C. Perez-Segarra, J. Rigola, O. Lehmkuhl, Y. Cui, M. Sommerfeld, J. Elcner, M. Jicha, I. Saveljic, N. Filipovic, F. Lizal, L. Nicolaou Regional aerosol deposition in the human airways: The SimInhale benchmark case and a critical assessment of in silico methods. *Eur J Pharm Sci.* 2018, 113:77-94. doi: 10.1016/j.ejps.2017.09.003

Reviewer #5 (Remarks to the Author):

This manuscript submitted to Nature Communications lies in the context of the prevention of aerosol-transmitted viral diseases. The authors developed an original formulation intended for intranasal administration and acting as an "intranasal mask". It consists of a chitosan-based thermosensitive gel

containing cell-derived microvesicles with specific receptors to capture and immobilize the virus within the gel in the nasal cavity, thus preventing it from reaching the lungs. In this study, the authors combined a full set of in vitro experiments, in vivo evaluation on mouse models, numerical simulations, and the development of an apparatus mimicking the human respiratory tract to establish the proof of concept on different variants of SARS-CoV-2 and influenza A. The experiments were generally well-designed and conducted. I share the main conclusions of the authors. In my opinion, these complementary approaches have led to very encouraging and promising results that deserve publication and dissemination to a large audience within the scientific community, especially in the current pandemic context. Indeed, the proposed strategy could significantly improve the prevention of these infectious diseases and have a high impact on public health. However, I think the authors should add some essential missing experimental details and clarify some aspects before publication in Nature Communications.

1) Can the authors obtain batches of microvesicles overexpressing ACE2 or α -2,6 sialic acid receptors with reproducible characteristics? Are these microvesicles stable in the hydrogel? For how long? Are they dispersed in a reproducible way in the hydrogel? Do these vesicles retain their integrity and efficacy when they experience high shear rates in the spraying device or when the formulation dries in air after deposition on the surface or after intranasal administration?

2) The hydrogel spraying ability was studied, and a spraying method was used in some experiments but not systematically. The authors should state more clearly if this method of administration is foreseen or not for future development. In this respect, the characteristics and quality of the coating of the nasal surface or material mimicking the nasal surface are critical for the efficacy of the hydrogel to entrap the virus. In my opinion, a weakness of this study is the incomplete description of this coating in the different experiments. The authors should clarify or discuss the following points:

- What is the size distribution of the droplets obtained when spraying the MV@GEL or other control systems?
- In some experiments, the method of coating needs to be detailed (for instance, the coating of the tube with PBS, aMV, GEL, or aMV@GEL for in vitro study of SPV aerosol interception).
- What area is covered by the hydrogel in the different types of experiments?
- What is the layer thickness?
- Could the authors comment on the wettability and adhesion properties, as these properties are also crucial for good surface coverage?
- What is the surface Zeta potential or surface charge of the coating? They have an impact on the electrostatic interactions between the virus droplets and the surface.
- How is performed the intranasal administration of the formulations in mice?

3) After virus interception by the gel, essential points are the ratio of virus that interacts with the receptors and is entrapped in the microvesicles within the gel and the ratio of virus that remains "free" in the gel. Depending on the gel thickness (not provided) and on the mobility of the virus within the gel, this free virus may be able to reach and infect the underlying epithelial cells of the nasal cavity, contrary to the assertion of the authors (see line 216). Can the authors comment on these aspects and the mobility and transport mechanisms of the virus within the gel? I suggest they could better comment and analyze Fig S11 to see if a diffusion mechanism applies. Did they perform single virus tracking experiments within the gel?

4) The authors claim they have a prolonged retention time of the gel (e.g., line 84 "...which was favorable for prolonging the retention time on the intranasal wall of the nose", Lines 388-389 "the MV with the viral receptor that was embedded into the hydrogel could be retained in the nasal cavity in long-term". The present study did not assess this aspect. For such an assertion, the authors should quantify this residence time or refer to data from the literature with a similar system.

5) Some groups have already published data on nasal sprays with mucoadhesive polymers (without microvesicles) to create a physical barrier in the nose to prevent infection with SARS-CoV-2. Some

clinical trials were also performed with such formulations. The authors should discuss their study and results in this context (see, for instance, the review by Nocini et al., *Biomedicines* 2022, 10, 2966, the work of Robinson et al., *Frontiers in Medical Technology*, 2021, 3, 687681 and the communication of Bentley and Stanton, *Viruses*, 2021, 13, 2345).

More specific aspects that require special attention from the authors are listed below:

- Line 70: "Once humans inhale those floating viral aerosols, infection occurs". Infection is not systematic. The infection can or may occur.
- In the size distributions, also provide the polydispersity index. What dispersion medium and dilution are used for the Dynamic Light Scattering experiments?
- Please provide the size distribution of nanosized vesicles (aNV).
- Fig 3I. What is the time scale for observing these different steps (contact of SPV with the gel, fusion of the viral aerosol with the gel, entrapment of SPV by aMV)? At what depth in the gel was image iv obtained?
- Replace "gelatinized" and "gelatinization" with "gelified" and "gelation".
- Pore size" would be more appropriate than "aperture size".
- Several experiments were conducted at 37°C (for instance, rheological experiments). The nasal temperature is more likely to be around 34°C. What about the gelation kinetics at this temperature?
- The rheological experiments must be better described. What are the geometry characteristics (cone-plate? plate-plate? material? dimension? truncature, or gap?). Were these experiments conducted in the linear regime of deformation? What was the applied strain or stress? The angular frequency cannot be 0.01 rad/s if the frequency is 0.1Hz. Were the rheological experiments reproducible? At least three repetitions on different aliquots are required for good precision.
- Fig S7A: For CS0.12, the dashed vertical line does not cross the experimental curves at $G'=G''$.
- Quartz crystal microbalance: the author should provide the protocol or refer to a previous publication describing it. How is the gel layer immobilized on the surface of the chip?
- FigS7D caption: "The downward frequency reflected an increased quality on the chip, thus reflecting the interaction force between hydrogel and SPV". The word "quality" is not appropriate. The frequency is linked to the mass of SPV that is adsorbed on the chip.
- Fig7E: What is the authors' interpretation of the images CLSM images?
- Fig7G: To obtain the radar images, how are the different relative parameters calculated? What reference is used for this normalization?
- Cytotoxicity of the formulation was only studied over a short time (1h for cell viability in Fig. S7 and observation of mice nasal mucosa in Figure S13). In the case of human application, a longer contact time will be necessary, with repeated administration. The authors should mention this limitation.
- In what material was the nasal apparatus printed? This has an impact on how the hydrogel aerosol and virus aerosol interact with the printed nasal surface.

RESPONSES TO REVIEWERS

(Note: All the responses here are in blue. All the changes have been highlighted in yellow in the main text of revised manuscript.)

For reviewers 1:

This manuscript by Hu et al. presents a very interesting series of studies designed to create an intranasal mixture that combines several features in order to capture and entrap incoming respiratory viruses, and inactivate them. A series of models, some innovative, are set up in the course of developing this strategy. The well-written and detailed paper lays out a proof of concept for several components of this nasal trapping approach, using the examples of a SARS pseudovirus and influenza, and will spur further study of this strategy. The setup to assess release of virus from aerosols, and other elements of the assays, are ingenious. There are several points that require attention.

1. A central concept in this paper is the use of a receptor-bearing vesicle as a “decoy” of sorts, and this is developed throughout the paper. Yet the topic is introduced (lines 109-111) as an original idea of the authors (“inspired by receptor-mediated viral infections in host cells, ref 23”) when the concept of using virus receptor-bearing decoys of various sorts is a topic that has been developed by others. While there is no need to review that field, this strategy should be introduced with at least the notion that using viral receptors on particles as decoys is not a new idea (it is not referenced at all). For example (among others) Porotto PLoS One 2011, where synthetic particles bearing viral receptors inactivate viral particles.

Response: As the reviewer mentioned, using receptor-bearing decoys of various sorts has been developed by others. This point had been realized by us and had been mentioned in the discussion section of our originally submitted manuscript (Page 14, line 401-403): “Inspired by viral receptor-mediated host cell infection, nanovesicles and cell membrane-coated particles displaying viral receptors have recently emerged as new tools to bind viruses and further suppress viral infection⁴⁹⁻⁵².”

Following this sentence, we had also provided the discussion in the aspect of the superiority of our macrovesicles (Page 14, line 403-408): “It should be emphasized that the ultrasmall size of the nanovesicles (e.g., ~100 nm for exosome) or the solid core of the membrane-coated particles (e.g., polymeric particle coated with ACE2-membrane) were unfavorable for harnessing the potential of entrapping virus inside, which failed to completely isolate the virus from host cells. In the present study, we rationally prepared MV displaying viral receptors and demonstrated their virus entrapment ability.”

To further eliminate the misunderstanding that the “decoy” idea is an original idea of us, we have now revised corresponding description of MV and added the reviewer mentioned reference as ref 28 in the revised version as follow.

Main text:

Page 4, Line 109-112: Inspired by receptor-mediated viral infections in host cells²³, studies that using viral receptor as virus decoy²⁸⁻³¹, and the size of most virus ranging

from 60 nm to 140 nm³², we envisioned that the cell-derived vesicles with abundant viral receptors and a large cavity could facilitate the viral entrapment³³.

Reference:

23. Hoffmann, M., et al. SARS-CoV-2 Cell Entry Depends on ACE2 and TMPRSS2 and Is Blocked by a Clinically Proven Protease Inhibitor. *Cell* **181**, 271-280 e278 (2020).
28. Porotto, M., Yi, F., Moscona, A. & LaVan, D.A. Synthetic protocells interact with viral nanomachinery and inactivate pathogenic human virus. *PLoS One* **6**, e16874 (2011).
29. Li, Z., et al. Cell-mimicking nanodecoys neutralize SARS-CoV-2 and mitigate lung injury in a non-human primate model of COVID-19. *Nat Nanotechnol* **16**, 942-951 (2021).
30. Coccozza, F., et al. Extracellular vesicles containing ACE2 efficiently prevent infection by SARS-CoV-2 Spike protein-containing virus. *J Extracell Vesicles* **10**, e12050 (2020).
31. Wang, C., et al. Membrane Nanoparticles Derived from ACE2-Rich Cells Block SARS-CoV-2 Infection. *ACS Nano* **15**, 6340-6351 (2021).
32. Yao, H., et al. Molecular Architecture of the SARS-CoV-2 Virus. *Cell* **183**, 730-738 e713 (2020).
33. Zhang, S., Gao, H. & Bao, G. Physical Principles of Nanoparticle Cellular Endocytosis. *ACS Nano* **9**, 8655-8671 (2015).

2. The authors aim for, and achieve, what appears to be entrapment of viral particles within the receptor-bearing vesicles. However, is entrapment necessary, or would binding to the decoy be sufficient? Also, have the authors considered viral inactivation by receptor engagement on these vesicles (ie, the S protein is inactivated upon engagement with its target)?

Response: We thank the reviewer for providing us this opportunity to clarify the necessity of entrapment. In Fig. 2I of our originally submitted manuscript (shown as Fig.R1 below), we had compared the anti-viral effect of aNV (nanosized vesicles displaying ACE2 for binding to virus) and aMV (microsized vesicles displaying ACE2 for binding and further entrapping virus). The results demonstrated that the antiviral effect of aNV was significantly lower than that of aMV, indicating a much superior shield and inactivation of virus by aMV's entrapment function. In other word, the virus has multiple active S proteins on the surface, and the decoy receptor (ACE2) on aNV only inactivated a portion of them, with other active S proteins on the virus surface remaining infectivity to host cells.

We have now added more description about the entrapment of aMV and emphasized the necessary of entrapment in our revised manuscript.

Main text:

Page 6, line 159-162: This should be attributed to the rational design of aMV by integrating the indispensable features of the cellular vesicle, micro-size, and ACE2 display for efficient entrapment of SPV, which together result in complete SPV shielding and avoid the infection risk induced by residual inactivated S proteins on the surface of SPV (Fig. S7).

Fig. R1 (as Fig. 2I in revision) Infection rate of SPV after incubation with indicated agents. CLSM images of ACE2-293T cells that were treated with the indicated agents and then challenged with SPV (wild-type), accompanied with the corresponding quantification of relative infection rate (right, n=3 biologically independent experiments). Prior to being challenged with SPV, the ACE2-293T cells were treated with PBS, purified recombinant ACE2 (ACE2 protein), nanoscale vesicles prepared from ACE2-293T cells (aNV), aMV, and aMV that was blocked with ACE2 antibody (aMV/anti-ACE2). GFP was expressed in infected cells (green), and the cell nuclei were stained with DAPI (blue). Data represent the means \pm SEM. Statistical significance was tested with one-way ANOVA with multiple comparison tests. ****P < 0.0001.

3. Figure legends are far too long and need to be made concise (minor point).

Response: According to the reviewer comment, we have now abbreviated the figure legends where possible. The deleted information could be found in main text or method section, where sufficient information had been provided for readers to understand the figures. Just list as few examples as below.

Mian text:

Caption of Fig. 2A (abbreviate words from 107 to 21): Schematic for MV with high angiotensin-converting enzyme II (ACE2) expression (aMV) preparation and its mechanism of entrapping the SARS-CoV-2 pseudovirus (SPV). ~~When SPV infected the cell, the S protein of SPV needed to bind to the ACE2 receptor on the host cell's surface to implement the infection. To interrupt this process, we constructed ACE2-overexpressed 293T cells (ACE2-293T) by transforming the ACE2 plasmid into 293T cells, then MV with high ACE2 expression (aMV) could be obtained by treating ACE2-293T cells with 10 μ g/ml cytochalasin B for 30 min. Using the receptor as the bait, these aMV were capable of entrapping the virus, thus blocking the infection of cells.~~

Caption of Fig. 4A (abbreviate words from 73 to 14): Schematic diagram for evaluating the distribution of free aMV and aMV@GEL after intranasal administration. ~~The retention of free aMV and aMV@GEL in the nasal cavity was assessed with in vivo imaging system at 0, 1, 2, 3, 4, 6, and 8 h. In another parallel experiment, mice were sacrificed at 8 h. The corresponding nasal cavity was excised for section analysis, and the distribution of free aMV and aMV@GEL in tissues was examined.~~

Caption of Fig. 5A (abbreviate words from 133 to 20): Schematic for the preparation of MV with α -2,6 sialic acid-overexpression (sMV) and the mechanism of sMV entrapping the H1N1 virus. ~~When H1N1 infected human cells, the hemagglutinin of the H1N1 virus needed to bind to the α -2,6 sialic acid (SA) receptor on the cell~~

~~surface to implement the infection. To interrupt this process, we constructed β -galactoside α -2,6-sialyltransferase 1 overexpressing 293T cells (ST-293T cells) with the ST6GAL1 plasmid. β -galactoside α -2,6-sialyltransferase 1 was expressed on the Golgi of cells and could increase the content of SA modification on the membrane surface protein, thus increasing the expression of SA. sMV could be obtained by treating ST-293T cells with 10 μ g/ml cytochalasin B for 30 min. Using the SA receptor as the bait, these sMV were capable of entrapping the H1N1 virus, thus, thus blocking the infection of cells.~~

4. While suggestions about whether or not the mixture also blocks transmission of virus between hosts (not within one host) can be drawn based on the data, the authors could be more explicit about this. It would be good to know whether the viruses after contact with the nasal cavity of one host cannot be transmitted to a new host.

Response: We appreciate the reviewer's insightful comments and agree that it is important to investigate whether the mixture (MV@GEL) can block the transmission of the virus between hosts. To address this, we have now designed an experiment based on H1N1 transmission model (Fig. R2A), which can be summarized into three parts.

(1) Preparation of infected donors

Each mouse in donor group was infected with 2×10^3 PFU H1N1-CA07 virus on Day 0 by intranasal administration (nasal trip).

(2) Co-housing of donors with intermediate recipients (IR)

For co-housing, we designed a cage divider, which allowed airflow but no direct contact or fomite (including diet and bedding) transmission. On Day 3, two donors were introduced into one side of the cage divider. Meanwhile, two sMV@GEL treated IR and two PBS treated IR were placed on the other side of the divider (downflow from the donor mouse). On Day 6, the half of IR were sacrificed for viral-loading assay.

(3) Co-housing of IR with terminal recipients (TR)

On Day 6, the rest IR were introduced into a new cage, wherein the cage divider separated two sMV@GEL IRs or two PBS IR from two terminal recipients (naïve mice, downflow from the IR mouse). On Day 9, the TRs were sacrificed for infection rate analysis.

As the results shown in Fig. R2B, the mRNA of H1N1 in nasal cavity and lung of sMV@GEL protected IR were significantly lower than that in PBS treated IR, indicating the strong protection provided by sMV@GEL against viral aerosol transmission. In addition, we observed that when TR were exposed to PBS treated IR, five of the six TR mice were infected (Fig. R2C). However, when TR were exposed to sMV@GEL treated IR, none of the TR were infected. These findings suggested that our sMV@GEL had the ability to effectively block the transmission of viral aerosols. We believe above data provided strong evidence that viral aerosol contacts with sMV@GEL treated nasal cavity could not be transmitted to a new host.

Fig. R2 (as Fig. 5M-O in revised version) Blocking effect of sMV@GEL in H1N1-CA07 transmission model. (A) Schematic showing the experimental design of H1N1-CA07 transmission model. (B) Relative level of H1N1-CA07 mRNA in the nasal cavity (left) and lung (right) in the PBS IR and sMV@GEL IR after 3 days exposed to infected donor. Each group included three cages and each cage included two donor, two PBS IRs and two sMV@GEL IRs (n=6 biologically independent mice). (C) Number of infected TR in each cage after 3 days exposed to PBS IR (left) or sMV@GEL IR (right) (n=6 biologically independent mice). Each group included three cages and each cage included two IRs and TRs. Data in B represent the means \pm SEM. Statistical significance was tested with one-way ANOVA with multiple comparison tests. **P < 0.01, ***P < 0.001.

We have now added the result, description and method about transmission experiment in our revised manuscript as follow.

Main text:

Page 11-12, Line 321-337: Encouraged by above excellent protective effect of sMV@GEL against virus infection, we were also interested in the performance of sMV@GEL for preventing host-host transmission. To this end, we designed a cage divider, which allowed airflow but no direct contact or fomite (including diet and bedding) transmission (Fig. 5M). Donor mice were infected with H1N1-CA07 virus on Day 0 by intranasal administration. On Day 3, two donors were introduced into one side of the cage divider. Meanwhile, two sMV@GEL treated intermediate recipients (IRs) and two PBS treated IRs were placed on the other side of the divider (downflow from the donor mice). On Day 6, the half of IRs were sacrificed for viral-loading assay, and the rest of IRs were introduced into a new cage, wherein the cage divider separated two sMV@GEL IRs or two PBS IRs from two terminal recipients (TRs, naïve mice, downflow from the IR mice). On Day 9, the TRs were sacrificed for infection rate analysis. As shown in Fig. 5N, the mRNA of H1N1-CA07 in sMV@GEL protected IRs significantly lower than that in PBS IRs, indicating the strong protection provided by sMV@GEL against viral aerosol transmission. In addition, we observed that 5 TRs (5/6) exposed to PBS IRs were infected, while none of TRs exposed to sMV@GEL IRs were infected (Fig. 5O). Above findings supported that our sMV@GEL had the ability to effectively block the host-host transmission of viral aerosols.

Method:

Page 23-24, Line 757-768: The protection effect of sMV@GEL in H1N1

transmission model. We designed a cage divider, which allowed airflow but no direct contact or fomite (including diet and bedding) transmission. Donor mice were infected with H1N1-CA07 virus on day 0 by intranasal administration. On day 3, two donors were introduced into one side of the cage divider. Meanwhile, two sMV@GEL treated intermediate recipients (IRs) and two PBS treated IRs were placed on the other side of the divider (downflow from the donor mice). On day 6, the half of IRs were sacrificed for viral-loading assay, and the rest of IRs were introduced into a new cage, wherein the cage divider separated two sMV@GEL IRs or two PBS IRs from two terminal recipients (TRs, naïve mice, downflow from the IR mice). On day 9, the TRs were sacrificed for infection rate analysis. The mRNA of H1N1-CA07 in the nasal cavity and lung was detected by using above-described method. In q-PCR analysis, the mouse was considered uninfected when relative Cq value was higher than 35.

5. The transgenic mouse model (introduced line 241) for SARS-CoV-2 has been determined over the last years be suboptimal in many ways. It serves its purpose in this paper but the authors should provide appropriate caveats and mention what model might be preferable in the future.

Response: We thank the reviewer for reminding us that transgenic mouse model for SARS-CoV-2 was suboptimal, especially on the aspects of virus tropism, susceptibility to deadly diseases and immune system differences. This need to be taken seriously in the studies focusing on virus transmission, pathology and host immune responses. Accordingly, more researchers alternatively utilized Syrian hamster, ferret, or even non-human-primate with natural expression of ACE2 for SARS-CoV-2 infection, which can acquire more convincing data¹⁻⁴.

In our study, we initially utilized this model because it was currently the most widely used, easily acquired and cost-effective model for studying SARS-CoV-2 infection¹⁻³. We also appreciate the reviewer's comment that it serves its purpose in this paper. According to the reviewer's suggestion, we have now supplemented additional caveats regarding the use of transgenic mouse model for SARS-CoV-2 and indicated some preferable models in the future studies as follow.

Main text:

Page 17, Line 500-505: Although the ACE2 transgenic mouse model was widely used, easily acquired and cost-effective model for SARS-CoV-2 infection, it still has some limitations^{40,65-67}, including the virus tropism, susceptibility to deadly diseases, and immune system differences. Syrian hamster, ferret, and non-human-primate with natural expression of ACE2 can be considered as preferable models in the studies focusing on virus transmission, pathology and host immune responses.

Reference:

40. Bao, L.N., et al. The pathogenicity of SARS-CoV-2 in hACE2 transgenic mice. *Nature* **583**, 830-+ (2020).

65. Jia, H., Yue, X. & Lazartigues, E. ACE2 mouse models: a toolbox for cardiovascular and pulmonary research. *Nat Commun* **11**, 5165 (2020).

66. Winkler, E.S., et al. SARS-CoV-2 infection of human ACE2-transgenic mice causes severe lung inflammation and impaired function. *Nat Immunol* **21**, 1327-1335 (2020).

67. Munoz-Fontela, C., et al. Animal models for COVID-19. *Nature* **586**, 509-515 (2020).

References in response:

1. Jia, H., Yue, X. & Lazartigues, E. ACE2 mouse models: a toolbox for cardiovascular and pulmonary research. *Nat Commun* **11**, 5165 (2020).
2. Winkler, E.S., et al. SARS-CoV-2 infection of human ACE2-transgenic mice causes severe lung inflammation and impaired function. *Nat Immunol* **21**, 1327-1335 (2020).
3. Bao, L.N., et al. The pathogenicity of SARS-CoV-2 in hACE2 transgenic mice. *Nature* **583**, 830+ (2020).
4. Munoz-Fontela, C., et al. Animal models for COVID-19. *Nature* **586**, 509-515 (2020).

6. Line 265. The authors overexpress α 2,6-type receptors for use with influenza virus, but earlier had correctly mentioned the importance of both α 2,3- and α 2,6-type influenza receptors. Influenza viruses target airway epithelial cells via both α 2,3- and α 2,6-type receptors, but the distribution of these receptors in species is uncertain and this influences infection, tropism etc. The authors should justify the choice of using only α 2,6-type and while this is fine as a POC, other linkages may be necessary in the future.

Response: We thank the reviewer for reminding us that influenza viruses target airway epithelial cells via both α 2,3- and α 2,6-type receptors, but the distribution of these receptors in species is uncertain and this influences infection, tropism, etc. We had realized above issues and selected α 2,6-type sialic acid for entrapping the influenza virus. We herein clarify the rationality mainly into two aspects.

- (1) As previously reported, the influenza viruses bind preferentially to α 2,3-type sialic acid in avian, whereas these viruses bind preferentially to α 2,6-type sialic acid in human^{5,6}. Such a tropism supports the utilization of α 2,6-type sialic acid for a superior binding and subsequent entrapment in our study aiming to provide protection to human.
- (2) Beyond convenience for controlling experimental variables, expression of one receptor on MV can facilitate the process of industrial production, thus improving the translational potential.

We also appreciate the reviewer's comment that α 2,6-type sialic acid is fine as a POC. In the revised manuscript, we have now justified the choice of using only α 2,6-type sialic acid and made a perspective on the investigation of other linkages in the future as follow.

Main text:

Page 10, Line 280-281: Considering the tropism of influenza virus to \$\alpha\$ -2,6 SA in human^{41,42}, we herein chose \$\alpha\$ -2,6 SA for overexpression on MV.

Page 17-18, line 505-509: In addition, our sMV@GEL focused solely on the \$\alpha\$ -2,6 SA receptors, and future studies could engineer vesicles overexpressed \$\alpha\$ -2,3 SA receptors or overexpressed both \$\alpha\$ -2,6 and \$\alpha\$ -2,3 SA receptors to investigate the antiviral effects across different species, thus providing guidance for the choice of receptors in different application scenarios.

Reference:

41. Yamada, S., et al. Haemagglutinin mutations responsible for the binding of H5N1 influenza A viruses to human-type receptors. *Nature* **444**, 378-382 (2006).
42. Krammer, F., et al. Influenza. *Nat Rev Dis Primers* **4**, 3(2018).

Reference in response:

5. Yamada, S., et al. Haemagglutinin mutations responsible for the binding of H5N1 influenza A viruses to human-type receptors. *Nature* **444**, 378-382 (2006).
6. Krammer, F., et al. Influenza. *Nat Rev Dis Primers* **4**, 3 (2018).

7. Line 298. Please comment on why the high-dose mice lost weight.

Response: We thank reviewer for the careful reading. In the high-dose group, the mice inhaled the five times higher concentrations of viral aerosols than low dose group. Upon this situation, mice in the PBS group suffered from severe sneezing, obvious inactive behavior, and poor appetite, thus leading to a rapid weight loss. In contrast, mice in the sMV@GEL group showed reduced sneezing, increased activity levels, and improved appetite than PBS group mice, which should be attributed to the substantially reduced viral loading upon sMV@GEL protection. We have now added more description in our revised manuscript as follow.

Main text:

Page 11, line 313-317: Regarding high-dose challenged mice, although the weight of both groups decreased, compared to the PBS group, mice in the sMV@GEL group started losing weight later and recovered more quickly (Fig. S28B), which should be attributed to the substantially reduced viral loading upon sMV@GEL protection.

8. Line 367 and ongoing paragraph. The organoid section lacks necessary context; the infected cells presumably are (should be) the cells lining the center of the organoid, not the outer surface. So, what do the authors take by infectivity at the outside of the organoid? What does this mean? There needs to be a little more understanding here of how the organoid is infected in this scenario and whether or how this represents the lung.

Response: We apologize for the omission of detailed organoid introduction at line 367 of originally submitted manuscript. The lung organoid was derived from human lung tissue, had similar cell types (ciliated cell, basal cell and club cell), cellular organization and function with the lung tissue (see next response for detail). This allowed us to gain valuable insights into how the virus interacts with lung cells and its potential impact on lung tissue⁷⁻⁹. We have now added the detail description about lung organoid in our revised manuscript as follow.

Main text:

Page 14, line 395-398: Meanwhile, we also prepared lung organoids from the paracancerous tissue of a lung cancer patient as the “lung module”, which were reported with similar cell types, cellular organization and function to the lung tissue and could be used for evaluating the lung infection in several studies⁵²⁻⁵⁴.

Page 14, line 406-410: We firstly determined that our lung organoids exhibited the

same markers (FOXJ1, ciliated cell marker; SCGB1A1, club cell marker; P63, basal cell marker; and SA, H1N1 receptor) as paracancerous lung tissue⁵²⁻⁵⁴ (Fig. 7C, Fig. S33), which indicated that our lung organoids composed of similar cell types with those in lung tissue and could be used for evaluating the protection performance against H1N1 viral aerosol.

As to the virus infection in organoid, we herein take this opportunity for more clearly describing how our integrated HRT model worked and how the organoid was infected in this scenario.

(1) Working principle of HRT model

In our integrated HRT model, the viral aerosol was produced into the nasal apparatus. From there, the viral aerosol traveled through the nasal apparatus and sterile tube by the inhalation air flow produced by pump. Subsequently, the viral aerosol entered the medium that contained lung organoid, resulting in infection.

(2) Lung organoid infection in HRT model

Upon entering the medium, the virus initially contacted with the outer surface of the organoid. Here, it is worth noting that most lung organoids we obtained were apparently in the form of a hollow sphere, which could be supported by the Z-stack CLSM images (Fig. R3A). Therefore, the viral infection mainly occurred on the outer surface of the organoid (Fig. R3B), wherein the cells tightly arranged to form a hollow sphere.

Fig. R3 Lung organoid (A) and the lung organoid after infection with H1N1-CA07 virus for 24 h (B). The cytoskeleton was stained by FITC-phalloidine (white, false color), cell nuclei were stained with DAPI (blue), and the N protein of H1N1-CA07 was immunofluorescently labeled with an N protein antibody and the corresponding secondary Alexa Fluor® 647 fluorescent antibody (green, false color).

We have now added more description about how lung organoid infected in HRT model in our revised manuscript as follow.

Main text:

Page14, line 412-415: Meanwhile, the respiratory module started to inhale and exhale

those viral aerosols with the respiratory rate, upon which the viral aerosols through the nasal module could reach the “lung module”, enter the culture medium and subsequently infect lung organoids.

Reference in response:

7. Zhou, J., et al. Differentiated human airway organoids to assess infectivity of emerging influenza virus. *Proc Natl Acad Sci U S A* **115**, 6822-6827 (2018).
8. Salahudeen, A.A., et al. Progenitor identification and SARS-CoV-2 infection in human distal lung organoids. *Nature* **588**, 670-675 (2020).
9. Hui, K.P.Y., et al. Tropism, replication competence, and innate immune responses of influenza virus: an analysis of human airway organoids and ex-vivo bronchus cultures. *Lancet Resp Med* **6**, 846-854 (2018).

9. Along these lines, there is inadequate description either here or in the methods about the generation and validation of these human lung organoids. In order to view them as representing human lung, the method of differentiation and validation needs to be clear. The section starting on line 677 (construction of human respiratory tract model) is inadequate – and has no references. This is a significant issue.

Response: We apologize for the inadequate description of the generation and validation of our human lung organoids. In the revised manuscript, we have now added more details and corresponding references in the experimental section (page 25, line 817-823): “For the preparation of lung organoids, we firstly obtained paracancerous tissues from surgically resected sample of a lung cancer patient. The tissues were washed with PBS which containing antibiotics and chopped into approximately 5 mm pieces. And tissues were further washed with 10 ml advanced DMEM/F12 which containing 1× Glutamax, 10 mM HEPES, and antibiotics, then digested in 10 ml advanced DMEM/F12 with 2% fetal calf serum and 2 mg/ml collagenase, shaking at 37 °C for 1-2 h. Dissociated cells were collected at 400 g, suspended in complete human organoid medium with 10% growth factor reduced Matrigel⁷⁷⁻⁷⁹.”

In order to validate the organoids representing the lung features, we also performed immunostaining using specific markers to identify different cell types and receptors within the organoids: FOXJ1 for ciliated cells, SCGB1A1 for club cells, P63 for basal cells, and SA for the H1N1 receptor. As shown in Fig. R4, the lung organoids exhibited an outer layer consisting of ciliated cells, club cells, and basal cells, indicating that the organoids had similar cell types to human lung tissue. Additionally, the expression of SA on the surface of the organoids indicated their susceptibility to influenza virus. According to the comment 2 of reviewer 3, we have now updated Fig. 7C with Fig. R4A, added Fig. R4B as Fig. S33, and provided more information for the immunostaining characterization in revised manuscript (page 25, line 823-827): “The lung organoids were harvested until those cells formed spheroids. For identification, the lung organoids were stained with FOXJ1, SCGB1A1, or P63 antibodies and corresponding secondary Alexa Fluor® 647 fluorescent antibodies, respectively. The SA in the lung organoid was stained with Cy3-SNL. The cytoskeleton was stained by FITC-phalloidine, and cell nuclei were stained with DAPI.”

Reference:

77. Nadkarni, R.R., Abed, S. & Draper, J.S. Organoids as a model system for studying human lung development and disease. *Biochem Biophys Res Commun* **473**, 675-682 (2016).

78. Chiu, M.C., et al. A bipotential organoid model of respiratory epithelium recapitulates high infectivity of SARS-CoV-2 Omicron variant. *Cell Discov* **8**, 57 (2022).

79. Kim, J., Koo, B.K. & Knoblich, J.A. Human organoids: model systems for human biology and medicine. *Nat Rev Mol Cell Biol* **21**, 571-584 (2020).

Fig. R4 (Fig. R4A as Fig. 7C, Fig. R4B as Fig. S33 in revision) Cellular characterization of the lung organoid. The representative CLSM images (A) and corresponding 3D reconstructed CLSM images (B) showing expression of FOXJ1 (ciliated cell marker), SCGB1A1 (club cell marker), P63 (basal cell marker), and SA (H1N1 receptor) in lung organoids. FOXJ1, SCGB1A1, and P63 were stained with corresponding antibodies and secondary Alexa Fluor® 647 fluorescent antibodies (red), SA was stained with Cy3-SNL (red, false color), cytoskeleton was stained with FITC-phalloidine (white, false color), and cell nuclei were stained with DAPI (blue).

10. Line 381 and elsewhere. The mention of the strategy being “independent of virus mutation” is too unspecific and is premature. The authors may mean “serotype” here, not mutation. Or perhaps CA07 vs. PR8 is considered a different of “mutation” but this would not be not a standard way to express this. The statement that there is indifference to strain or sequence (if this is as meant) is premature and overstated and should be re-worded and tempered.

Response: We thank the reviewer for pointing out the unstandardized expression of “independent of virus mutation” in influenza-related section. Accordingly, we have now rephrased corresponding statements as follows.

Main text:

Page 2, line 47-49: Initially, using mouse models, we show that MV@GEL could protect the nasal cavity and downstream lung of mice against viral aerosols, and this protection is independent of viral mutation/serotype.

Page 4, line 103-106: The unique merits of independent ability to viral mutation/serotype, flexible receptors engineering for matching different types of viruses, and translational potential supported our intranasal mask as a promising shield modality for improving public health in the pandemic (Fig. 1C).

Page 10, line 296-298: Furthermore, sMV also achieved a similar protective effect

against the H1N1-PR8 virus (Fig. S27), suggesting that its protection against the H1N1 virus was independent of the serotype.

Reviewer #2 (Remarks to the Author):

Hu et al reported an interesting intranasal mask to entrap virus for the protecting the respiratory tract against viral aerosols. While engineering cell-derived vesicles to express viral binding receptors for entrapping viruses is not a new idea, using hydrogels as an 'intranasal mask' may provide some additional respiratory protection.

Response: We appreciate the reviewer's positive comments for our intranasal mask idea. Taking this opportunity, we herein clarify the advance of our micro-sized cell-derived vesicles over previously reported nano-sized cell-derived vesicles. Take SARS-CoV-2 for example, the nanosized cell-derived vesicles could only inactivate a portion of active S proteins on the surface of virus, with other active S proteins on the viral surface remaining infectivity to host cells. While the micro-sized cell-derived vesicles in our study could entrap virus, leading to the shielding and inactivation of virus. In our originally submitted manuscript, we had provided the discussion in the aspect of the superiority of our macrovesicles (Page 14, line 403-408): "It should be emphasized that the ultrasmall size of the nanovesicles (e.g., ~100 nm for exosome) or the solid core of the membrane-coated particles (e.g., polymeric particle coated with ACE2-membrane) were unfavorable for harnessing the potential of entrapping virus inside, which failed to completely isolate the virus from host cells. In the present study, we rationally prepared MV displaying viral receptors and demonstrated their virus entrapment ability."

Main concerns:

1. MV may not be a viable treatment option in clinics. As each patient's human leukocyte antigen (HLA) is different (with the exception of identical twins), to avoid unwanted immune responses, the authors have to make a personalized MV for each patient, which would be both expensive and impractical. HLA-matched cells must be engineered for individual patients, which can result in exorbitant costs and is thus unattainable by many patients.

Response: According to the reviewer's concern about the potential challenges of MV translation, we herein made some clarifications. For invasive administration (such as intravenous injection and subcutaneous injection), we agree with the reviewer's opinion that HLA-matched or personalized MV is indeed necessary. However, in our study, the MV was designed for topical application in nasal cavity rather than aforementioned invasive administration. Moreover, the embedment of MV in the gel material ensured the stable retention upper the mucosa rather than penetration into the submucosa. These two points together excluded the possibility of inducing unwanted immune responses, which could be supported by our supplemented new data (blood routine analysis, serum biochemistry index analysis, nasal inflammatory cytokine detection and histological analysis of major tissues) in the next response (Fig. R5).

We hope above explanation now can eliminate the reviewer's concern about the translation potential. We have now emphasized this issue in the discussion section of revised manuscript as follow.

Main text:

Page 17, line 490-493: On one hand, the system is used for topical application, which do not require HLA matching or individualization of MV. On the other hand, chitosan

and β -sodium glycerophosphate are materials approved by the FDA, exhibiting high biocompatibility for intranasal use.

2. MVs are composed of random unknown components from 293T cells which can result in significant variability and safety concerns and can induce unwanted immune responses.

Response: As we explained above, the topical application of MV excluded the possibility of causing unwanted immune response. For verification, we have now conducted additional safety evaluations of our MV@GEL. Briefly, MV@GEL was intranasally administered to mice every other day for a period of two weeks. At day 7 and day 14, we collected blood samples for blood routine and serum biochemistry index analysis, prepared nasal homogenate for inflammatory cytokine detection, and excised major tissues for histological analysis.

As shown in Fig. R5A and R5B, the blood routine analysis of white blood cells (WBC), red blood cells (RBC), and platelets (PLT), as well as the serum biochemistry indices of blood urea nitrogen (BUN), aspartate alanine aminotransferase (ALT), aminotransferase (AST), lactate dehydrogenase (LDH), and alkaline phosphatase (ALP) of mice receiving MV@GEL were all within the normal reference ranges. Furthermore, the levels of cytokines (IFN- γ , TNF, IL-2 and IL-6) in Fig. R5C exhibited no significant difference between MV@GEL group and health group, suggesting that MV@GEL did not induce an inflammatory response in the nasal cavity. In addition, the H&E staining results in Fig. R5D indicated that long-term use of MV@GEL did not cause apparent damage to the nasal cavity, brain, lung, heart, liver, spleen, and kidney. Beyond the data of basic respiratory function and structure of nasal epithelium layer provided in the originally submitted manuscript, above data further provided strong evidence to support the safety of MV@GEL material upon topical application in the nasal cavity.

Like other cellular therapeutics, MVs are composed of random unknown components, which require special considerations for the quality control. Various techniques such as Western blotting, mass spectrometry, RNA sequencing, and lipidomic analysis are employed to achieve this. These analyses provide valuable insights into the components of MVs, enabling batch-to-batch consistency. By employing these quality control measures, the composition and purity of MVs can be thoroughly evaluated, ensuring their safety and efficacy for topical application.

Fig. R5 (as Fig. S21 in revision) Biosafety evaluations upon multiple MV@GEL administrations. MV@GEL was intranasally administered to mice every other day for a period of two weeks. At day 7 and day 14, the blood and the major tissues (including the nasal cavity, brain, lung, heart, liver, spleen, and kidney) were collected for biosafety evaluations.

(A) Blood analysis of white blood cells (WBC), red blood cell (RBC), and platelet (PLT) at day 7 and day 14 (n=3 biologically independent mice). The blue background presented the normal range.

(B) Serum biochemistry indices analysis of urea nitrogen (BUN), aspartate alanine aminotransferase (ALT), aminotransferase (AST), alkaline phosphatase (ALP), and lactate dehydrogenase (LDH) at day 7 and day 14 (n=3 biologically independent mice). The blue background presented the normal range.

(C) Inflammatory cytokines (IFN-γ, TNF, IL-2 and IL-6) analysis of the nasal homogenate at day 7 and day 14 (n=3 biologically independent mice).

(D) Representative H&E staining images of major tissues at day 7 and day 14.

Data in A, B and C represent the means ± SEM. Statistical significance in C was calculated using one-way ANOVA with multiple comparison tests. n.s. meant no significant difference.

Above data have now been added as Fig. S21, and corresponding description and experimental method have been added in revised version as follow.

Main text:

Page 8-9, line 236-239: In addition, the comprehensive evaluations, including blood routine analysis, serum biochemistry index analysis, nasal inflammatory cytokine detection, and histological analysis of major tissues, confirmed the safety upon multiple aMV@GEL administrations (Fig. S21).

Method:

Page 21, line 653-661: The biosafety evaluations of the long-term intranasal administration of aMV@GEL. The mice were intranasally administered with aMV@GEL every other day in a period of two weeks. The blood and the major tissues (including the nasal cavity, brain, lung, heart, liver, spleen, and kidney) were collected at both day 7 and day 14. The blood was used for blood routine (WBC, RBC and PLT) and biochemical indicators (BUN, ALT, AST, LDH and ALP) analysis by using automatic haemocytometer analyzer and biochemical analyzer, respectively. The inflammatory cytokines (IFN-γ, TNF, IL-2 and IL-6) of the nasal homogenate were detected by the corresponding ELISA kit. The major tissues were fixed and prepared

into paraffin sections (3-4 \$\mu\text{m}\$ ) for H&E staining.

I also have the following concerns:

1. a. The thermosensitive hydrogel is one of the main components of MV@GEL. However, the actual thermosensitivity regarding the critical temperature and time duration leads to transition from liquid to gel was not characterized.

Response: In the supplementary material of originally submitted manuscript, we had provided the gelation time curves of MV@GEL at 37 °C in Fig. S9 (also shown as Fig. R6 here). The MV@GEL transitioned from liquid to gel at 368 s and the corresponding temperature was 37°C, which had also been emphasized at page 7, line 186-189: “Compared with the GEL (Fig. S8), aMV@GEL maintained many similar features in terms of its porous structure (aperture size approximately 10-20 μm , Fig. 3C), thermosensitive gelling property (368 s at 37 °C, Fig. 3D and Fig. S9), spray characteristic (spray area 2.3 cm^2 , 10 cm away from the nozzle, Fig. 3E), paving the way for application in the nasal cavity.”

Fig. R6 (Fig. S9 in originally submitted manuscript) Evolution of dynamic loss modulus (G' , red) and storage modulus (G'' , gray) of aMV@GEL at 37°C. When $G'' > G'$, aMV@GEL transformed from liquid state to gel state.

b. This needs to be carefully studied considering an actual case when a patient is holding a spray of the MV@GEL using warm hands. What if it turns into gel before even the spray?

Response: We thank the reviewer for reminding us this issue. Given that the gelling time of 368 s at 37°C and the storage in a refrigerator, we envision that holding a spray with warm hands will not make MV@GEL rapidly turning into gel. For verification, we have now carried out additional experiment under the following scenario (Fig. R7A): the MV@GEL that stored in 4°C refrigerator was taken out to room temperature and then held with warm hand for various times (1 min, 3 min and 5 min). As shown in the photographic images of Fig. R7B, the MV@GEL maintained liquid state even at 5 min, thus providing sufficient time window for the spray into the nasal cavity.

Fig. R7 (as Fig. S13B-C in revision) Photographic images of aMV@GEL that was taken out from the refrigerators and was held with warm hand (A) for the indicated times (B).

Above data and corresponding caption have now been added in Fig. S13 in our

revised manuscript.

c. Can it transit from gel to liquid freely and does it change the performance of the MVs loaded?

Response: The thermosensitive gelation of our MV@GEL formulation is irreversible. In other word, once MV@GEL is sprayed into the nasal cavity and undergoes gelation, it will not revert back to a liquid state, ensuring the performance of the loaded MVs. We have now emphasized the irreversibility of the gel's thermosensitive gelation in the revised manuscript as follow.

Main text:

Page 3, Line 79-81: Pursuing this, we engineered an intranasal mask composed of an irreversibly thermosensitive hydrogel with positive charges, into which we introduced engineered cell-derived micro-sized vesicles (MV).

Page 6, Line 176-178: We had experience working with thermosensitive hydrogels, and in the present study, we explored the use of chitosan and β -sodium glycerophosphate to prepare irreversibly thermosensitive hydrogels with good wettability and adhesion property^{35,36}.

d. The thermosensitive gel also adds instability and risks for delivery during severe weather conditions like high environmental temperature.

Response: We agree with the reviewer's concern about additional instability and risks for delivery during severe weather conditions like high environmental temperature. In our originally submitted manuscript, we had provided corresponding discussions and perspectives about this point (page 15, line 425-434): "Considering the thermosensitive property of our gel, some points should be noted. First, our hydrogel should be stored at low temperature (below 25 °C) to avoid unexpected gelatinization. Moreover, people should avoid extremely cold environments when using our intranasal mask because the excessively low ambient temperature (below 8 °C) may delay the gelation time in the nose. To further upgrade our hydrogel formulation, we can use some special hydrogels with shear-thinning properties, in which gelatinization was independent of the temperature. For example, peptide-based hydrogels, protein-based hydrogels, and hydrogels from blends can be sprayed into the nasal cavity under shear stress and immediately recover their mechanical properties^{54,55}, thus remaining fixed at the inner wall of the nasal cavity."

2. The authors mentioned the hydrogel with positive charges could trap some intercepts the negatively charged viral aerosols. Such an interaction seems not specific to viruses. Could it also bind some negatively charged cell fragments or bacteria? If so, would it block some interaction of MVs and actual viruses?

Response: We agree with the reviewer that the positive charges on the hydrogel may facilitate interactions with various negatively charged particles, including cell fragments or bacteria. Even though, we still envision that the unspecific interception will not substantially compromise the performance of our MV@GEL. On the one hand, the viral aerosol counts are much higher than cell fragment and bacteria counts in the

high-risk environments. On the other hand, even if a few cell fragments and bacteria are absorbed in the gel, it is unlikely to impede the diffusion of viruses within the gel. Importantly, the specific trapping of viruses by MV occurs through receptor-mediated binding, allowing for targeted capture despite the presence of other particles.

3. The authors showed some viruses were entrapped by MVs, while there are still some viruses on the surface of the MVs (Step3, Fig1A). The viruses trapped on the cell surface may have the possibility to infect cells, particularly for authentic viruses.

Response: We apologize for the apparently inappropriate presentation in the schematic diagram (step3 in Fig. 1A, also shown as Fig. R8A) of our original submitted manuscript, which caused the reviewer's misunderstanding that there were still some viruses on the surface of the MVs in practical application. In this panel, we had intended to illustrate the mechanism that viruses entered MV by binding to the surface receptors on MV. To avoid misunderstanding, we have now revised this schematic diagram in our revised manuscript as the following Fig. R8B.

Fig. R8 (A) Step 3 of Fig. 1A in originally submitted manuscript. (B) Step 3 of Fig. 1A in revised version.

To further address the reviewer's concerns, we have now performed additional experiments to compare the antiviral effect of aNV (nanosized vesicles displaying ACE2 for binding to virus) and aMV (microsized vesicles displaying ACE2 for binding and further entrapping virus). aNV and aMV were separately incubated with SPV for sufficient time and then were added to ACE2-293T cells. As shown in Fig. R9, the antiviral effect of aNV was significantly lower than that of aMV, which could be attribute to the incomplete inactivation of virus in aNV group. The virus binding on the surface of aNV still had the possibility to infect cells, while the virus entrapped by aMV was completely shielded from host cells, thus leading to a much superior antiviral effect.

Fig. R9 Representative CLSM images (A) and flow cytometer assay (B) showing the infection effect of the SPV after binding to aNV (SPV-aNV) or entrapped inside aMV (SPV@aMV) (n=3 biologically independent experiments). Data in B represent the means \pm SEM. Statistical significance in B was calculated using one-way ANOVA with multiple comparison tests. ****P < 0.0001.

As for authentic virus, our result had demonstrated the protection effect of sMV against authentic influenza viruses in Fig. 5 and Fig. 7 of our originally submitted

manuscript, indicating that the MV displaying viral receptor could entrap authentic viruses and subsequently inactivate the viruses.

4. On page 5, the authors first showed that SPV signal on the aMV surface (Fig. S3A) and then showed that overtime, the SPV signal would be seen in the interior of aMV (Fig. 2G-H). The authors should provide the control of using nMV in Fig. 2G-H to exclude some nonspecific binding in this prolonged time point.

Response: We appreciate the reviewer's careful review. Beyond providing the STED image of nMV after incubation with virion for 30 min (Fig. S3 in the originally submitted supplementary information), we herein further added the corresponding STED and TEM images captured at 3 h. As shown in Fig. R10, the virus signal neither on the surface of nMV nor within the nMV, thus excluding nonspecific binding in this prolonged time point.

Fig. R10 (as Fig. S4B-D in revision) STED images of nMV incubation with SPV for 30 minutes (A) and 3 hours (B), accompanying with TEM image of nMV incubation with SPV for 3 hours (C). For STED observation, SPV (wild-type) were stained with FITC (green), and the membrane of nMV was stained with DiD (red).

The above data have now been reorganized in Fig. S4, along with description following the caption.

Mian text:

Page 5-6, line 147-149: In contrast, no obvious SPV signal appeared on or in nMV (Fig. S4B-Fig. S4D), thus excluding nonspecific binding of SPV on MV and again suggesting that ACE2 specifically mediated the entrapment of SPV into aMV.

5. On page 7, the authors mentioned the size distribution of 1 to 10 μm (Fig. S10). However, the a-axis is nm instead of μm . This needs to be corrected and proper citations need to be added.

Response: We appreciate the reviewer's careful review and apologize for the mistake in the Fig. S10 of our previous version. We have now revised the incorrect unit to " μm " in a-axis of this figure (also shown as Fig. R11). Additionally, proper citation has been added in the caption of this figure (*BMC Infect Dis* **19**, 101 (2019)).

Fig. R11 (as Fig. S16 in revision) The size distribution of aerosols produced by the aerosol generator. The vast majority of viral aerosols ranged from 1 µm to 10 µm, which was the same as real viral aerosols¹.

Reference in response:

1. Tellier, R., Li, Y., Cowling, B.J. & Tang, J.W. Recognition of aerosol transmission of infectious agents: a commentary. *BMC Infect Dis* **19**, 101 (2019).

6. On page 9, the quantification of relative expression of GFP does not seem to be consistent with the fluorescent images in Fig. 4G. The GFP intensity of Gel group seems much lower than that of PBS in the images, while the quantification showed similar intensities (n.s.). The authors should double check the quantification results and also clearly mention the method to do the quantification. Also, only 3 (N=3) seemed insufficient for such image analysis, considering the region-of-interest and varieties of such imaging experiments.

Response: We appreciate the reviewer’s careful review. Taking this opportunity, we herein clarify that the quantification of GFP was conducted by q-PCR analysis, which could avoid the region-of-interest and varieties during image analysis. In this aspect, 3 biologically independent experiments were sufficient for q-PCR analysis.

In corresponding captions and experimental section of our originally submitted manuscript, we had mentioned the quantification was related to GFP mRNA. Even though, we still apologize for our inappropriate descriptions of “relative expression of GFP” for the Y-axis in Fig. 4G-4I of our originally submitted manuscript, which have now been revised into “relative level of GFP mRNA”. For clear descriptions, we have also revised these captions as follow.

Main text:

Fig. 4G caption: Representative frozen section images and enlarged views of mouse nasal cavity (left) and relative level of GFP mRNA in the nasal cavity by using q-PCR detection (right, n=3 biologically independent mice) after 3 days of challenge with SPV (wild-type) aerosols. GFP was produced by SPV-infected cells (green), and the cell nuclei were stained with DAPI (blue).

Fig. 4H caption: Representative frozen section images and enlarged views of mouse lung (left) and relative level of GFP mRNA in the lung tissue by using q-PCR detection (right, n=3 biologically independent mice) after 3 days of challenge with SPV (wild-type) aerosols. GFP was produced by SPV-infected cells (green), and the cell nuclei were stained with DAPI (blue).

Fig. 4I caption: Relative level of GFP mRNA that was produced by SPV-infected cell

in nasal cavity and lung using q-PCR detection after 3 days of challenge with SPV (S protein B.1.1.529 mutation (Omicron)) aerosols (n=3 biologically independent mice).

7. The advantage of protecting mice weight loss in influenza infection is marginal. In Fig. 5M, the weight losses in day 10 day are comparable between sMV@Gel and PBS groups.

Response: Taking this opportunity, we herein make more clarifications for the data of weight loss. At a first glance, the weight of the two groups appeared similar on Day 10. However, it is important to note that at this time point three mice (50%) in the PBS group had already died due to severe infection, and their weight data were not included in the analysis. Therefore, analyzing significant differences in weight at this time point, in our opinion, was inappropriate. Alternatively, the difference of these two groups was described as “Regarding high-dose challenged mice, although the weight of both groups decreased, compared to the PBS group, mice in the sMV@GEL group started losing weight later and recovered more quickly (Fig. S28B), which should be attributed to the substantially reduced viral loading upon sMV@GEL protection.” Specifically, mice in the PBS group suffered from severe sneezing, obvious inactive behavior, and poor appetite, thus leading to a rapid weight loss. In contrast, mice in the sMV@GEL group showed reduced sneezing, increased activity levels, and improved appetite than PBS group mice, which should be attributed to the substantially reduced viral loading upon sMV@GEL protection, thus leading to lose weight later and recovered more quickly.

We have now moved Fig. 5M to supplementary information as Fig. S28B in revision and have emphasized that the weight data of PBS group mice that succumbed to the infection were excluded from the analysis as follow.

Mian text:

Page 11, line 313-317: Regarding high-dose challenged mice, although the weight of both groups decreased, compared to the PBS group, mice in the sMV@GEL group started losing weight later and recovered more quickly (Fig. S28B), which should be attributed to the substantially reduced viral loading upon sMV@GEL protection.

Fig. S28B caption: Weight change curves of mice in the PBS and sMV@GEL groups within 14 days after being challenged with high dose H1N1-CA07 aerosols (n=6 biologically independent mice). One mouse in PBS group succumbed to the infection on day 6, two on day 10, and one on day 12. The weight data of these mice were excluded from the analysis at the corresponding days.

8. Although the authors used transgenic ACE2 mice to test the protection effect of aMV@GEL, however, they only used pseudovirus and cannot represent the authentic virus and claim the protection effects. What if they challenged the aMV@GEL group using real SARS-CoV2 viruses? Also, would the protection be the same or different across different variants?

Response: We appreciate the reviewer's comment. While we acknowledge the significance of testing the protective effects of aMV@GEL against the authentic SARS-CoV-2 virus, we regret to inform that our cooperative P3 laboratory is restricted from

using authentic SARS-CoV-2 virus aerosols due to policy and regulatory constraints set by relevant administrative departments. However, as an alternative approach, we had conducted infection experiments using authentic influenza viruses, which allowed us to evaluate the protective efficacy of our intranasal mask (sMV@GEL) against aerosolized authentic viruses. The results depicted in Fig. 5 and Fig. 7 demonstrated that sMV@GEL significantly reduced virus infection from authentic H1N1 viral aerosol. Moreover, in the revised manuscript, we further provided data about the efficient protection effect of our intranasal mask in the transmission model upon authentic influenza virus (see more details in the reviewer 1 comment 4). We hope the reviewer could understand the difficulty of generating authentic SARS-CoV-2 virus aerosols in P3 laboratory and accept the utilization of aerosolized authentic influenza viruses as an alternative.

Taking this opportunity, we herein emphasize that aMV's prominent performance was independent of SARS-CoV-2 mutation. This unique merit should be attributed to the SARS-CoV-2's naturally binding to the ACE2 protein on the surface of MV, which remained unaffected by viral mutations. For verification, we had evaluated the antiviral effect of aMV against 9 different strains of SPV and the protection effect of aMV@GEL against B.1.1.529 (Omicron) pseudoviral aerosol in mouse. As Fig. 2J and Fig. 4I shown in our originally submitted manuscript, the antiviral effect of aMV and the protection effect of aMV@GEL was independent on viral mutations. As aforementioned, we regret that our cooperative P3 laboratory is restricted from using authentic SARS-CoV-2 virus aerosols. Alternatively, we had evaluated the antiviral effect of sMV against two different serotypes of real H1N1 virus (H1N1-CA07 and H1N1-PR8) and the protection effect of sMV@GEL against above two serotypes of real H1N1 viral aerosol in mouse. As Fig. 5G, Fig. S18, Fig. 5J and Fig. S20 shown in our originally submitted manuscript (also shown as Fig. 5G, Fig. S27, Fig. 5J and Fig. S30 in revision), the antiviral effect of sMV and the protection effect of sMV@GEL was independent on viral serotypes.

We hope above explanations and additional experimental data can now eliminate the reviewer's concern about the performance of our intranasal mask against real viral aerosols. Meanwhile, we have now added more descriptions to emphasized the merit that that aMV's prominent performance was independent of mutation/serotype in the revised manuscript as follow.

Main text:

Page 6, line 162-168: Moreover, aMV also persistently showed potent inhibition of host cell infection against the variants of SPV, including D614G, the variant of interest (B.1.617.1, P.2, B.1.429), and the variant of concern (B.1.1.7, B.1.351, P.1, B.1.617.2 (Delta), B.1.1.529 (Omicron)) (Fig. 2J), highlighting the unique merit that aMV's prominent performance was independent of SARS-CoV-2 mutation. This merit could be attributed to the SARS-CoV-2's naturally binding to the ACE2 protein on the surface of MV, which remained unaffected by viral mutations.

Page 16, line 453-455: In the present study, we rationally prepared MV displaying viral receptors and demonstrated their virus entrapment ability, which could complete shielding the virus and could further reduce the infection risk.

9. The reviewer appreciates the attempts of using computational fluid dynamics-discrete particle method (CFD-DPM) and integrated human respiratory tract (HRT) model to investigate the protection of respiratory system. Could the authors describe how many patients they used for reconstructing the CT data considering the possible differences between individuals?

Response: We thank the reviewer for the appreciation on our usage of CFD-DPM and HRT model. As there may be slight variations in the shape and position of the nasal septum and turbinate among adult individuals², it will have no significant influences on the simulation calculations. Therefore, the nasal CT data of only one healthy adult volunteer had been used for the proof-of-concept validation in our initial manuscript.

Even though, we still appreciate the reviewer's comments, which reminded us to note significant differences of nasal cavity between adults and children. While the overall structure of the nasal cavity in children is similar to that of adults, there is a notable difference in size. Accordingly, we have now scaled down the adult nasal cavity model to represent a 12-year-old child nasal cavity, using a scaling factor of 0.65³⁻⁵ (Fig. R12A). The resulting child digital nasal cavity model was then employed for computer simulations to assess the interception effect of MV@GEL. Additionally, we utilized 3D printing technology to construct an authentic child nasal apparatus for updating the nasal module in HRT model, and the child-derived HRT model was integrated following the construction method of HRT model. Note that all conditions in child-derived HRT model were kept consistent with HRT, except for the adjustment of the breath air flow rate from 10 L/min to 8 L/min.

In the computer simulations, we calculated the distribution of viral aerosols in the child nasal cavity model during the inhalation process of 1.5 seconds. As shown in Fig. R12B, the unprotected situation showed that approximately 60.4% of viral aerosols entered the downstream trachea from the nostril with the airflow in the end of inhalation. This proportion value was higher than that of adult nasal cavity model, which could be attributed to the shorter airway of child's nasal cavity compared to adult nasal cavity. When the MV@GEL was applied, a large number of aerosol particles accumulated in the anterior part of the nasal cavity, and only 3.4% of the viral aerosols entered the downstream trachea from the nostril (Fig. R12B). Such a superior interception effect over that in adult nasal cavity model could be explained by the narrower airway in child's nasal cavity than that in adult, which leading to the stronger charge interaction between the MV@GEL with viral aerosol. Owing to above potent interception performance of MV@GEL in the child nasal cavity model, we again observed with child-derived HRT model that the lung organoid in sMV@GEL group was not infected after 48 h (Fig. R13).

Fig. R12. (as Fig. S32 in revision) Prediction of viral aerosol interception effect of MV@GEL in child digital nasal cavity by using CFD-DPM in simulation.

A. The comparison of 3D human adult and child digital nasal cavity model.

B. The distribution of inhaled viral aerosols (blue dot) at different time points (0.3 s, 0.6 s, 0.9 s, 1.2 s, and 1.5 s) of unprotection situation (upper) or MV@GEL situation (lower) after the beginning of inhalation (left), and the corresponding percentage of viral aerosols that retained in the children nasal cavity or flowed into trachea at 1.5 s (right).

Fig. R13. (as Fig. S35 in revision) The protection effect of sMV@GEL in child-derived HRT model.

A. Representative 3D reconstructed CLSM images of lung organoids in different groups after 48 h of challenge with H1N1-CA07 aerosols. The cytoskeleton was stained by FITC-phalloidine (white, false color), cell nuclei were stained with DAPI (blue), and the N protein of H1N1-CA07 was immunofluorescently labeled with an N protein antibody and the corresponding secondary Alexa Fluor® 647 fluorescent antibody (green, false color).

B. Relative level of H1N1-CA07 mRNA of lung organoids in different groups after 48 h of challenge with H1N1-CA07 aerosols (n=3 biologically independent experiments).

Data in A represent the means \pm SEM. Statistical significance in B was calculated using t-tests. ****P < 0.0001.

We believe above supplemented data can support the potent performance of our MV@GEL group for different individuals. Above data, the corresponding description and method have now been added in revised manuscript as follow.

Main text:

Page 13, line 385-388: However, almost no viral aerosol flowed out to the atmosphere at the end of the breath when MV@GEL was applied. Note that we extended above

simulation to a child's 3D digital model and observed much similar result (Fig. S32), showing the applicability for protecting children from the viral aerosols infection.

Page 15, line 425-427: Similar results were obtained when we repeated this experiment with H1N1-PR8 viral aerosols (Fig. 7F) or with child-derived HRT model (nasal module updated with a realistic child nasal apparatus, Fig. S35).

Method:

Page 24, line 770-772: Based on the nasal CT data from a healthy adult volunteer, the digital human nasal model was reconstructed by 3D-slicer software. The child nasal model was obtained by scaling down the adult nasal cavity model, using a scaling factor of 0.65^{75,76}.

Page 26, line 852-855: The child-derived HRT model was integrated by following the construction method of HRT model. The breath air flow rate was adjusted to 8 L/min from the 10 L/min of HRT model. The protection effect of sMV@GEL in child-derived HRT model was evaluated following the method in HRT model.

Reference:

75. Golshahi, L., Noga, M.L., Thompson, R.B. & Finlay, W.H. In vitro deposition measurement of inhaled micrometer-sized particles in extrathoracic airways of children and adolescents during nose breathing. *J Aerosol Sci* **42**, 474-488 (2011).

76. Sawant, N. & Donovan, M.D. In Vitro Assessment of Spray Deposition Patterns in a Pediatric (12 Year-Old) Nasal Cavity Model. *Pharm Res* **35**, 108 (2018).

Reference in response:

2. Zhao, K., Scherer, P.W., Hajiloo, S.A. & Dalton, P. Effect of anatomy on human nasal air flow and odorant transport patterns: implications for olfaction. *Chem Senses* **29**, 365-379 (2004).

3. Foo, M.Y., Sawant, N., Overholtzer, E. & Donovan, M.D. A Simplified Geometric Model to Predict Nasal Spray Deposition in Children and Adults. *AAPS PharmSciTech* **19**, 2767-2777 (2018).

4. Golshahi, L., Noga, M.L., Thompson, R.B. & Finlay, W.H. In vitro deposition measurement of inhaled micrometer-sized particles in extrathoracic airways of children and adolescents during nose breathing. *Journal of Aerosol Science* **42**, 474-488 (2011).

5. Sawant, N. & Donovan, M.D. In Vitro Assessment of Spray Deposition Patterns in a Pediatric (12 Year-Old) Nasal Cavity Model. *Pharm Res* **35**, 108 (2018).

Minor point:

1. The author should properly cite existing literature that already showed the cell-derived vesicles can trap viruses. There are several have been published. On page 4, the authors mentioned 'we envisioned that the cell-derived vesicles that contained overexpressed receptor on the surface could potentially entrap the virus' without proper citations.

Response: We again thank the reviewer for reminding us some studies that reported apparently similar TRAP idea. Accordingly, we have now added the existing literature about the antiviral effect of cell-derived vesicles and revised corresponding description of MV as follow.

Main text:

Page 4, Line 109-112: Inspired by receptor-mediated viral infections in host cells²³, studies that using viral receptor as virus decoy²⁸⁻³¹, and the size of most virus ranging from 60 nm to 140 nm³², we envisioned that the cell-derived vesicles with abundant viral receptors and a large cavity could facilitate the viral entrapment³³.

Reference:

23. Hoffmann, M., et al. SARS-CoV-2 Cell Entry Depends on ACE2 and TMPRSS2 and Is Blocked by a Clinically Proven Protease Inhibitor. *Cell* **181**, 271-280 e278 (2020).
28. Porotto, M., Yi, F., Moscona, A. & LaVan, D.A. Synthetic protocells interact with viral nanomachinery and inactivate pathogenic human virus. *PLoS One* **6**, e16874 (2011).
29. Li, Z., et al. Cell-mimicking nanodecoys neutralize SARS-CoV-2 and mitigate lung injury in a non-human primate model of COVID-19. *Nat Nanotechnol* **16**, 942-951 (2021).
30. Coccozza, F., et al. Extracellular vesicles containing ACE2 efficiently prevent infection by SARS-CoV-2 Spike protein-containing virus. *J Extracell Vesicles* **10**, e12050 (2020).
31. Wang, C., et al. Membrane Nanoparticles Derived from ACE2-Rich Cells Block SARS-CoV-2 Infection. *ACS Nano* **15**, 6340-6351 (2021).
32. Yao, H., et al. Molecular Architecture of the SARS-CoV-2 Virus. *Cell* **183**, 730-738 e713 (2020).
33. Zhang, S., Gao, H. & Bao, G. Physical Principles of Nanoparticle Cellular Endocytosis. *ACS Nano* **9**, 8655-8671 (2015).

Overall, this is manuscript has done significant work to characterize the MV@GEL and its protection. However, the novelty and biosafety are the primary concerns in the current format.

Response: Beyond explaining the advances of our MV over previously reported nano-sized vesicle as aforementioned, we herein further provided another two aspects that could strengthen our novelties.

(1) Cooperation with GEL for intranasal interception

Unlike previous studies that aimed at the virus itself, our designed “intranasal mask” was the successful paradigm to use biomaterial to protect against viral aerosols. The hydrogel could intercept the negatively charged viral aerosols by coulombic interactions.

(2) Multiple advanced models for deep and systemic investigations

In our study, we conducted multiple evaluations to assess the protective effect of sMV@GEL. Except for the traditional aerosol challenge model, we used a further transmission model to investigate its protective effect. Furthermore, we utilized CFD simulations to evaluate the aerosols interception performance of MV@GEL in the human (adult and child) digital nasal cavity. In addition, we assessed the protective effect of sMV@GEL using the HRT model, which closely resembled the respiratory tract in humans. Through these comprehensive evaluations, we aimed to provide a thorough understanding of the efficacy and potential applications of sMV@GEL.

For the reviewer's concern of safety, we have provided detail explanations in the responses to comment 1 and comment 2. Briefly, we emphasized the topical administration route of MV@GEL and supplemented comprehensive safety data.

We believe above supplemented data and explanations could now eliminate the reviewer's concerns about the novelty and biosafety of our MV@GEL.

Reviewer #3 (Remarks to the Author):

Xiaoming Hu et al describes a novel method to prevent viral infections by retaining/entrapping viral aerosols thru the use of an intranasal mask made of a gel containing cell-derived vesicles that display specific viral receptors. The overall design of the experiments and data display are well done. The manuscript includes several innovative approaches to evaluate the safety, efficacy and efficiency of the proposed mask. Nevertheless, some important aspects of this device need to be evaluate to convey its real applicability to prevent infections in humans.

Major points:

1. During the in vivo evaluation of the aMV@GEL safety, there were not significant differences on the overall cellular structure and respiration of the mice. However, the exposure occurred only once for a short time. Since this product is meant to be used several times during an extended period, it will be important to demonstrate that several applications of the aMV@GEL during different days/weeks would not lead to adverse effects like, for instance, damage of the epithelium, recruitment of inflammatory mediators, etc. In this line, the accumulation of the particles in vital organs was only assessed 8h after application of the aMV@GEL, could the authors present later time points like 24/48h post initial exposure?

Response: We appreciate the reviewer's suggestions. Accordingly, we have now intranasally administered aMV@GEL to mice every other day for a period of two weeks, and blood and major tissues have been collected at both day 7 and day 14 for analysis. As shown in Fig. R14A and R14B, the blood analysis of white blood cells (WBC), red blood cells (RBC), and platelets (PLT) as well as the serum biochemistry indices analysis of blood urea nitrogen (BUN), aspartate alanine aminotransferase (ALT), aminotransferase (AST), lactate dehydrogenase (LDH), and alkaline phosphatase (ALP) were all within the normal ranges. Furthermore, Fig. R14C demonstrated that the levels of cytokines (IFN- γ , TNF, IL-2 and IL-6) exhibited no significant difference between aMV@GEL-treated mice and healthy mice, suggesting that aMV@GEL did not induce an inflammatory response in the nasal cavity. In addition, the H&E staining results in Fig. R14D indicated that long-term use of aMV@GEL did not cause apparent damage to the nasal cavity, brain, lung, heart, liver, spleen, and kidney.

Fig. R14 (as Fig. S21 in revision) Biosafety evaluations upon multiple aMV@GEL administrations.

aMV@GEL was intranasally administered to mice every other day for a period of two weeks. At day 7 and day 14, the blood and the major tissues (including the nasal cavity, brain, lung, heart, liver, spleen, and kidney) were collected for biosafety evaluations.

(A) Blood analysis of white blood cells (WBC), red blood cell (RBC), and platelet (PLT) at day 7 and day 14 (n=3 biologically independent mice). The blue background presented the normal range.

(B) Serum biochemistry indices analysis of urea nitrogen (BUN), aspartate alanine aminotransferase (ALT), aminotransferase (AST), alkaline phosphatase (ALP), and lactate dehydrogenase (LDH) at day 7 and day 14 (n=3 biologically independent mice). The blue background presented the normal range.

(C) Inflammatory cytokines (IFN- γ , TNF, IL-2 and IL-6) analysis of the nasal homogenate at day 7 and day 14 (n=3 biologically independent mice).

(D) Representative H&E staining images of major tissues at day 7 and day 14.

Data in A, B and C represent the means \pm SEM. Statistical significance in C was calculated using one-way ANOVA with multiple comparison tests. n.s. meant no significant difference.

According to the reviewer's request, we have now included the assessment of particles accumulation in major tissues at 24 h and 48 h post the initial application of aMV@GEL. As shown in Fig. R15, small amount of free aMV was observed to remain in the nasal cavity at 8 h, while the majority entered the lung upon respiration. In sharp contrast, the embedded aMV in GEL (aMV@GEL group) exhibited a significantly different behavior, with the majority retained in the nasal cavity at 8 h and even persisted until 24 h, indicating that GEL prolonged the retention of aMV at the nasal cavity while substantially reduced the exposure to other organs.

Fig. R15 Representative *ex vivo* fluorescence images of various tissues (nasal cavity, brain, lung, heart, liver, spleen, and kidney) at 8 h, 24 h, and 48 h after intranasal administration of free aMV or aMV@GEL. aMV in both groups were stained with Cy5.

These results further strengthened the biosafety of our MV@GEL material upon multiple topical applications in the nasal cavity. Above biosafety data have now been added as Fig. S21, corresponding description and experimental method have also been added in revised version as follow.

Main text:

Page 8-9, line 236-239: In addition, the comprehensive evaluations, including blood routine analysis, serum biochemistry index analysis, nasal inflammatory cytokine detection, and histological analysis of major tissues, confirmed the safety upon multiple aMV@GEL administrations (Fig. S21).

Method:

Page 21, line 653-661: The biosafety evaluations of the long-term intranasal administration of aMV@GEL. The mice were intranasally administered with aMV@GEL every other day in a period of two weeks. The blood and the major tissues (including the nasal cavity, brain, lung, heart, liver, spleen, and kidney) were collected at both day 7 and day 14. The blood was used for blood routine (WBC, RBC and PLT) and biochemical indicators (BUN, ALT, AST, LDH and ALP) analysis by using automatic haemocytometer analyzer and biochemical analyzer, respectively. The inflammatory cytokines (IFN- γ , TNF, IL-2 and IL-6) of the nasal homogenate were detected by the corresponding ELISA kit. The major tissues were fixed and prepared into paraffin sections (3-4 μ m) for H&E staining.

2. Regarding the HRT model, while it is an innovative and interesting set up, several pitfalls need to be addressed to carry the author's message. The cellular characterization of the organoids needs to be re-done. The stainings presented are not convincing and appear to show the same cells (no cilia in the ciliated cell marker), instead of using 2D pictures would be better to present whole mount 3D pictures as done in the NP panel. Regarding the organoid infection, I would recommend to do more readouts than just the staining NP or viral RNA expression. The authors could include a cell viability marker, apoptosis measurements, analysis of other relevant inflammatory mediators, plaque assay analysis.

Response: We appreciate the reviewer's valuable suggestions. Upon HRT model, we have now re-done the cellular characterizations and provided 2D and corresponding 3D pictures. As shown in Fig. R16A-B, the lung organoid was labelled by the several specific markers, including FOXJ1 (ciliated cell marker), SCGB1A1 (club cell marker), P63 (basal cell marker), and SA (H1N1 receptor). Both 2D and 3D pictures showed lung organoid consist of cell types as expected.

Moreover, we carried out additional experiments with the updated HRT model (according to comment 3) to comprehensively evaluate the organoid infection. To evaluate the cell apoptosis, we observed the shape and cell arrangement of lung organoid by optical imaging and detected the phosphatidylserine reversion (AnnexinV-mCherry staining) by CLSM imaging. As shown in Fig. R16C, the organoid in PBS group lost the normal cell arrangement after infection by the virus, and the phosphatidylserine of some exfoliated cells has reversed. On the contrary, no sign of morphological changes and phosphatidylserine reversion was observed in the lung organoid of sMV@GEL group. Moreover, we utilized Live/Dead analysis for detection of cell viability. Supporting the data of apoptosis, the dead cell dominated the cell population in PBS group, while no dead cells were detected in sMV@GEL group (Fig. R16D).

Fig. R16 (Fig. R16A as Fig. 7C, Fig. R16B as Fig. S33, and Fig. R16C-D as Fig. S34A-B in revision) Cellular characterization of lung organoid and the viability of lung organoid in different groups. (A, B) Representative 2D CLSM images (A) and 3D reconstructed CLSM images (B) showing expression of FOXJ1 (ciliated cell marker), SCGB1A1 (club cell marker), P63 (basal cell marker), and SA (H1N1 receptor) in lung organoid. FOXJ1, SCGB1A1, and P63 were stained with corresponding antibodies and secondary Alexa Fluor® 647 fluorescent antibody (red), SA was stained with Cy3-SNL (red, false color), cytoskeleton was stained with FITC-phalloidine (white, false color), and cell nuclei were stained with DAPI (blue). (C) Cell apoptosis analysis of lung organoid in different groups after 48 h of challenge with H1N1-CA07 aerosols. The phosphatidylserine (red) was stained by Annexin V-mCherry. (D) Live/Dead analysis of lung organoid in different groups after 72 h of challenge with H1N1-CA07 aerosols. The dead cells were stained by Propidium iodide (red), and the live cells were stained by Calcein (green).

In the terms of inflammatory mediator detection and plaque assays, we did not find any literature using these two methods for investigating the viral infection to lung organoid. After discussion with our team members and experts, we thought these two methods might not be applicable. Owing to the lack of immune cells in the lung organoid, the production of inflammatory mediators was very low, which could be supported by the trace amount of TNF, IL-2 and IL-6 we detected (< 5 pg/ml). Given that the lung organoid was the independently dispersed spheres, it is too much difficult to form tightly connected layers, excluding the possibility of conducting plaque assays.

Overall, the above cellular characterizations have proved the lung organoid as a reliable model to simulate the viral infection in lung, and the apoptosis and viability assays have supported the efficacy of sMV@GEL in protecting the lung organoid against viral aerosol exposure in HRT model. Above results, corresponding description and relative method have now been added in our revised manuscript.

Main text:

Page 15, line 423-425: Consequently, lung organoid in the PBS group showed

substantially reduced cell viability, while lung organoid in the sMV@GEL group had no sign of either apoptosis or death (Fig. S34).

Method:

Page 25, line 840-843: To analyze the cell apoptosis and cell viability of lung organoids, the Annexin V-mCherry Apoptosis Detection Kit and the Calcein/PI Cell Viability/Cytotoxicity Assay Kit were used following the instructions, respectively.

3. In addition, the organoid model described reflects the airway epithelium of the proximal airways. Therefore, it will make more sense to have the cultures closer to the nasal apparatus. Another option will be to replace the current model for one that will better reflect the distal epithelium of the lung, this will be particular more relevant since influenza infection of the distal epithelium is a major

Response: We would like to take this opportunity to clarify the human respiratory tract (HRT) model in our study. A sterile tube was utilized to connect the nasal apparatus with the lung organoid, which ensured the consistent loading of viral aerosol in the air flow throughout the model. Therefore, the different positions of the organoid culture in the HRT model are unlikely to significantly influence the anti-infection performance of our sMV@GEL. For verification, we have now repeated the experiment by reducing the tube length to the half (Fig. R17A), leading to the organoid culture closer to the nasal apparatus. As expected, we obtained much similar infection results of 3D reconstructed images and the q-PCR analysis (Fig. R17B-D), which could also be supported by additionally provided data of cell apoptosis and cell viability (details in the response to comment 2). And we have now updated the data certifying the protective effect of sMV@GEL in Fig. 7.

Fig. R17 (Fig. R17A as Fig. 7B and Fig. R17B-D as Fig. 7D-F in revision) Confirmation of the protective

effect of sMV@GEL against H1N1 viral aerosols by using the updated human respiratory tract (HRT) model.

A. Schematic diagram of the experimental design. To imitate the HRT, we constructed an integrated HRT model by placing the lung organoids into a container with the following vents: one vent was connected to the nasal apparatus via a sterile tube, and the other was connected to a pump for providing respiratory airflow. The inside of the nasal apparatus was coated with PBS or sMV@GEL by spraying liquid PBS or sMV@GEL into the nostril. After placing the HRT model at 37 °C, the H1N1 aerosols were produced around the nostril of the human nasal apparatus and inhaled into the nasal apparatus by the pump under inhalation state. Finally, we detected the infection of the lung organoids to evaluate the protective effect of sMV@GEL.

B. Representative 3D reconstructed CLSM images of lung organoids in different groups after 24 h or 48 h of challenge with H1N1-CA07 aerosols. The cytoskeleton was stained by FITC-phalloidine (white, false color), cell nuclei were stained with DAPI (blue), and the N protein of H1N1-CA07 was immunofluorescently labeled with an N protein antibody and the corresponding secondary Alexa Fluor® 647 fluorescent antibody (green, false color).

C. Relative level of H1N1-CA07 mRNA in lung organoids of different groups after 24 h or 48 h of challenge with H1N1-CA07 aerosols (n=3 biologically independent experiments).

D. Relative level of H1N1-PR8 mRNA in lung organoids of different groups after 24 h or 48 h of challenge with H1N1-PR8 aerosols (n=3 biologically independent experiments).

Data in C and D represent the means ± SEM. Statistical significance was tested with unpaired t-test (C and D). ***P < 0.001.

We also appreciate the reviewer to provide another good option. We totally agree that comparing the protection effects for airway epithelium and distal epithelium is a very interesting topic. However, conducting these experiments require new ethical review, and it seems beyond the aim of this manuscript. Actually, we are developing an iterative modality for trapping the viral aerosol in the respiratory tract, and we will keep the reviewer's good suggestion in mind for this study in the future.

4. Regarding the product applicability, one major point raised by the authors in the conclusions is that health workers could use the intranasal mask as an additional method for protection against viral infections. Could the authors evaluate if the use of the intranasal mask plus the regular mask compared to only use one or the other really adds up to a higher protection against viral infections?

Response: We thank the reviewer for promoting us to conduct additional experiments with our HRT model to evaluate the performance of using the intranasal mask plus the regular mask (surgical mask). Specifically, we included four groups for comparison: PBS, mask, intranasal mask (sMV@GEL), combined masks (mask+sMV@GEL). To simulate real-life scenarios where masks may not be worn correctly, we introduced a "loose wearing" condition by creating a gap (1-3 mm) between the mask and the nasal cavity model.

As shown in Fig. R18, compared with the mask showing low level of viral loading, combined masks further reduced the viral loading approximating to an undetectable level, indicating an almost complete interception performance. These data strengthened the applicability and effectiveness of the proposed intranasal mask as an additional

protective measure against viral infections, particularly for healthcare workers and individuals in high-risk settings.

Fig. R18 (as Fig. S36 in revision) Relative level of H1N1-CA07 mRNA of lung organoids in different groups after 48 h of challenge with H1N1-CA07 aerosols (n=3 biologically independent experiments). Data represent the means \pm SEM. Statistical significance was using one-way ANOVA with multiple comparison tests. **P < 0.01.

Above data have now been added as Fig S36, and corresponding description and experimental method have been added in revised version as follow.

Main text:

Page 15, line 441-444: Moreover, aiming at high-risk individuals, such as doctors and nurses, our intranasal mask could also be combined with face masks to further reduce the risk of infection from aerosols containing threatening viruses (Fig. S36), such as SARS-CoV-2, H1N1, and SARS.

Method:

Page 25-26, line 844-851: To evaluate the performance of using the intranasal mask plus the regular mask, four groups including PBS, mask, intranasal mask (sMV@GEL), and combined masks (mask+sMV@GEL) were included. Simulating real-life scenarios where masks may not be worn correctly, we introduced a "loose wearing" condition by creating a gap (1-3 mm) between the mask and the nasal module. For the combination group of "mask+sMV@GEL", the sMV@GEL was applied in nasal apparatus, and then the surgical mask was worn on the nasal apparatus in "loose wearing" way. The infection experiment was conducted following the previous mentioned HRT infection method (2.4×10^5 PFU, 30 min).

Minor points:

5. The use of color in the background of some of the graphs is a bit distracting and do not follow a pattern. Maybe be better to remove.

Response: According to the reviewer's suggestion, we have now removed the background colors from the graphs (Fig. 2I, Fig. 2J, Fig. 4G-4I, Fig. 5J and Fig.7 E-F) to ensure a consistent and uncluttered visual presentation. Just list a few examples as below.

Fig. R19 Examples of the revised graphs (Fig. R19A as Fig. 4I, Fig. R19B as Fig. 7E).

Reply to reviewer's comments:

Reviewer #4:

For this paper I was asked to review the following: “we would specifically require input regarding the computational aspects (Figure 6) from you, as we have the other aspects featured in this study covered by other referees”.

After reading the manuscript I find the computational methodology and the numerical simulations appropriate and useful. However, I would like to ask the authors extend a little bit the discussion and clarify the following points/comments.

1. According to Koullapis et al. (2018) “the flow in the upper airways is mostly turbulent and/or transitional in nature, even at low inhalation rates”. However, it seems that the performed simulations have been for laminar conditions. Therefore, the authors should convincingly argue why they decided to perform laminar flow simulations. Which was the flow Reynolds number at inlet and outlet of the computational domain?

Response: Thanks for reviewer’s careful review. We firstly take this opportunity to clarify the issue of flow Reynolds number. The inlet velocity was a function with time: $u_{inlet}=0.64\sin(2\pi/3\times t)$, corresponding to 10 L/min. Taking the phenomenon with maximum velocity ($u_{inlet, max}=0.64$ m/s, $t=0.75$ s) into account, the Reynolds number was 734 at inlet, and 1261 at outlet. Above range of Reynolds number provided us an initial reference to use laminar solver for aerosol simulation, since researchers has reached the agreement that laminar solver is applicable upon the situation of Reynolds number below 2300.

We also thank the reviewer for providing us the reference, which reported that “the flow in the upper airways is mostly turbulent and/or transitional in nature, even at low inhalation rates”. Note that this study investigated the upper airways across nasal cavity, trachea and primary bronchi, and high-speed profile mainly occurred at the conducting zone (trachea or primary bronchi). On the contrary, we herein focused on the nasal cavity alone, which did not provide enough channels for air to induce notable turbulence effect upon normal breath (inhalation rate: 10 L/min). Supporting our explanation, the simulation of airway in nasal cavity conducted by H. Shi et al¹. demonstrated that the turbulent kinetic energy was very low and turbulent airflow effect was possibly negligible for normal, equal nasal inhalation (i.e., $Q_{max}\leq 20$ L/min).

Although turbulence effect was not notable in the nasal cavity, to eliminate the reviewer’s concern, we have now included $k-\omega$ turbulence model to simulate the air and particle flow under the unprotected condition. Note that all settings are completely consistent with the original laminar case, except the $k-\omega$ turbulence model being added. The results could be summarized into following two aspects.

(1) Transit velocity distribution

Fig. R20 showed transit velocity distribution at central YZ plane using different models. At different time, the velocity distributions calculated by laminar model and by $k-\omega$ turbulence model were much similar. Fig. R21 showed transit average velocity in y direction at different sections, which had the same positions as Fig. 6 in our manuscript. Sinusoidal distribution was presented from velocity curve, unifying with

the inlet velocity law of sine distribution with time. In plane 3, 4 and 5, two average velocity curves almost coincided. In plane 1 and 2, velocity computed by $k-\omega$ turbulence model was slightly higher than that obtained by the laminar model, with the maximum relative error only being 4.1% and 7.6%, respectively. These data thus demonstrated the addition of turbulence model had neglectable impact on the flow field.

Fig. R20 Transit velocity distribution at central plane under two models.

Fig. R21 Transit average velocity in y direction at different sections.

(2) Transit particle dynamics

We have also countered and compared the particle number in nasal cavity with laminar model and $k-\omega$ turbulence model (Fig. R22). The particles were allowed to flow out of throat in the inhalation stage and out of nostrils in the exhalation stage. During the whole inhalation process (0-1.5 s) and early stage of exhalation (1.5-2.0 s), laminar model and $k-\omega$ model showed the numbers of particles with almost the same values. During the later stage of exhalation (2.0-3.0 s), the particle numbers calculated by $k-\omega$ turbulence model were slightly higher than those calculated by laminar model, with average relative error only being 8.2 %. Therefore, these data demonstrated the addition of turbulence model had neglectable impact on the particle dynamics.

Fig. R22 Changes of particle number in nasal cavity with time under different models.

The relatively low Reynolds number, the focus of nasal cavity, and the neglectable impact of additional turbulence model together supported our rationale of choosing laminar solver for the simulation of air flow in the nasal cavity. Some explanations about using laminar solver have now been added in the manuscript as follow.

Main text:

Page 24, line 779-782: Taking the maximum inlet velocity ($u_{inlet,max}=0.64$ m/s, $t=0.75$ s) into account, the Reynolds number was 734 at inlet, and 1261 at outlet. Considering the relatively low Reynolds number and the focus of nasal cavity, we proposed that the situation can be approximately regarded as in laminar region.

Reference in response:

1. H. Shi, C. Kleinstreuer & Z. Zhang, Dilute suspension flow with nanoparticle deposition in a representative nasal airway model, *Physics of Fluids* **20**, 1 (2008).

2. Repeatability of the numerical results. Authors include only 18000 particles for the total simulation time, which are very few. I would like to know if the simulations have been repeated several times under identical conditions (except for a random location of the tracked particles) and if the same results were obtained.

Response: Thanks for the reviewer’s comments. Firstly, we would like to take this opportunity to clarify the rationale of particles numbers for simulation. Previous study reported the emissivity of viral aerosol could be set in the range of 1000-3000 particles per second for normal condition². To mimic the high-risk environment, we herein emitted 6000 particles per second.

In the aspect of repeatability, the solver was compiled into executable file, and the same particle emission law was ensured in each simulation (pseudo random). Therefore, the results were fully consistent when simulation was repeated several times upon unchanged settings. For verification, we repeated the simulation with a different hardware. Taking the MV@GEL situation for example (Fig. R23), we obtained the same results, including the distribution of inhaled viral aerosols (blue dot) in different cross sections and percentage of viral aerosols in nasal cavity at $t=1.5$ s.

Fig. R23 The distribution of inhaled viral aerosols (blue dot) of MV@GEL situation (left) and the corresponding percentage of viral aerosols that retained in the nasal cavity or flowed into trachea at $t=1.5$ s (right) upon different hardware.

To further eliminate the reviewer’s concern, we emitted the particles with random location at the inlet (nostril), while other settings were identical to repeat the simulation three times. Again, taking the MV@GEL situation for example, the overall distributions of particles in the nasal cavity were apparently same in three simulations with random emission (Fig. R24). After quantification, we found a very tiny difference of particles retained in nasal cavity (original case: 93.20%; random emission: 92.94%-93.33%). Although random emission might lead to $\pm 0.2\%$ variation of above proportion value among repeated simulations, such a slight variation would not compromise the repeatability and convincingness of our simulation data.

Fig. R24 The distribution of inhaled viral aerosols (blue dot) of MV@GEL situation (left) and the corresponding percentage of viral aerosols that retained in the nasal cavity or flowed into trachea at $t=1.5$ s (right) upon random emitting.

Reference in response:

2. Cheng, Y.F., et al. Face masks effectively limit the probability of SARS-CoV-2 transmission. *Science* 372, 1439-+ (2021).

3. Authors should explain why some phenomena as Brownian motion, Saffman lift forces or buoyancy effects have not been considered in the simulations.

Response: We thank the reviewer for reminding us this issue. In the terms of Brownian motion and Saffman lift, most studies focusing on micron-sized particles in the upper airways have also discounted these two factors, since their magnitudes are much smaller than those of drag and gravitational force⁴⁻⁶. Taking this opportunity, we herein further explain the reason for the very low magnitudes of these two factors in our simulation system.

For Brownian motion, it has been well accepted that “Brownian motion was shown to be the dominant mechanism for diffusion of particles smaller than 0.1 μm . For particles larger than 0.5 μm , the effect of Brownian diffusion was negligibly small”⁷. As the particle diameter we simulated was 5 μm , the effect of Brownian diffusion could be considered negligible. For Saffman lift force, it has been considered exerting effect for large particles and small particle-to-fluid density in shear flow^{8,9}. In our simulation, however, the small particles (5 μm) with large particle-to-fluid density (1000/1.2) were utilized to mimic the viral aerosol environment.

In the term of buoyancy effect, we apologize for our careless of omitting the corresponding factor in the equation. Actually, the buoyancy effect had been taken into consideration, and we had used the equivalent gravity with the form of $F_g = m_p \mathbf{g} (\rho_p - \rho_f) / \rho_p$. We have now added the buoyancy in the revised Fig. 6B (also shown as Fig. 25) and the governing equation for particles (page 24, line 784) has also been replaced as follow.

Fig. R25 (Fig. 6B in revision) Schematic diagram of the charge interaction between MV@GEL and aerosols in the nasal cavity during simulation calculation. Each tiny part on the nasal wall could generate a tiny coulomb force to the specific viral aerosol, the drag force of fluid, buoyancy, and gravity were considered, and the vector sum was the resultant force.

$$m_p \frac{d^2 \mathbf{x}_p}{dt^2} = m_p \frac{d\mathbf{u}_p}{dt} = \mathbf{F}_d + \mathbf{F}_e + m_p \mathbf{g} \frac{\rho_p - \rho_f}{\rho_p}$$

Reference in response:

3. Matida, E.A., Finlay, W.H., Lange, C.F. & Grgic, B. Improved numerical simulation of aerosol deposition in an idealized mouth–throat. *J Aerosol Sci* **35**, 1-19 (2004).
4. Jayaraju, S.T., Brouns, M., Verbanck, S. & Lacor, C. Fluid flow and particle deposition analysis in a realistic extrathoracic airway model using unstructured grids. *J*

Aerosol Sci **38**, 494-508 (2007).

5. Dehbi, A. Prediction of Extrathoracic Aerosol Deposition using RANS-Random Walk and LES Approaches. *Aerosol Sci Technol* **45**, 555-569 (2011).

6. Lambert, A.R., O'Shaughnessy, P., Tawhai, M.H., Hoffman, E.A. & Lin, C.L. Regional deposition of particles in an image-based airway model: large-eddy simulation and left-right lung ventilation asymmetry. *Aerosol Sci Technol* **45**, 11-25 (2011).

7. Koullapis, P., et al. Regional aerosol deposition in the human airways: The SimInhale benchmark case and a critical assessment of in silico methods. *Eur J Pharm Sci* **113**, 77-94 (2018).

8. Kallio, G.A. & Reeks, M.W. A numerical-simulation of particle deposition in turbulent boundary-layers. *International Journal of Multiphase Flow* **15**, 433-446 (1989).

9. Young, J. & Leeming, A. A theory of particle deposition in turbulent pipe flow. *Journal of Fluid Mechanics* **340**, 129-159 (1997).

4. Which were the exact boundary conditions in the exhalation phase?

Response: The boundary condition in the exhalation phase is exactly the same as the boundary in inhalation stage. The inlet boundary condition was *codedFixedValue* set at nostril: $u_{inlet} = 0.64 \sin(2\pi/3 \times t)$. It meant that in inhalation stage (0-1.5 s), the inlet velocity was a positive value, representing the air was drawn into the nasal cavity from atmosphere, while inlet velocity was a negative value when t was in range of 1.5-3 s, which represented that air flowed out of the atmosphere from the nasal cavity. The outlet boundary condition was *pressureInletOutletVelocity* at pharynx, which was a kind of zero gradient boundary in OpenFOAM, and air was allowed to flow in and out from outlet.

In the revised manuscript, we have now emphasized the same boundary condition for both inhalation stage and exhalation stage as follow.

Method:

Page 24-25, line 805-809: “The same boundary conditions were used in the inhalation and exhalation processes. The transient velocity boundary was imposed at the nostrils to simulate a complete respiratory cycle, with $u_{inlet}=0.64\sin(2\pi/3 \times t)$. *pressureInletOutletVelocity*, which is a kind of zero gradient boundary in OpenFOAM, was used at the throat for gas flow into and out of nasal cavity freely.”

5. “one order Euler scheme” should be “first order Euler scheme”.

Response: Thanks for the reviewer’s careful review. The “one order Euler scheme” has now been replaced with “first order Euler scheme” in our revised manuscript as follow.

Method:

Page 24, line 802-804: “The PIMPLE algorithm was used to solve the partial differential equation, with second-order scheme for spatial discretization and first order Euler scheme in time integration.”

6. Which was the density of aerosol particles?

Response: According to previous study, the aerosol particles were regarded as monodisperse and spherical particles with the density of liquid water (1000 kg/m^3)¹⁰, we have now described it more clearly in our revised manuscript as follow.

Mian text:

Page 12, line 354-357: The discrete viral aerosols were regarded as monodisperse and spherical particles with the density of liquid water (1000 kg/m^3), which were added into the nasal cavity at 6,000 per second in a period of inhalation with the same velocity as that of the airflow.

Reference in response:

10. Ma, B. & Lutchen, K. CFD Simulation of Aerosol Deposition in an Anatomically Based Human Large-Medium Airway Model. *Ann Biomed Eng* **37**, 271-285 (2009)

Reviewer #5 (Remarks to the Author):

This manuscript submitted to Nature Communications lies in the context of the prevention of aerosol-transmitted viral diseases. The authors developed an original formulation intended for intranasal administration and acting as an “intranasal mask”. It consists of a chitosan-based thermosensitive gel containing cell-derived microvesicles with specific receptors to capture and immobilize the virus within the gel in the nasal cavity, thus preventing it from reaching the lungs. In this study, the authors combined a full set of *in vitro* experiments, *in vivo* evaluation on mouse models, numerical simulations, and the development of an apparatus mimicking the human respiratory tract to establish the proof of concept on different variants of SARS-CoV-2 and influenza A. The experiments were generally well-designed and conducted. I share the main conclusions of the authors. In my opinion, these complementary approaches have led to very encouraging and promising results that deserve publication and dissemination to a large audience within the scientific community, especially in the current pandemic context. Indeed, the proposed strategy could significantly improve the prevention of these infectious diseases and have a high impact on public health. However, I think the authors should add some essential missing experimental details and clarify some aspects before publication in Nature Communications.

(1) Question1: Can the authors obtain batches of microvesicles overexpressing ACE2 or α -2,6 sialic acid receptors with reproducible characteristics?

Response: We appreciate the reviewer for reminding us this issue. Given that the engineered 293T cell line was stably transfected with viral receptors, the microvesicles derived from those cells could overexpress corresponding viral receptors in reproducible way. In our preliminary experiment of aMV and sMV preparation, we had investigated the reproducibility of different batches of aMV or sMV. As shown in Fig. R26A, three different batches of aMV exhibited similar size distribution and ACE2 expression level, contributing to the consistent anti-viral effects against SPV. Similar to aMV, the sMV showed the good reproducibility in terms of size distribution, SA expression, and anti-viral effects against H1N1 (Fig. R26B).

Fig. R26 (the left two panels of Fig. R26A as Fig. S3 in revision, the left two panels of Fig. R26B as Fig. S26 in revision) The reproducible characteristics of aMV and sMV.

(A) Average size, ACE2 expression level and anti-SPV effect of aMV in three different batches (n=3 biologically independent experiments).

(B) Average size, SA expression level and anti-H1N1 effect of sMV (B) in three different batches (n=3 biologically independent experiments).

Data in A and B represent the means \pm SEM. Statistical significance in A and B were calculated using one-way ANOVA with multiple comparison tests. n.s. meant no significant difference.

Above reproducibility results of average size and viral receptor expression, corresponding description and caption have now been added in our revised manuscript as follow.

Main text:

Page 5, line 134-136: Additionally, the aMV showed reproducible characteristics of average size and ACE2 expression (Fig. S3), paving the way for subsequent interaction and entrapment of SARS-CoV-2.

Page 10, line 289-291: Upon treating the cells with cytochalasin B, we observed the expected formation of MV on the cell surface (Fig. 5C), which we separated to generate sMV in a reproducible way (Fig. 5D, Fig. S25 and Fig. S26).

Question2: Are these microvesicles stable in the hydrogel? For how long? Are they dispersed in a reproducible way in the hydrogel?

Response: We thank the reviewer for reminding us these issues. We herein first clarify that the MV were stable in the hydrogel network. For verification, we have now used CLSM imaging to monitor the vesicle structure of DiD-labeled MV after dispersing in the hydrogel at 37 °C. As shown in Fig. R27, MV remained the vesicle structure and micro size during 24 h observation. Such a good stability of MV in GEL provide sufficient time window for exerting the entrapment function in the nasal cavity.

Fig. R27 (as Fig. S11 in revision) CLSM image showing the aMV's vesicle structure in hydrogel at 24 h post gelation. The aMV was stained by DiD (red), the GEL was stained by FITC (cyan, false color).

In the aspect of reproducible dispersion, we herein make some explanation of application guide. The MV and liquid hydrogel could be stored in separate containers at 4 °C for a long time and be simply mixed before use. In this case, the MV was easily to disperse in the liquid hydrogel. If the MV@GEL was not used up, the MV@GEL should be putted back in the refrigerator as soon as possible, and be simply shaken for 10 times to redisperse the MV before the next using. To provide experimental evidence, the DiD-labeled MV was mixed with liquid hydrogel by shaking 10 times, and then the liquid MV@GEL was stored in refrigerator for one day, and then was taken out and shaken for 10 times. As shown in Fig. R28, the MV exhibited good redispersion property in the liquid state hydrogel.

Fig. R28 (as Fig. S10 in revision) CLSM images of aMV@GEL upon initial mixture of aMV and GEL (A) and upon redispersion after 24 h storage in refrigerator (B). The aMV was initially mixed with liquid hydrogel by shaking 10 times. The liquid aMV@GEL was then stored in the refrigerator for 24 hours, followed by shaking for 10 times to redisperse the aMV in the liquid GEL. The aMV was stained by DiD (red).

We have now added the above stability, dispersion and redispersion data in our revised manuscript, accompanying with corresponding description as follow.

Main text:

Page 7, line 191-193: Then, GEL was mixed with aMV to generate aMV@GEL (Fig. 3A), wherein, as shown in the CLSM image (Fig. S10, Fig. S11 and Fig. 3B), DiD-stained aMV was well dispersed in FITC-stained GEL and remained the micro-sized vesicle structure.

Question3: Do these vesicles retain their integrity and efficacy when they experience high shear rates in the spraying device or when the formulation dries in air after deposition on the surface or after intranasal administration?

Response: We thank the reviewer for drawing this point to our attention. To address this concern, we have now conducted additional experiment to investigate the

vesicles' integrity and efficacy when they experience high shear rates in the spraying device.

First, we utilized CLSM to detect the integrity of the DiD-labeled MV in the FITC-labeled hydrogel after spray. As shown in Fig. R29A, the MV within the hydrogel maintained their vesicle structure and micro size, indicating that their integrity was well retained after experiencing spray-induced shear rates.

Using Transwell model, we continued to evaluate the antiviral efficacy of aMV@GEL after drop or spray, with GEL (drop) and GEL (spray) as controls. Then we monitored the expression of green fluorescent protein (GFP), which SPV can express upon entry into the host cells (ACE2-293T) cultured in the bottom chamber. As shown in Fig. R29B, compared with the obvious SPV infection (green) in the GEL (drop) and GEL (spray) groups, a substantially reduced SPV infection was observed in both the aMV@GEL (drop) and aMV@GEL (spray) groups. Note that the reduction showed no significance between aMV@GEL (drop) and aMV@GEL (spray) groups (Fig. R29C), indicating that the microvesicles in the GEL retained their antiviral efficacy even after experiencing spray-induced shear rates.

Fig. R29 (Fig. R29A as Fig. S14 and Fig. R29B-C as Fig. S15 in revision) Integrity and anti-viral efficacy of aMV@GEL after experiencing spray-induced shear rates.

- (A) CLSM image showing the aMV remained vesicle structure in hydrogel after experiencing spray-induced shear rates. aMV were labeled with DiD (red), the GEL was stained with FITC (cyan, false color).
- (B) Schematic illustration of a Transwell™ model for evaluating the infection rate of SPV (wild-type) challenged ACE2-293T cells (in the bottom chamber) with GEL (drop), GEL (spray), aMV@GEL (drop) or aMV@GEL (spray) treatment (in upper chamber) and the corresponding representative CLSM images of ACE2-293T cells. The infected cells expressed GFP protein (green), and the nuclei were stained with DPAI (blue).
- (C) Relative infection rates of ACE2-293T cells in the GEL (drop), GEL (spray), aMV@GEL (drop) and aMV@GEL (spray) groups (n=3 biologically independent experiments). Data represent the means ± SEM. Statistical significance was calculated using one-way ANOVA with multiple comparison tests. ****P < 0.0001, n.s. meant no significant difference.

To eliminate the reviewer's concern of formulation drying in air after deposition on the surface or after intranasal administration, we'd like to take this opportunity to make some clarifications. The chitosan hydrogel in our intranasal mask possesses moisturizing properties^{1,2}, which effectively prevent rapid drying in the nasal cavity. Additionally, the nasal cavity's naturally moist environment, with relative humidity ranging from 60% to 80%³, aids in maintenance of the gel's moisture content. Above property paved the way for the protection performance of MV@GEL within the long-

term viral aerosol exposure, which has been evidenced in our H1N1 virus transmission model (see detail in the response to comment 4 of reviewer 1). The mouse was administrated with sMV@GEL in nasal cavity every 8 hours, and inhaled the H1N1 viral aerosols which produced by infected mouse for 3 days. The sMV@GEL protected mouse remained uninfected at day 3. This sustained protective effect of sMV@GEL in the nasal cavity further suggested the sMV@GEL did not readily dry out in the nasal cavity.

We believe above data and explanations could eliminate the reviewer's concerns about the integrity and efficacy of MV after experiencing spray-induced shear rates and their effect after prolonged retention in the nasal cavity. We have now added the above data, description and method in our revised manuscript as follow.

Main text:

Page 7, line 197-199: Additionally, the integrity and anti-viral efficacy of aMV embedded in GEL was not influenced after experiencing spray-induced shear rates (Fig. S14 and Fig. S15), ensuring the entrapment function upon spray in the nasal cavity.

Method:

Page 22, line 692-696: To evaluated the anti-viral efficacy of aMV@GEL experiencing dropping or spraying, we added the GEL (drop), GEL (spray), aMV@GEL (drop) and aMV@GEL (spray) in the upper chamber of Transwell™, respectively. Then the infection experiment and the infection rate detection of the ACE2-293T cells in lower chamber was conducted following the aforementioned method.

Reference in response:

1. Kim, K., Kim, K., Ryu, J.H. & Lee, H. Chitosan-catechol: a polymer with long-lasting mucoadhesive properties. *Biomaterials* **52**, 161-170 (2015).
2. Chaiwong, N., et al. Antioxidant and moisturizing properties of carboxymethyl chitosan with different molecular weights. *Polymers (Basel)* **12**, 7 (2020).
3. Keck, T., Leiacker, R., Heinrich, A., Kuhnemann, S. & Rettinger, G. Humidity and temperature profile in the nasal cavity. *Rhinology* **38**, 167-171 (2000).

2) The hydrogel spraying ability was studied, and a spraying method was used in some experiments but not systematically. The authors should state more clearly if this method of administration is foreseen or not for future development. In this respect, the characteristics and quality of the coating of the nasal surface or material mimicking the nasal surface are critical for the efficacy of the hydrogel to entrap the virus. In my opinion, a weakness of this study is the incomplete description of this coating in the different experiments. The authors should clarify or discuss the following points:

Response: We appreciate the reviewers' valuable comments and suggestions regarding the hydrogel spraying and the coating of the nasal surface. We have now conducted the additional experiments to address reviewer's concerns and enhance the clarity and completeness of the provided information regarding the coating and method of administration. Please see more details in the following responses.

Question1: - What is the size distribution of the droplets obtained when spraying the

MV@GEL or other control systems?

Response: We have now conducted experiments to measure the size distribution of droplets when spraying PBS, aMV, GEL, and aMV@GEL. Briefly, the droplets were sprayed onto a hydrophobic film, and the droplet size was subsequently determined by microscope imaging and Image J analysis. As shown in Fig. R30, the average droplet size and the size distributions for PBS, MV, GEL, and MV@GEL were 72.43 μm , 95.92 μm , 169.7 μm , 220.5 μm , respectively. The variation in particle size and size distribution among the different groups should be primarily attributed to differences of solution viscosity. Note that the droplets would fuse with each other for deposition, we envisioned that the moderately increased droplet size of MV@GEL might not substantially impact the final formation of the protective layer in the nasal cavity.

Fig. R30 Size distribution of the droplets when spraying PBS, MV, GEL or MV@GEL.

Question2: - In some experiments, the method of coating needs to be detailed (for instance, the coating of the tube with PBS, aMV, GEL, or aMV@GEL for *in vitro* study of SPV aerosol interception). What area is covered by the hydrogel in the different types of experiments?

Response: We appreciate the reviewer for reminding us to provide these details. In the *in vitro* study of SPV aerosol interception, the tube was coated by slowly dripping the cold solution (PBS, aMV, GEL, or MV@GEL) onto the inside surface of the tube. During this process, the tube was rotated slowly to ensure uniform coverage of the coating. At the end, we observed that the entire inner surface of tube was coated by solution. Therefore, coating area (50.1 cm^2) is consistent with the inner surface.

For the nasal cavity of mice, given the small structure of the mice nasal cavity, the coating procedure was performed using the nasal drip, which was commonly used in studies related to gel nasal administration⁴. The cold solution (PBS, MV, GEL or MV@GEL) with identical volume was administered slowly into the nasal cavity of the mouse to prevent leakage, which could ensure the consistence of the coating layer for each mouse. In the HRT model, the cold solution (PBS or MV@GEL) with identical volume was sprayed into nasal cavity with a fixed position (0.5 cm inside the nasal cavity) and fixed direction (45°), which could also ensure the consistence of the coating layer for each test. Given that the structure of nasal cavity was very complex, we hope reviewer can understand the difficulty of calculating the area covered by the hydrogel. We have now supplemented additional experimental details in our revised manuscript as follow.

Method:

Page 20, line 610-613: For investigating the SPV aerosols interception effect of CS_{0.10}, CS_{0.12}, and CS_{0.14}, the gel layer was firstly formed on the inner wall of a tube (length:

16 cm, diameter: 10 mm). During this process, the tube was slowly rotated to ensure uniform coverage of the coating (coating area 50.1 cm², coating thickness 60-80 μm).

Page 21, line 671-674: To detect the interception effect of PBS, aMV, GEL, and aMV@GEL, the inner wall of the paper tube (length: 16 cm, diameter 10 mm) was coated by a layer of those agents, respectively. During this process, the tube was slowly rotated to ensure uniform coverage of the coating (coating area 50.1 cm², gel coating thickness 60-80 μm).

Page 22, line 700-702: In detail, the cold solution (aMV or aMV@GEL) with identical volume was administered slowly into the nasal cavity of the mouse to prevent leakage. The coating thickness of aMV@GEL ranged from 60 μm to 120 μm.

Page 22, line 712-713: The transgenic mice with high human ACE2 expression were treated with PBS, aMV, GEL, and aMV@GEL by intranasal administration (nasal drip), respectively.

Page 23, line 746-747: The BALB/c mice were treated with PBS or sMV@GEL by aforementioned intranasal administration (nasal drip), respectively.

Page 25, line 833-835: The cold solution (PBS or sMV@GEL) with identical volume was spray in to nasal cavity with a fixed position (0.5 cm inside the nasal cavity) and fixed direction (45°), which could ensure the consistence of the coating layer (thickness 60-150 μm) for each test.

Reference in response:

4. Wu, J., Wei, W., Wang, L.Y., Su, Z.G. & Ma, G.H. A thermosensitive hydrogel based on quaternized chitosan and poly (ethylene glycol) for nasal drug delivery system. *Biomaterials* **28**, 2220-2232 (2007).

Question3: - What is the layer thickness?

Response: During our investigation, we had realized the potential impact of the layer thickness on the protection performance. Therefore, different experimental setups had been adopted with matched volume of MV@GEL, ensuring the layer thickness distribution varying slightly. For verification, we have now conducted additional experiments by using SEM to evaluate the thickness of different layers coating on the tube, mouse nasal cavity and HRT models. Here, we provided the representative SEM images of MV@GEL layer on above models in Fig. R31. Upon sampling at different positions, we found the thickness distributions of the gel layer in the tube, mouse nasal cavity and HRT model were 60-80 μm, 60-120 μm, and 60-150 μm, respectively. Such a slight variation of layer thickness among different experiment thus strengthened the convincingness of protection data obtained by the tube-mouse-HRT deduction. We have now added more descriptions in method of our revised manuscript.

Method:

Page 20, line 610-613: For investigating the SPV aerosols interception effect of CS_{0.10}, CS_{0.12}, and CS_{0.14}, the gel layer was firstly formed on the inner wall of a tube (length: 16 cm, diameter: 10 mm). During this process, the tube was slowly rotated to ensure uniform coverage of the coating (coating area 50.1 cm², coating thickness 60-80 μm).

Page 21, line 671-674: To detect the interception effect of PBS, aMV, GEL, and

aMV@GEL, the inner wall of the paper tube (length: 16 cm, diameter 10 mm) was coated by a layer of those agents, respectively. During this process, the tube was slowly rotated to ensure uniform coverage of the coating (coating area 50.1 cm², gel coating thickness 60-80 μm).

Page 22, line 700-702: In detail, the cold solution (aMV or aMV@GEL) with identical volume was administered slowly into the nasal cavity of the mouse to prevent leakage. The coating thickness of aMV@GEL ranged from 60 μm to 120 μm.

Page 22, line 712-713: The transgenic mice with high human ACE2 expression were treated with PBS, aMV, GEL, and aMV@GEL by intranasal administration (nasal drip), respectively.

Page 23, line 746-747: The BALB/c mice were treated with PBS or sMV@GEL by aforementioned intranasal administration (nasal drip), respectively.

Page 25, line 833-835: The cold solution (PBS or sMV@GEL) with identical volume was spray in to nasal cavity with a fixed position (0.5 cm inside the nasal cavity) and fixed direction (45°), which could ensure the consistence of the coating layer (thickness 60-150 μm) for each test.

Fig. R31 Representative SEM images of MV@GEL layer in different models (A-tube, B-mouse nasal cavity, C-HRT). The yellow arrows showed the thickness at the indicated position of MV@GEL layer.

Question4: - Could the authors comment on the wettability and adhesion properties, as these properties are also crucial for good surface coverage?

Response: We thank the reviewer for drawing these two points to our attention. One notable feature of our intranasal mask is hydrogel, which is well-known for its favorable wettability properties due to its high-water content^{1,2}. This characteristic enabled the liquid-state hydrogel to well spread within the nasal cavity, ensuring comprehensive coverage and effective contact on the wettish wall of nasal cavity. As the main component of hydrogel, chitosan with abundant -NH₂ and -OH resulted in effective binding with the nasal epithelium cells⁵. This property enhanced the adhesion of our intranasal mask, allowing a stable residence within the nasal cavity.

In the revised manuscript, we have now emphasized the good wettability and adhesion properties of the intranasal mask/GEL as follow.

Main text:

Page 6, line 176-178: We had experience working with thermosensitive hydrogels, and in the present study, we explored the use of chitosan and β-sodium glycerophosphate to prepare irreversibly thermosensitive hydrogels with good wettability and adhesion property^{35,36}.

Reference:

- 35. Kim, K., Kim, K., Ryu, J.H. & Lee, H. Chitosan-catechol: a polymer with long-lasting mucoadhesive properties. *Biomaterials* **52**, 161-170 (2015).
- 36. Yang, J., Bai, R. & Suo, Z. Topological adhesion of wet materials. *Adv Mater* **30**, e1800671 (2018).

Reference in response:

1. Kim, K., Kim, K., Ryu, J.H. & Lee, H. Chitosan-catechol: a polymer with long-lasting mucoadhesive properties. *Biomaterials* **52**, 161-170 (2015).
2. Chaiwong, N., et al. Antioxidant and moisturizing properties of carboxymethyl chitosan with different molecular weights. *Polymers (Basel)* **12**, 7(2020).
5. Yang, J., Bai, R. & Suo, Z. Topological adhesion of wet materials. *Adv Mater* **30**, e1800671 (2018).

Question5: - What is the surface Zeta potential or surface charge of the coating? They have an impact on the electrostatic interactions between the virus droplets and the surface.

Response: To address this issue, we test the surface Zeta potential of MV@GEL coating using Solid Surface Zeta Potential Analyzer, and the value was 10.1 mV. The positive charged MV@GEL could intercept the negatively charged viral aerosol by electrostatic interactions. We herein also invite the reviewer to note that charge density had been considered during our investigation, as we had demonstrated the disorganized tight junction of epithelium cell for the formula containing excessive chitosan (see details in the response to question 12 in page 60-61). We have now added the surface Zeta potential value and corresponding method in our revised manuscript as follow.

Main text:

Page 7, line 193-197: Compared with the GEL (Fig. S12), aMV@GEL maintained many similar features in terms of its porous structure (pore size approximately 10-20 μm , Fig. 3C), thermosensitive gelling property (360 s at 37 °C, Fig. 3D and Fig. S13), spray characteristic (spray area 2.3 cm^2 , 10 cm away from the nozzle, Fig. 3E), and positive charge (10.1 mV).

Method:

Page 21, line 668-669: The surface Zeta potential of aMV@GEL was tested by Solid Surface Zeta Potential Analyzer (Attract2.0).

Question 6: - How is performed the intranasal administration of the formulations in mice?

Response: As aforementioned, given the small structure of the mouse's nasal cavity, we used nasal drip to perform the intranasal administration of the formulations in mice. Details can be seen in our response to the question 2 of this comment.

3) After virus interception by the gel, essential points are the ratio of virus that interacts with the receptors and is entrapped in the microvesicles within the gel and the ratio of virus that remains “free” in the gel. Depending on the gel thickness (not provided) and on the mobility of the virus within the gel, this free virus may be able to reach and infect

the underlying epithelial cells of the nasal cavity, contrary to the assertion of the authors (see line 216). Can the authors comment on these aspects and the mobility and transport mechanisms of the virus within the gel? I suggest they could better comment and analyze Fig S11 to see if a diffusion mechanism applies. Did they perform single virus tracking experiments within the gel?

Response: As shown in the Fig. S11 (Fig. R32 attached below), when viral aerosol was intercepted by the gel, there is a concentration gradient from the top to the bottom of the gel, which might drive the movement of the virus towards the interior of the gel until a uniform distribution. We appreciate the reviewer's suggestion of single virus tracking experiment, which can be helpful for deeper understanding the movement of the virus inside the gel. However, it may be not applicable to tracking the movement of single virus towards the interior of the gel, since the resolution at the Z-axis was always compromised, even for the super-resolution imaging.

Fig. R32 (as Fig.S17 in revision) CLSM images of SPV aerosols after nebulizing to aMV@GEL for 0 min (left, contacting) and 10 min (right, fusing). The x-z images showed that the SPV aerosols released the SPV into the downward inferior aMV@GEL. We dissolved fluorescein sodium in the water of SPV aerosols (purple, false color), and the SPV in SPV aerosols were stained with Cy5 (green, false color).

Alternatively, we have now conducted computer simulations to comparatively investigate the concentration-driven movement of the virus at the Z-axis in both aqueous solution and GEL. Briefly, for simulation of virus diffusion in GEL, we first constructed a gel-framework filled with aqueous solution and considered the virus could diffuse freely in pores of GEL. Then, we endowed the virus solution on the top surface of aqueous solution or GEL. Following the assumption of continuous media, virus diffusion can be governed by convection-diffusion equation. As depicted in Fig. R33, the virus exhibited a slower movement towards the interior of the GEL compared to that of aqueous solution. Such a reduced diffusion of virus within the GEL facilitated the encounter with the MV, ensuring the efficient capture and entrapment of the virus by the MV.

Fig. R33 (as Fig. S18 in revision) Simulation depicting the diffusion of the virus in aqueous solution or GEL. The aqueous solution is represented by the white color, virus is represented by the orange color, and the GEL framework is represented by the cyanine color.

Finally, we have now performed additional experiments to determine the relationship between gel thickness and the protection effect. In our Transwell™ model, we applied aMV@GEL with different thicknesses in upper chamber, followed by the addition of SPV (1×10^5 PFU SARS-CoV-2 pseudovirus). The lower chamber contained ACE2-293T cells, which were the host cells for SPV. As shown in Fig. R34, aMV@GEL with 50 μm thickness could reduce 90% viral infection. Given that the gel layer in the tube, mouse and HRT models were all thicker than 50 μm (see more details in the response to question 3 of comment 2), the possibility of the infection in underlying epithelial cells of the nasal cavity could be negligible.

Fig. R34 Protective efficiency of aMV@GEL with different thicknesses. The Data represent the means \pm SEM. Statistical significance was tested with one-way ANOVA with multiple comparison tests. ****P < 0.0001. n.s. meant no significant difference.

We have now added above simulation data and method in our revised manuscript as follow.

Main text:

Page 8, line 223-227: These free SPVs then diffused in the GEL (Fig. S18), and colocalized with DiD-aMV (red) in the FITC-stained GEL (cyan) (Fig. 3I (iv)), indicating that the released SPV was entrapped by aMV and excluding the possibility of viral infection of the underlying host cells (Fig. S19) *in vitro* and the underlying epithelial cells in the nasal cavity *in vivo*.

Method:

Page 20-21, line 633-661: The simulation of virus diffusion in aMV@GEL. In simulation of virus diffusion in aqueous solution and GEL, GEL was regarded as gel-framework filled with aqueous solution and virus was considered diffusing freely in pores of GEL. Then, the virus solution was added on the top surface of aqueous solution or GEL. Following the assumption of continuous media, virus diffusion could be governed by convection-diffusion equation:

$$\frac{\partial c^*}{\partial t} + \mathbf{u} \cdot \nabla c^* = D \nabla^2 c^*$$

Here convection effect was neglected and the virus velocity \mathbf{u} was zero. c^* denoted dimensionless concentration which was defined as c/c_0 , where c was real concentration and c_0 was initial concentration of virus. D represented diffusion coefficient that could be described by Stokes-Einstein equation:

$$D = \frac{kT}{6\pi\mu r}$$

where k was Boltzmann's constant, T was temperature and set as 300 K, μ was viscosity of water, and r denoted radius of virus and set as 0.1 μm . Here lattice Boltzmann method with D3Q19 model⁷⁰⁻⁷¹ was used to solve the convection-diffusion equation, and the simulation was implemented by GPU-based in-house code, which has been comprehensively verified and used in our previous work⁷²⁻⁷⁴. In initial condition, virus was added on the top of aqueous solution or GEL, and c^* was set as 1, while in other regions c^* was 0. No-slip boundary condition was set on the gel-matrix, as well as top and bottom of simulation domain, with half bounce-back scheme used, while periodic boundary condition was set on other boundaries.

Reference:

70. Chen, S. & Doolen, G.D. Lattice Boltzmann method for fluid flows. *Annu Rev Fluid Mech* **30**, 329-364 (1998).

71. Qian, Y.H., Dhumieres, D. & Lallemand, P. Lattice BGK models for Navier-Stokes equation. *Europhysics Letters* **17**, 479-484 (1992).

72. Fu, S. & Wang, L. GPU-based unresolved LBM-DEM for fast simulation of gas-solid flows. *Chem Eng J* **465**, 142898 (2023).

73. Liu, X.W., Ge, W. & Wang, L.M. Scale and structure dependent drag in gas-solid flows. *Aiche Journal* **66**, e16883(2020).

74. Xiong, Q.G., et al. Large-scale DNS of gas-solid flows on Mole-8.5. *Chem Eng Sci* **71**, 422-430 (2012).

4) The authors claim they have a prolonged retention time of the gel (e.g., line 84 “...which was favorable for prolonging the retention time on the intranasal wall of the nose”, Lines 388-389 “the MV with the viral receptor that was embedded into the hydrogel could be retained in the nasal cavity in long-term”. The present study did not assess this aspect. For such an assertion, the authors should quantify this residence time or refer to data from the literature with a similar system.

Response: Actually, the data demonstrating that “the MV that was embedded into the hydrogel could be retained in the nasal cavity in long-term” had been provided in Fig. 4B-C of our initially submitted manuscript (also shown as the following Fig. R35 below). As shown in this figure, free aMV rapidly disappeared from the nasal cavity within 4 h. In sharp contrast, the gel prominently prolonged the retention of aMV in the nasal cavity, as the signal was still detectable at 8 h in the aMV@GEL group.

Fig. R35 (as Fig. 4B-C in revision) The retention of aMV and aMV@GEL in nasal cavity of mouse after intranasal administration.

(A) Representative in vivo fluorescence images of mice at the indicated time points after intranasal administration of free aMV or aMV@GEL. aMV in both groups were stained with Cy5.

(B) Relative fluorescence intensity (F.I.) of free aMV or aMV@GEL in mouse nasal cavity in A (left, n=3 biologically independent mice) and corresponding half-life (right, n=3 biologically independent mice).

The Data represent the means \pm SEM. Statistical significance was calculated using unpaired t-test. **P < 0.01.

5) Some groups have already published data on nasal sprays with mucoadhesive polymers (without microvesicles) to create a physical barrier in the nose to prevent infection with SARS-CoV-2. Some clinical trials were also performed with such formulations. The authors should discuss their study and results in this context (see, for instance, the review by Nocini et al., *Biomedicines* 2022, 10, 2966, the work of Robinson et al., *Frontiers in Medical Technology*, 2021, 3, 687681 and the communication of Bentley and Stanton, *Viruses*, 2021, 13, 2345).

Response: We appreciate the reviewer for providing these useful references. Accordingly, the references and corresponding discussion about these mucoadhesive polymers have now been added in our revised manuscript as follow.

Mian text:

Page 16, line 456-465: Some researchers had also utilized mucoadhesive polymers (such as carrageenan and gellan gum) to develop an intranasal barrier against virus⁵⁸⁻⁶¹. Typically, those polymers could wrap around the cell to form a physical barrier on the surface of epithelium cells in the nasal cavity, thus reducing the contact between the cell and virus. However, these studies focusing on the virus within the nasal cavity failed to prevent virus entering the lung through airflow. In our study, we introduced a positively charged hydrogel that not only prevented infection in the nasal cavity but also intercepted viral aerosols, thus effectively preventing their entry into the lung and substantially reducing the risk of lung infection. Further incorporation of MV displaying viral receptors in our study resulted in a notable reduction of infection rates within both the nasal cavity and lung.

Reference:

- 58. Moakes, R.J.A., Davies, S.P., Stamataki, Z. & Grover, L.M. Formulation of a composite nasal spray enabling enhanced surface coverage and prophylaxis of SARS-CoV-2. *Adv Mater* **33**, e2008304 (2021).
- 59. Nocini, R., Henry, B.M., Mattiuzzi, C. & Lippi, G. Improving nasal protection for preventing SARS-CoV-2 infection. *Biomedicines* **10**, 11 (2022).
- 60. Robinson, T.E., Moakes, R.J.A. & Grover, L.M. Low acyl gellan as an excipient to

improve the sprayability and mucoadhesion of iota carrageenan in a nasal spray to prevent infection with SARS-CoV-2. *Front Med Technol* **3**, 687681 (2021).

61. Bentley, K. & Stanton, R.J. Hydroxypropyl methylcellulose-based nasal sprays effectively inhibit in vitro SARS-CoV-2 Infection and Spread. *Viruses* **13**, 2345 (2021).

More specific aspects that require special attention from the authors are listed below:

Question 1: - Line 70: “Once humans inhale those floating viral aerosols, infection occurs”. Infection is not systematic. The infection can or may occur.

Response: We apologize for the inaccurate description, which has been revised as follow.

Mian text:

Page 3, line 69-70: These viral aerosols can remain infective for up to several days^{12,13}; once humans inhale those floating viral aerosols, infection may occur.

Question 2: - In the size distributions, also provide the polydispersity index. What dispersion medium and dilution are used for the Dynamic Light Scattering experiments?

Response: According to the reviewer’s suggestion, we have now provided the polydispersity index (PDI) of aMV and sMV in our revised Fig. S2 and Fig. S25 (also shown in Fig. R36), which were 0.332 and 0.264, respectively. The dispersion medium and dilution used for the Dynamic Light Scattering experiments was PBS, which has now been described in the revised caption of Fig. S2 and Fig. S25.

Fig. R36 (Fig. R36A as Fig. S2 and Fig. R36B as Fig. S25 in revision) Size distribution and the corresponding polydispersity index (PDI) of aMV (A) and sMV (B) in PBS solution.

Question 3: - Please provide the size distribution of nanosized vesicles (aNV).

Response: According to the reviewer’s suggestion, the size distribution of nanosized vesicles (aNV) has now been added as Fig. S5 (also shown as Fig. R37). The result indicates that the average size of aNV is 95.5 nm.

Fig. R37 (as Fig.S5 in revision) Size distribution and the corresponding PDI of aNV in PBS solution.

Question 4:- Fig 3I. What is the time scale for observing these different steps (contact of SPV with the gel, fusion of the viral aerosol with the gel, entrapment of SPV by aMV)? At what depth in the gel was image iv obtained?

Response: We would like to take this opportunity to clearly describe the time scale of Fig. 3I (also shown as Fig. R38). In detail, the top view of viral aerosols contacting the aMV@GEL (ii), the top view of viral aerosols fusing with aMV@GEL (iii), and the entrapment of the released SPV by aMV inside GEL (iv) were captured at 0 min, 10 min and 60 min, respectively. Specifically, for the (iv), the detection depth was 15 μm from the surface of the gel. We have now added more description in the revised caption of Fig. 3I as follow.

Fig. R38 (as Fig. 3I in revision) Schematic and corresponding CLSM images illustrating the detailed process of SPV (wild-type) aerosol interception and virus entrapment by aMV@GEL. i) The drops of viral aerosols consisting of water and SPV. ii) The top view of viral aerosols contacting the aMV@GEL (0 min). iii) The top view of viral aerosols fusing with aMV@GEL (10 min). iv) The entrapment of the released SPV by aMV inside GEL (60 min), which image was captured at a distance of 15 μm from the surface of the GEL. The water in viral aerosols was stained with fluorescein sodium (purple, false color), the virus in viral aerosols was stained by cyanine 3 N-hydroxysuccinimide ester (Cy3, green, false color), aMV were stained by Cy5 (red), and the gel was stained with FITC (cyan, false color).

Question 5:- Replace “gelatinized” and “gelatinization” with “gelified” and “gelation”.

Response: In the revised manuscript, the "gelatinized" and "gelatinization" have now been replaced with "gelified" and "gelation" (examples as follows).

Main text:

Page 7, line179-181: These hydrogels were liquid at room temperature (25 $^{\circ}\text{C}$) but gelified at the body temperature of ~ 37 $^{\circ}\text{C}$, providing an ideal medium for embedding aMV and prolonging retention in the nasal cavity.

Page 17, line 479-481: Considering the thermosensitive property of our gel, some points should be noted. First, our hydrogel should be stored at low temperature (below

25 °C) to avoid unexpected gelation.

Page 17, line 483-485: To further upgrade our hydrogel formulation, we can use some special hydrogels with shear-thinning properties, in which gelation was independent of the temperature.

Question 6:- Pore size” would be more appropriate than “aperture size”.

Response: According to the reviewer’s suggestion, the "pore size" has now been used to replace "aperture size" in our revised manuscript.

Main text:

Page 7, line 193-197: Compared with the GEL (Fig. S12), aMV@GEL maintained many similar features in terms of its porous structure (pore size approximately 10-20 μm , Fig. 3C), thermosensitive gelling property (360 s at 37 °C, Fig. 3D and Fig. S13), spray characteristic (spray area 2.3 cm^2 , 10 cm away from the nozzle, Fig. 3E), and positive charge (10.1 mV).

Question 7:- Several experiments were conducted at 37°C (for instance, rheological experiments). The nasal temperature is more likely to be around 34°C. What about the gelation kinetics at this temperature?

Response: We appreciate the reviewer for reminding us this point. Accordingly, we have now conducted additional experiments to analyze the gelation time of aMV@GEL at 34 °C. As shown in Fig. R39, the average gelation time of our intranasal mask at this temperature was 436 seconds. Note that the gelation time was a little bit longer than that in 37 °C (~357 s), but such a slightly extended gelation time would not compromise suitability and effectiveness of the intranasal mask for application in the nasal cavity.

Fig. R39 Evolution of dynamic loss modulus (G' , red) and storage modulus (G'' , gray) of aMV@GEL at 34°C with 3 repetitions on different aliquots. When $G'' > G'$, aMV@GEL transformed from liquid state to gel state.

Question 8: - The rheological experiments must be better described. What are the geometry characteristics (cone-plate? plate-plate? material? dimension? truncature, or gap?). Were these experiments conducted in the linear regime of deformation? What was the applied strain or stress? The angular frequency cannot be 0.01 rad/s if the frequency is 0.1Hz. Were the rheological experiments reproducible? At least three repetitions on different aliquots are required for good precision.

Response: We appreciate the reviewer for reminding us these technique issues. In rheological experiments, a stainless-steel plate-plate with smooth surface (50 mm

diameter) was used as clamp. The rheological experiment was carried out under the condition of linear regime with the constant stress (1 Pa) and frequency (1 Hz), which was erroneously written as 0.1 Hz in the originally submitted manuscript. We have now added more description about rheological experiments in method as follow.

Method:

Page 20, line 604-608: The gelation time of CS_{0.10}, CS_{0.12}, and CS_{0.14} was detected by Rheometer (MCR302, Anton-Paar) using a stainless-steel plate-plate with smooth surface (50 mm diameter) as clamp. The rheological experiment was carried out under the condition of linear regime of deformation with the constant stress (1 Pa) and frequency (1 Hz) at 37 °C. Three repeated tests for each gel were conducted.

According to the reviewer’s suggestion, we have now conducted 3 repeated tests for each gel. A shown in Fig. R40, there was negligible difference in their gelation time among 3 repetitions, indicating good reproducibility of our rheological experiments. In the revised manuscript, we have now displayed the data with representative curves, and emphasized the repetitions in the method.

Fig. R40 Evolution of dynamic loss modulus (G' , red) and storage modulus (G'' , gray) of CS_{0.10} (A), CS_{0.12} (B), CS_{0.14} (C) and aMV@GEL (D) in 37°C with 3 repetitions on different aliquots. When $G'' > G'$, gel

transformed from liquid state to gel state.

Question 9:-Fig S7A: For CS_{0.12}, the dashed vertical line does not cross the experimental curves at $G'=G''$.

Response: We appreciate the reviewer's careful review and apologize for our carelessness. In our updated version, the dashed vertical line has now been positioned crossing the experimental curves at $G'=G''$.

Question 10:-Quartz crystal microbalance: the author should provide the protocol or refer to a previous publication describing it. How is the gel layer immobilized on the surface of the chip?

Response: According to the reviewer's suggestions, we have now provided detailed description about Quartz crystal microbalance experiment in the revised version, such as gel layer immobilization and operating steps. We herein invite the reviewer to see the details as follow.

Method:

Page 20 line 618-627: The interaction force between the CS_{0.10}, CS_{0.12}, or CS_{0.14} and SPV was determined by quartz crystal microbalance (Q-SENSE E4, Biolin) according to a typical protocol⁶⁹. First, spin-coating of different hydrogel solutions (10 μ l) onto a silicon chip was conducted to form a membrane. The chip was then incubated at 37 °C for 1 hour to allow the hydrogel's gelation. Subsequently, the chip was installed into the instrument, and pure water was continuously pumped at the speed of 200 μ l/min to remove non-immobilized hydrogel. When the frequency curve reached a plateau, the SPV solution was pumped at the speed of 50 μ l/min to interact with the hydrogel, and the frequency curve was monitored until it reached a new plateau. Finally, pure water was pumped again to remove dissociated SPV until the frequency curve reached a plateau.

Reference:

69. Sha, X., et al. Quartz crystal microbalance (QCM): useful for developing procedures for immobilization of proteins on solid surfaces. *Anal Chem* **84**, 10298-10305 (2012).

Question 11: FigS7D caption: "The downward frequency reflected an increased quality on the chip, thus reflecting the interaction force between hydrogel and SPV". The word "quality" is not appropriate. The frequency is linked to the mass of SPV that is adsorbed on the chip.

Response: According to reviewer's suggestion, we have revised the caption of this figure as follow.

Fig. S9D caption: Quartz crystal microbalance curve of CS_{0.10}, CS_{0.12}, and CS_{0.14} during interaction with SPV. The downward frequency reflected an increased mass of SPV that adsorbed on the chip, thus reflecting the interaction force between hydrogel and SPV

Question 12: - FigS7E: What is the authors' interpretation of the images CLSM images?

Response: The chitosan hydrogel used in the study carried a positive charge in

aqueous solution, which had the potential to impact the tight junctions of epithelial cells in the nasal mucosa, thereby leading to the disruption of the nasal mucosa barrier. Keeping this in mind, we screened out an appropriate concentration without impact the tight junctions of epithelial cells in the nasal mucosa.

To this end, we utilized CLSM to observed the phalloidin-labeled cytoskeleton (red) of nasal epithelial cells. As shown in Fig. R41, the cytoskeleton exhibited signs of discontinuity or even disappearance when the chitosan concentration increased to CS_{0.14}, indicating the reduced and damaged tight connections between the nasal epithelial cells. We also added more description in our revised manuscript as follow.

Fig. R41. (as Fig. S9E in revision) CLSM images of human nasal epithelial cells after incubation with CS_{0.10}, CS_{0.12}, or CS_{0.14} for 1 h, accompanied with the corresponding quantitative data of relative cell tight junction degree (n=3 biologically independent experiments). The cytoskeleton was stained with FITC-phalloidin (red, false color), and cell nuclei were stained with DAPI (blue). The cytoskeleton exhibited signs of discontinuity or even disappearance when the chitosan concentration increased to CS_{0.14}, indicating the reduced and damaged tight connections between the nasal epithelial cells. Data represent the means ± SEM. Statistical significance was calculated using one-way ANOVA with multiple comparison tests. ***P < 0.001, n.s. meant no significant difference.

Question 13: - FigS7G: To obtain the radar images, how are the different relative parameters calculated? What reference is used for this normalization?

Response: In Fig. S9G (Fig. S7G in original version), the radar images were generated based on the quantified parameters of Fig. S9A-F. These parameters included gelation time, spray area, fluorescence intensity of intercepted SPV, interaction force between gel and SPV, tight junction degree between cells, and the relative cell viability. In detail, for each parameter, the highest value of three gel groups was normalized as a score of 100, representing the best performance or level in three groups. Then, the remaining values in the relative parameter were calculated as the corresponding scores. This normalization allowed for a relative comparison of the different parameters and visually showed the strengths and weaknesses of different groups in multiple aspects. We have added more description in our revised figure caption as follow.

Fig. S9G caption: Radar images of CS_{0.10}, CS_{0.12}, and CS_{0.14} showing the indicated parameters. A: relative thermosensitive gelling property; B: relative spray characteristic; C: relative interception ability to SPV aerosols; D: relative interaction force with SPV; E: relative effect on cell tight junction; F: relative cytotoxicity. To generate the radar images, the highest value of three gel groups in terms of each parameter was normalized as a score of 100, the remaining values in the relative parameter were then calculated as the corresponding scores. After normalizing these six aspects, the area (enclosed by the blue line) of these three recipes in the radar map was calculated. Considering the

overall performance in the above six aspects, we chose CS_{0.12} as our final hydrogel recipe.

Question 14: - Cytotoxicity of the formulation was only studied over a short time (1h for cell viability in Fig. S7 and observation of mice nasal mucosa in Figure S13). In the case of human application, a longer contact time will be necessary, with repeated administration. The authors should mention this limitation.

Response: We have now evaluated the long-term safety of aMV@GEL by intranasally administered aMV@GEL to mice every other day for a period of two weeks. At day 7 and day 14, we collected blood samples for blood routine and serum biochemistry index analysis, prepared nasal homogenate for inflammatory cytokine detection, and excised major tissues for histological analysis.

As shown in Fig. R42A and Fig. R42B, the blood routine analysis of white blood cells (WBC), red blood cells (RBC), and platelets (PLT), as well as the serum biochemistry indices of blood urea nitrogen (BUN), aspartate alanine aminotransferase (ALT), aminotransferase (AST), lactate dehydrogenase (LDH), and alkaline phosphatase (ALP) of mice receiving aMV@GEL were all within the normal reference ranges. Furthermore, the levels of cytokines (IFN- γ , TNF, IL-2 and IL-6) in Fig. R42C exhibited no significant difference between aMV@GEL group and health group, suggesting that aMV@GEL did not induce an inflammatory response in the nasal cavity. In addition, the H&E staining results in Fig. R42D indicated that long-term use of aMV@GEL did not cause apparent damage to the nasal cavity, brain, lung, heart, liver, spleen, and kidney. Beyond the data of basic respiratory function and structure of nasal epithelium layer provided in the originally submitted manuscript, above data further provided strong evidence to support the safety of aMV@GEL material upon multiple administrations in the nasal cavity.

Fig. R42 (as Fig. S21 in revision) Biosafety evaluations upon multiple aMV@GEL administrations. aMV@GEL was intranasally administered to mice every other day for a period of two weeks. At day 7 and day 14, the blood and the major tissues (including the nasal cavity, brain, lung, heart, liver, spleen, and kidney) were collected for biosafety evaluations.

(A) Blood analysis of white blood cells (WBC), red blood cell (RBC), and platelet (PLT) at day 7 and day 14 (n=3 biologically independent mice). The blue background presented the normal range.

(B) Serum biochemistry indices analysis of urea nitrogen (BUN), aspartate alanine aminotransferase (ALT), aminotransferase (AST), alkaline phosphatase (ALP), and lactate dehydrogenase (LDH) at day 7 and day 14 (n=3 biologically independent mice). The blue background presented the normal range.

(C) Inflammatory cytokines (IFN- γ , TNF, IL-2 and IL-6) analysis of the nasal homogenate at day 7 and day 14 (n=3 biologically independent mice).

(D) Representative H&E staining images of major tissues at day 7 and day 14.

Data in A, B and C represent the means \pm SEM. Statistical significance in C was calculated using one-way ANOVA with multiple comparison tests. n.s. meant no significant difference.

We have now added above data, description and method in our revised manuscript as follow.

Main text:

Page 8-9, line 236-239: In addition, the comprehensive evaluations, including blood routine analysis, serum biochemistry index analysis, nasal inflammatory cytokine detection, and histological analysis of major tissues, confirmed the safety upon multiple aMV@GEL administrations (Fig. S21).

Method:

Page 21, line 653-661: The biosafety evaluations of the long-term intranasal administration of aMV@GEL. The mice were intranasally administered with aMV@GEL every other day in a period of two weeks. The blood and the major tissues (including the nasal cavity, brain, lung, heart, liver, spleen, and kidney) were collected at both day 7 and day 14. The blood was used for blood routine (WBC, RBC and PLT) and biochemical indicators (BUN, ALT, AST, LDH and ALP) analysis by using automatic haemocytometer analyzer and biochemical analyzer, respectively. The inflammatory cytokines (IFN- γ , TNF, IL-2 and IL-6) of the nasal homogenate were detected by the corresponding ELISA kit. The major tissues were fixed and prepared into paraffin sections (3-4 μ m) for H&E staining.

Question 15: - In what material was the nasal apparatus printed? This has an impact on how the hydrogel aerosol and virus aerosol interact with the printed nasal surface.

Response: We employed polyethylene terephthalate glycol (PETG) as the material for fabricating the nasal apparatus due to its unique advantages as follow. Firstly, PETG exhibits excellent light transmission property, enabling clear visualization of the internal state of the nasal cavity. Secondly, its wettability facilitates the gel to spread well on the surface, ensuring comprehensive coverage and effective contact of gel on the apparatus. Lastly, PETG possesses low charge density, promoting efficient gel droplet dispersion and minimizing the interference to virus aerosol movement.

REVIEWER COMMENTS

Reviewer #1 (Remarks to the Author):

Major Concerns:

1. Statistical tests should be clearly stated and applied correctly. Figure 2 I and Line 158 specify that a multiple comparison test was used following the 1-way ANOVA. It appears an error may have been made when reporting the p-values. The variance between the 3 replicates within the groups is likely too high for the p-value for all of the multiple comparisons to be <0.0001.

2. In figure 3, the authors demonstrate that GEL alone was sufficient to limit viral travel. Were GEL only controls (GEL without MV) included in the in vivo trials in order to determine if MV incorporation is necessary for the protective effects? Additionally, was a relationship between density of MV in GELs and antiviral efficacy determined?

3. Writing needs editing in many places, including these selected examples :

Line 103-This sentence clause needs to be edited for clarity "The unique merits of independent ability to viral mutation/serotype,".

Line 109- "Inspired by receptor-mediated viral infections in host cells²³, studies that using viral receptor as virus decoy²⁸⁻³¹, and the size of most virus ranging from 60 nm to 140 nm³²" and "We envisioned that the cell-derived vesicles with abundant viral receptors and a large cavities could facilitate the viral entrapment³³

Line 134-"Additionally, the aMV showed reproducible characteristics of average size and ACE2 expression (Fig. S3), paving the way for subsequent interaction and entrapment of SARS-CoV-2."

Line 271-"Slimier"

Other Concerns:

The techniques described would likely not be sufficient to determine the absolute amount of ACE2 per vesicle. In Line 132- "per vesicle" may not be the appropriate denominator as vesicles of different sizes would contain different amounts of ACE2.

Line 199- aMV surviving spray induced shear rates is not sufficient evidence to "ensure" the entrapment function.

Line 73- "However, the protective effect of face masks against severe acute respiratory syndrome coronavirus 2 (SARS-CoV-2) was only 63% " The citations describe masking, but do not all support the statement. The original paper noting 63% efficacy should be cited.

Line 224 and Figure 2 F and H- The GEL without aMV had a lower mean Relative passing rate than aMV. Please clarify if aMV@GEL conferred an advantage over GEL alone in preventing viral travel.

Line 267- "nasal cavity and lung of the aMV@GEL group contained few if any infected cells". Instead of "few if any" this needs a description of the data. Please state if the GFP expression is below the limit of detection and if infection was therefore not detectable with MV@GEL treatment.

Line 298- Protection against two strains of H1N1 is not sufficient evidence to demonstrate the treatment will work for all influenzas, independent of serotype.

Line 435- Long-term may be inaccurate considering the half-life reported in mice.

The discussion highlights potential advantages and pitfalls on the road to clinical application of the described technologies, but does not address the <12 hr half-life of the treatment.

Reviewer #2 (Remarks to the Author):

The reviewer appreciated the authors' great efforts of revision. The manuscript quality has significantly improved. However, some concerns need to be further addressed or clarified.

Main concern:

1) The review appreciated that the authors acknowledge the limitations of the MV@gel. How the authors ensure the MV@gel in the nose not do not enter the mucosal system/respiratory system for long-term use? If it becomes a solid-phase gel in the nose, would it block the normal function of nose

for air exchange and other functions or cause discomfort?

2) The reviewer appreciated the authors' significant efforts to address the safety concern. Can the authors provide T cell immune response data as it is essential for HLA-mismatch concern? Also, can the authors provide the statistical analyses for Panels A and B as they did for Panel C?

Other concerns:

1a. Addressed.

1b. Addressed.

1c. Addressed.

1d. While the authors acknowledge the limitations, the discussion did not really address the high temperature issue.

2. The authors should provide evidence to justify their claims.

3. Addressed.

4. Addressed.

5. Addressed.

6. Addressed.

7. Addressed.

8. Okay. The reviewer understands experimental facility restriction. Please provide a discussion regarding the limitations of this study on this matter.

9. Addressed.

Minor point: while the authors have cited several more papers, there are still significant ones missed.

Reviewer #3 (Remarks to the Author):

The authors have satisfactorily addressed all my main concerns. This reviewer appreciates all the effort and work put on the revised manuscript. Pending more in-depth studies, these type of masks could indeed represent a novel and safe method to prevent viral infections in the future.

Reviewer #4 (Remarks to the Author):

Authors have addressed properly of my comments, so from my side the manuscript can be accepted

Reviewer #5 (Remarks to the Author):

I thank the authors for considering my comments. They answered my questions with great care and convincingly. They supplied missing experimental data. Additional experiments were performed and added to the manuscript or as supplementary information to support some conclusions further. I think this revised version of the article deserves publication in Nature Communications.

RESPONSES TO REVIEWERS

(Note: All the responses here are in blue. All the changes have been highlighted in yellow in the main text of revised manuscript.)

For Reviewer #1:

Major Concerns:

1. Statistical tests should be clearly stated and applied correctly. Figure 2 I and Line 158 specify that a multiple comparison test was used following the 1-way ANOVA. It appears an error may have been made when reporting the p-values. The variance between the 3 replicates within the groups is likely too high for the p-value for all of the multiple comparisons to be <0.0001 .

Response: We appreciate the reviewer's careful review. Actually, we had realized the importance of the variance homogeneity in the one-way ANOVA statistical analysis. Taking this opportunity, we herein make clarifications for the statistical analysis of Fig. 2I. First, we performed Brown-Forsythe test in GraphPad Prism 8.4.3 to assess the variance homogeneity of various groups (Table R1). As shown in Fig. R1A, there was no significant difference in the variance within the groups, which was further confirmed by variance homogeneity test in Origin2023b software (Fig. R1B). Then we performed the ordinary one-way ANOVA analysis in GraphPad Prism 8.4.3 (Fig. R1C). As shown in Fig. R1D, the p-value of aMV vs. PBS, aMV vs. ACE2 protein, aMV vs. aNV, aMV vs. nMV and aMV vs. aMV/anti-ACE2 were all less than 0.0001.

Table R1. Original data of Figure 2I in manuscript

PBS	ACE2 protein	aNV	nMV	aMV	aMV/anti-ACE2
103.8367	85.43672	49.03904	87.33549	7.046516	90.01976
98.69163	84.45431	43.22938	73.94201	6.711645	109.2983
97.47178	81.05382	51.27369	81.93523	6.847732	97.11348

A

Brown-Forsythe test	
F (DFn, DFd)	3.787 (2, 6)
P value	0.0864
P value summary	ns
Are SDs significantly different (P < 0.05)? No	

B

Homogeneity of Variance Test

Levene's Test (Absolute Deviations)

	DF	Sum of Squares	Mean Square	F Value	Prob>F
Model	5	86.61848	17.3237	2.68544	0.07471
Error	12	77.41177	6.45098		

At the 0.05 level, the population variances are not significantly different.

Levene's Test (Squared Deviations)

	DF	Sum of Squares	Mean Square	F Value	Prob>F
Model	5	8645.61771	1729.12354	2.69319	0.07414
Error	12	7704.42295	642.03525		

At the 0.05 level, the population variances are not significantly different.

Brown-Forsythe Test

	DF	Sum of Squares	Mean Square	F Value	Prob>F
Model	5	76.13424	15.22685	1.25204	0.34515
Error	12	145.93966	12.16164		

At the 0.05 level, the population variances are not significantly different.

C

Parameters: One-Way ANOVA (and Nonparametric or Mixed)

Experimental Design | Repeated Measures | Multiple Comparisons | Options | Residuals

Experimental design

No matching or pairing
 Each row represents matched, or repeated measures, data

	Group A	Group B	Group C	Group D
	Data Set-A	Data Set-B	Data Set-C	Title
	Y	Y	Y	Y
1				
2				

Assume Gaussian distribution of residuals?

Yes. Use ANOVA.
 No. Use nonparametric test.

Assume equal SDs?

Yes. Use ordinary ANOVA test.
 No. Use Brown-Forsythe and Welch ANOVA tests.

Based on your choices (on all tabs), Prism will perform:
 - Ordinary one-way ANOVA.

Learn Cancel OK

D

ANOVA results

	Dunnnett's multiple comparisons t	Mean Diff.	95.00% CI of diff.	Significant?	Summary	Adjusted P Value	E-?
5	aMV vs. PBS	-93.13	-105.9 to -80.36	Yes	****	<0.0001	A PBS
7	aMV vs. ACE2 protein	-76.78	-89.55 to -64.01	Yes	****	<0.0001	B ACE2 protein
8	aMV vs. aNV	-40.98	-53.75 to -28.21	Yes	****	<0.0001	C aNV
9	aMV vs. nMV	-74.20	-86.98 to -61.43	Yes	****	<0.0001	D nMV
10	aMV vs. aMV/anti-ACE2	-91.94	-104.7 to -79.17	Yes	****	<0.0001	F aMV/anti-ACE2

Fig. R1 Statistical test process of Figure 2I.

- A. Brown-Forsythe test result of various groups conducted by GraphPad Prism 8.4.3.
- B. Variance homogeneity test results of various groups conducted by Origin2023b.
- C. Experimental design of one-way ANOVA in GraphPad Prism 8.4.3.

D. Ordinary one-way ANOVA result of various groups conducted by GraphPad Prism 8.4.3.

2. a) In figure 3, the authors demonstrate that GEL alone was sufficient to limit viral travel. Were GEL only controls (GEL without MV) included in the *in vivo* trials in order to determine if MV incorporation is necessary for the protective effects?

Response: We thank for the reviewer's careful review. We herein clarify that the aim of figure 3 was to demonstrate the interception effect of GEL *in vitro*, which did not involve the evaluation of aMV's entrapment function on the virus inactivation and subsequent infection inhibition. Actually, in Fig. 4G and 4H of our previous submitted manuscript (also shown as Fig. R2A and R2B here), we had included the GEL only control in the *in vivo* SARS-CoV-2 pseudovirus aerosols infection experiment. In this group, a large area of infected GFP signal appeared in the nasal cavity, but a small area appeared in the lung, indicating that GEL alone prevented viral aerosols into lung but was helpless to prevent the virus infection to nasal epithelial cells. By combining aMV and GEL, the infected GFP signal neither appeared in nasal cavity nor lung, indicating the necessity of incorporating aMV in GEL for the efficient protection against virus in nasal cavity. To further address the reviewer's concerns, we have now emphasized the necessity of both GEL and MV in our revised manuscript as follow.

Main text:

Page 9-10, line 269-272: By combining aMV and GEL, the infected GFP signals neither appeared in nasal cavity nor lung of aMV@GEL group, which should be attributed to the viral aerosol interception of GEL (Fig. S23) and virus inactivation of aMV in nasal cavity.

Fig. R2 (Fig. 4G and 4H of the manuscript) *In vivo* protective effect of different treatments against SPV aerosols.

A. Representative frozen section images and enlarged views of mouse nasal cavity (left) and relative level of GFP mRNA in the nasal cavity by using q-PCR detection (right, n=3 biologically independent mice) after 3 days of challenge with SPV (wild-type) aerosols. GFP was produced by SPV-infected cells (green), and the cell nuclei were stained with DAPI (blue).

B. Representative frozen section images and enlarged views of mouse lung (left) and relative level of GFP mRNA in the lung by using q-PCR detection (right, n=3 biologically independent mice) after 3 days of challenge with SPV (wild-type) aerosols. GFP was produced by SPV-infected cells (green), and the cell nuclei were stained with DAPI (blue). Data in A and B represent the means \pm SEM. Statistical significance was calculated using one-way ANOVA with multiple comparison tests (A and B). ***P < 0.001, ****P < 0.0001, n.s. meant no significant difference.

b) Additionally, was a relationship between density of MV in GELs and antiviral efficacy determined?

Response: Actually, we had conducted *in vitro* experiment to evaluate the antiviral efficacy of aMV@GEL with different aMV loading densities (0 μ g, 1 μ g, 5 μ g, 10 μ g and 100 μ g aMV per 100 μ l GEL) (Fig. R3A). Briefly, we applied aMV@GEL in upper chamber, following by the addition of SPV (1×10^5 PFU SARS-CoV-2 pseudovirus). The lower chamber contained ACE2-293T cells, which were the host cells for SPV. As shown in Fig. R3B, the infection rates decreased with increasing loading density of aMV, indicating there was a positive relationship between density of MV in GELs and antiviral efficacy. Note that the infection rate was almost undetectable when the loading density of aMV reached to 100 μ g.

Fig. R3 Protection efficiency of different aMV@GELs with the indicated loading density of aMV.

A. Schematic illustration of a Transwell™ model for evaluating the infection rate of SPV (wild-type) challenged ACE2-293T cells (in the bottom chamber) upon aMV@GELs treatment with different aMV loading (in upper chamber).

B. Relative infection rates of ACE2-293T cells in the different aMV@GELs groups (n=3 biologically independent experiments). Data in B represent the means \pm SEM.

3. Writing needs editing in many places, including these selected examples:

Response: For better readability of our manuscript, we had asked American Journal Experts (AJE) for language improvement (code: F998-D162-8E95-5985-BB56) during the first revision (Fig. R4). Corresponding certificate was listed below.

Fig. R4 Language improvement certificate of English editing in AJE.

In this revision, we have further carefully edited the text for improvement, such as the following reviewer mentioned examples.

a) Line 103-This sentence clause needs to be edited for clarity “The unique merits of independent ability to viral mutation/serotype,”.

Response: The sentence has now been re-edited in our revised manuscript as follow.

Main text:

Page 4, line 105-108: The unique merits of broad protection against different viral mutation/serotype, flexible receptors engineering for matching different types of viruses, and translational potential supported our intranasal mask as a promising shield modality for improving public health in the pandemic (Fig. 1C).

b) Line 109- “Inspired by receptor-mediated viral infections in host cells²³, studies that using viral receptor as virus decoy²⁸⁻³¹, and the size of most virus ranging from 60 nm to 140 nm³²” and “We envisioned that the cell-derived vesicles with abundant viral receptors and a large cavities could facilitate the viral entrapment³³”

Response: This sentence has now been re-edited as follow.

Mian text:

Page 4, line 111-113: Inspired by receptor-mediated viral infections in host cells²³, studies using viral receptors as virus decoys²⁸⁻³⁴, and the size of most viruses ranging from 60 nm to 140 nm³⁵, we envisioned that cell-derived vesicle with abundant viral receptors and a large cavity could facilitate viral entrapment³⁶.

c) Line 134-“Additionally, the aMV showed reproducible characteristics of average size and ACE2 expression (Fig. S3), paving the way for subsequent interaction and entrapment of SARS-CoV-2.”

Response: This sentence has now been re-edited as follow.

Main text:

Page 5, line 136-138: Additionally, aMV exhibited consistent characteristics in terms of average size and ACE2 expression (Fig. S3), thus ensuring the production repeatability and paving the way for subsequent interaction and entrapment of SARS-CoV-2.

d) Line 271-“Slimier”

Response: We appreciate the reviewer's careful review and apologize for the spelling mistake in our previous manuscript. The “slimier” has been replaced with “similar” in our revised manuscript as follow.

Main text:

Page 10, line 272-275: In another experiment in which mice were challenged with SPV variant (B.1.1.529, Omicron) aerosols, corresponding GFP expression analysis in the nasal cavity and lung also showed the similar protective effect of aMV@GEL (Fig. 4I, Fig. S24), indicating that the protective effect of aMV@GEL was independent of the variants.

Other Concerns:

1. The techniques described would likely not be sufficient to determine the absolute amount of ACE2 per vesicle. In Line 132- “per vesicle” may not be the appropriate denominator as vesicles of different sizes would contain different amounts of ACE2.

Response: We appreciate the reviewer for pointing out this issue. We have now revised the description about the amount of ACE2 on vesicle as follow.

Main text:

Page 5, line 132-134: As expected, there was no obvious ACE2 signal (white) on nMV, but aMV contained a substantial accumulation of ACE2 on the surface (Fig. 2E), with a loading amount of 50 µg ACE2 per mg aMV.

2. Line 199- aMV surviving spray induced shear rates is not sufficient evidence to “ensure” the entrapment function.

Response: According to reviewer’s suggestion, the sentence has been revised as follow.

Main text:

Page 7, line 199-202: Additionally, the integrity and *in vitro* anti-viral efficacy of aMV embedded in GEL was not influenced after experiencing spray-induced shear rates (Fig. S14 and Fig. S15), paving the way for exerting entrapment function upon spray in the nasal cavity.

3. Line 73- “However, the protective effect of face masks against severe acute respiratory syndrome coronavirus 2 (SARS-CoV-2) was only 63% “ The citations describe masking, but do not all support the statement. The original paper noting 63% efficacy should be cited.

Response: We appreciate the reviewer's careful review. We have now moved the reference 15 (*Science* 372, 1439-1443 (2021)) to the previous sentence in our revised manuscript as follow.

Main text:

Page 3, line 73-74: Utilizing face masks has been among the important elements of public health efforts to reduce the rates of respiratory infections^{14,15}, especially during the COVID-19 pandemic.

Reference in main text:

14. Chan, J.F., et al. Surgical mask partition reduces the risk of noncontact transmission in a golden syrian hamster model for coronavirus disease 2019 (COVID-19). *Clin Infect Dis* 71, 2139-2149 (2020).

15. Cheng, Y.F., et al. Face masks effectively limit the probability of SARS-CoV-2 transmission. *Science* 372, 1439-1443 (2021).

As for the paper noting protective efficacy of face masks against SARS-CoV-2, actually, we had cited the original paper as reference 16 (*The Lancet* 395, 1973-1987 (2020)) in our previous submitted manuscript. In this review, the authors assessed the use of face masks to prevent transmission of viruses. Based on this review, the protective efficacy of face masks was indicated as 67% in page 188 of another paper published on Nature (*Nature* 586, 186-189 (2020)), which had also been cited as reference 17 in our previous manuscript. We herein apologize for the incorrect value (63%) provided in our previous manuscript. We have now corrected the protective efficacy in our revised manuscript as follow.

Main text:

Page 3, line 75-76: However, the protective effect of face masks against severe acute respiratory syndrome coronavirus 2 (SARS-CoV-2) was only 67%^{16,17}.

Reference in main text:

16. Chu, D.K., et al. Physical distancing, face masks, and eye protection to prevent person-to-person

transmission of SARS-CoV-2 and COVID-19: a systematic review and meta-analysis. *The Lancet* **395**, 1973-1987 (2020).

17. Peeples, L. What the data say about wearing face masks. *Nature* **586**, 186-189 (2020).

4. Line 224 and Figure 2 F and H- The GEL without aMV had a lower mean Relative passing rate than aMV. Please clarify if aMV@GEL conferred an advantage over GEL alone in preventing viral travel.

Response: We suppose the reviewer's concern should be related to Figure 3F and 3H. The positively charged GEL could intercept the negatively charged viral aerosols that were present in airflow. Focusing on the aspect of preventing viral travel alone, GEL and aMV@GEL shared much similar effect *in vitro*. However, it bears mentioning that the receptor on the vesicles could interact with the virus, thereafter mediating the entrapment of virus for inactivation. Therefore, aMV@GEL conferred an advantage over GEL in the aspect of preventing the infection of the nasal epithelial cells *in vivo* (Fig. R2).

5. Line 267- "nasal cavity and lung of the aMV@GEL group contained few if any infected cells". Instead of "few if any" this needs a description of the data. Please state if the GFP expression is below the limit of detection and if infection was therefore not detectable with MV@GEL treatment.

Response: We appreciate the reviewer's suggestion that we should describe data instead of using "few if any". Accordingly, we have now revised this sentence as follow.

Main text:

Page 9-10, line 269-272: By combining aMV and GEL, the infected GFP signals neither appeared in nasal cavity nor lung of aMV@GEL group, which should be attributed to the viral aerosol interception of GEL (Fig. S23) and virus inactivation of aMV in nasal cavity.

As for the question that if the GFP expression is below the limit of detection, we herein make a statement. For the CLSM imaging, the GFP signal of MV@GEL group was below the threshold of detection. For a more sensitive detection, we also conducted the q-PCR analysis, showing a very low level of GFP mRNA in the MV@GEL group.

6. Line 298- Protection against two strains of H1N1 is not sufficient evidence to demonstrate the treatment will work for all influenzas, independent of serotype.

Response: We appreciate the reviewer's rigorous academic attitude. Considering most of influenzas virus binding to the sialic acid on the surface of host cell, we propose our sMV@GEL, in principle, have potential to provide broad protection against most of influenzas virus. Although

two strains of H1N1 were tested in our manuscript, we trust the reviewer will concur that our results are informative to show above potential of our sMV@GEL. To further address the reviewer's concern, we have now revised the corresponding sentences in our revised manuscript as follow.

Main text:

Page 10-11, line 299-301: Furthermore, sMV also achieved a similar protective effect against the H1N1-PR8 virus (Fig. S27), suggesting the protective efficacy of sMV@GEL against the H1N1 virus had the potential to be independent of the viral serotype.

Page 11, line 321-324: Similar protective effects of sMV@GEL were also observed in mice challenged with H1N1-PR8, again indicating that the protective effect of sMV@GEL had the potential to be independent of the viral serotype (Fig. S30).

7. Line 435- Long-term may be inaccurate considering the half-life reported in mice.

Response: We thank the reviewer for reminding us this point. We have now revised this description in our revised manuscript as follow.

Main text:

Page 15, line 438-440: the MV with the viral receptor that was embedded into the hydrogel could be retained in the nasal cavity for 8 h and effectively entrapped the viruses that were released from aerosols.

8.The discussion highlights potential advantages and pitfalls on the road to clinical application of the described technologies, but does not address the <12 hr half-life of the treatment.

Response: We have now emphasized the retention time and provided additional usage instruction for high-risk individuals in our revised discussion as follow.

Main text:

Page 17, line 483-487: Considering the thermosensitive property and intranasal retention time (about 8 h) of our intranasal mask, some points should be noted. First, our hydrogel should be stored at temperature below 25 °C to avoid unexpected gelation, and should be reapplied for individuals in time with repeated exposure in high-risk environments.

For Reviewer #2:

The reviewer appreciated the authors' great efforts of revision. The manuscript quality has significantly improved. However, some concerns need to be further addressed or clarified.

Main concern:

1) a. The review appreciated that the authors acknowledge the limitations of the MV@gel. How the authors ensure the MV@gel in the nose not do not enter the mucosal system/respiratory system for long-term use?

Response: We thank the reviewer for reminding us this point. The rational design of MV@GEL formulation excluded the possibility of entering the mucosal system/respiratory system. Regarding the respiratory system, it's important to note that only small particles (<10 μm) could flow into the lung¹. Given that the average size of our MV@GEL spray was 220.5 μm (Fig. R5), it was unlikely that MV@GEL would enter the respiratory system with intranasal administration. Upon further rapid gelation, the MV@GEL stably adhered on the epithelium cells in the nasal cavity^{2,3}, which restricted the translocation into the lung. Note that such a bulk gel form also prevented the infiltration into the submucosa, further eliminating the safety concern for long-term use.

For verification, we conducted an additional experiment. Briefly, Cy5 labeled aMV@GEL was intranasally administered to mice every other day for a period of two weeks. At the day 15, nasal cavity, trachea and lung were collected for fluorescence detection. Beyond observation of aMV@GEL remaining in nasal cavity *ex vivo*, it was clear from the frozen section image that the aMV@GEL signal appeared atop the nasal cell layer (Fig. R6A), indicating that the submucosa infiltration did not occur. As to the respiratory system, both *ex vivo* fluorescence imaging and frozen section images showed that the aMV@GEL signal did not exist in the respiratory system (Fig. R6B). These results collectively demonstrated that MV@GEL in the nasal cavity did not enter the mucosal and respiratory systems even with long-term use.

Fig. R5 Size distribution of the droplets when spraying MV@GEL.

Fig. R6 Representative *ex vivo* fluorescence imaging and frozen section images of nasal cavity, trachea and lung of mice after long-term use of aMV@GEL.

A. Representative *ex vivo* fluorescence imaging and corresponding frozen section images of nasal cavity after long-term use of MV@GEL. aMV@GEL was stained by Cy5 (red), the nuclei were stained by DAPI (blue).

B. Representative *ex vivo* fluorescence imaging and corresponding frozen section images of trachea and lung after long-term use of MV@GEL. aMV@GEL was stained by Cy5 (red), the nuclei were stained by DAPI (blue).

Reference in response:

1. Morawska, L. & Buonanno, G. The physics of particle formation and deposition during breathing. *Nat Rev Phys* **3**, 300-301 (2021).
2. Kim, K., Kim, K., Ryu, J.H. & Lee, H. Chitosan-catechol: a polymer with long-lasting mucoadhesive properties. *Biomaterials* **52**, 161-170 (2015).
3. Yang, J., Bai, R. & Suo, Z. Topological adhesion of wet materials. *Adv Mater* **30**, e1800671 (2018).

b. If it becomes a solid-phase gel in the nose, would it block the normal function of nose for air exchange and other functions or cause discomfort?

Response: Actually, we had detected the respiratory function of mouse before and after aMV@GEL treatment in Fig. S20 of our previous submitted manuscript (also shown as Fig. R7 here). The respiratory rate, tidal volume, and ventilation capacity of mice after aMV@GEL treatment were all in normal range, indicating the function of nose was not influenced by MV@GEL treatment. This should be attributed to the fact that the liquid MV@GEL only formed a gel layer on the surface of the nasal epithelium cell (Fig. R6A in previous response) and the remained nasal cavity was available for air exchange.

To our knowledge, multiple nasal gels (such as Otrivin Nasal Spray, Restylane, RegenKit PRP) have been widely used in clinic. Upon consulting with rhinology experts in clinic, we have now realized that this formulation is well accepted and has negligible effect on the air exchange and other functions, although it may bring a bit of discomfort. Therefore, we propose that the recipients will be willing to use MV@GEL especially when they expose to high-risk environment with a high concentration of viral aerosols.

Fig. R7 The respiratory rate, tidal volume and ventilation capacity of mice before and after intranasal administration with aMV@GEL. The above three indicators were all in the normal ranges, indicating the existence of aMV@GEL did not disturb the normal breathing of the mice.

2. a) The reviewer appreciated the authors' significant efforts to address the safety concern. Can the authors provide T cell immune response data as it is essential for HLA-mismatch concern?

Response: According to the reviewer's comment, we have now evaluated the T cell immune response in nose after the long-term use of aMV@GEL. In brief, aMV@GEL was intranasally administered to mice every other day for two weeks. The nose and lung were collected for T cell activation analysis at day 5 post the last aMV@GEL administration.

As shown in Fig. R8A, the proportion of both CD4⁺ T cells and CD8⁺ T cells in nose exhibited no significant difference between aMV@GEL group and PBS group. Moreover, the activated T cells (CD69⁺ cells and IFN- γ ⁺ cells) proportions in CD4⁺ T cells and CD8⁺ T cells were also found to be comparable between the two groups. We also assessed above aspects in lung and again found much similar results between these two groups (Fig. R8B). Above result collectively indicated that our intranasal application of MV@GEL did not induce T cell immune response in either nose or lung, which underscores that our MV@GEL formulation without HLA-matching concern does not elicit T cell response.

Fig. R8 Immune response data of T cell in nose and lung of mice with PBS or aMV@GEL treatment.

A. Percentage of CD4⁺ T cells and CD8⁺ T cells in nose, and corresponding percentage of activated (CD69⁺ or IFN- γ ⁺) cells in CD4⁺ T cells and CD8⁺ T cells (n=3 biologically independent mice).

B. Percentage of CD4⁺ T cells and CD8⁺ T cells in lung, and corresponding percentage of activated (CD69⁺ or IFN- γ ⁺) cells in CD4⁺ T cells and CD8⁺ T cells (n=3 biologically independent mice).

Data in A and B represent the means \pm SEM. Statistical significance in A and B were calculated with unpaired t-test. n.s. meant no significant difference.

b. Also, can the authors provide the statistical analyses for Panels A and B as they did for Panel C?

Response: We appreciate the reviewer's suggestion on the statistical analyses. Accordingly, the statistical analyses for Panels A and B have now been provided in the Fig. S21A-S21B of our revised version. As shown in the Fig. R9, all the blood cells and serum biochemistry indices showed no

significant differences among three groups.

Fig. R9. Blood and serum biochemistry indices analysis upon multiple aMV@GEL administrations.

A. Blood analysis of white blood cells (WBC), red blood cell (RBC), and platelet (PLT) at day 7 and day 14 (n=3 biologically independent mice). The blue background presented the normal range.

B. Serum biochemistry indices analysis of urea nitrogen (BUN), aspartate alanine aminotransferase (ALT), aminotransferase (AST), alkaline phosphatase (ALP), and lactate dehydrogenase (LDH) at day 7 and day 14 (n=3 biologically independent mice). The blue background presented the normal range.

Data in A and B represent the means \pm SEM. Statistical significance in A and B were calculated using one-way ANOVA with multiple comparison tests. n.s. meant no significant difference.

Other concerns:

1a. Addressed.

1b. Addressed.

1c. Addressed.

1d. While the authors acknowledge the limitations, the discussion did not really address the high temperature issue.

Response: According to the reviewer's comment, we have now added the additional discussion about the high temperature issue in our revised manuscript as follow.

Main text:

Page 17, line 487-491: Moreover, people should avoid extremely cold environments or torrid environments when using our intranasal mask, because the excessively low ambient temperature (below 8 °C) may delay the gelation time in the nose and the long-time exposure to excessively high ambient temperature (over 38 °C) may lead to unexpected gelation in the container.

2. The authors should provide evidence to justify their claims.

Response: According to the reviewer's suggestion, we have now conducted an additional experiment to evaluate the interaction of embedded MV with SPV when bacteria exit in GEL. Briefly, we added hocheist-labeled bacteria in aMV@GEL. Then, the Cy3-labeled SPV was added into above bacteria doped aMV@GEL and the interaction of Cy5-labeled aMV with Cy3-SPV was evaluated by CLSM. As shown in Fig. R10, the SPV signal (green) appeared in aMV (red) when

bacteria (blue) existed in GEL (cyan). This observation indicated that the interaction of MV with SPV was not influenced when the bacteria existed in GEL.

Fig. R10 CLSM images of embedded aMV interaction with SPV when bacteria exist in GEL. Bacteria was stained by hocheist-33342 (blue), gel was stained by FITC (cyan, false color), virus was stained by cyanine 3 N-hydroxysuccinimide ester (Cy3, green, false color), and aMV were stained by Cy5 (red).

3. Addressed.
4. Addressed.
5. Addressed.
6. Addressed.
7. Addressed.

8. Okay. The reviewer understands experimental facility restriction. Please provide a discussion regarding the limitations of this study on this matter.

Response: We thank the reviewer for understanding the facility restriction. Accordingly, we have now provided an additional discussion in our revised manuscript as follow.

Main text:

Page 18, line 512-516: Moreover, due to the constraints of our cooperative P3 laboratory, we employed SPV aerosols instead of authentic virus aerosols in our SARS-CoV-2 infection experiment, which might not fully replicate real-world scenarios. Alternatively, we employed authentic H1N1 viral aerosols to provide convincing evidences for MV@GEL's protection effect in real-world scenario.

9. Addressed.

Minor point: while the authors have cited several more papers, there are still significant ones missed.

Response: Accordingly, we have now cited additional literatures that using viral receptor as virus decoy as follow.

Main text:

Page 4, Line 111-113: Inspired by receptor-mediated viral infections in host cells²³, studies using viral receptors as virus decoys²⁸⁻³⁴, and the size of most viruses ranging from 60 nm to 140 nm³⁵, we envisioned that cell-derived vesicle with abundant viral receptors and a large cavity could facilitate viral entrapment³⁶.

Reference in main text:

23. Hoffmann, M., et al. SARS-CoV-2 cell entry depends on ACE2 and TMPRSS2 and is blocked by a clinically proven protease inhibitor. *Cell* **181**, 271-280 e278 (2020).

28. Porotto, M., Yi, F., Moscona, A. & LaVan, D.A. Synthetic protocells interact with viral nanomachinery and inactivate pathogenic human virus. *PLoS One* **6**, e16874 (2011).

29. Li, Z., et al. Cell-mimicking nanodecoys neutralize SARS-CoV-2 and mitigate lung injury in a non-human primate model of COVID-19. *Nat Nanotechnol* **16**, 942-951 (2021).

30. Coccozza, F., et al. Extracellular vesicles containing ACE2 efficiently prevent infection by SARS-CoV-2 Spike protein-containing virus. *J Extracell Vesicles* **10**, e12050 (2020).

31. Wang, C., et al. Membrane Nanoparticles Derived from ACE2-Rich Cells Block SARS-CoV-2 Infection. *ACS Nano* **15**, 6340-6351 (2021).

32. Zhang, Q., et al. Cellular nanosponges inhibit SARS-CoV-2 infectivity. *Nano Lett* **20**, 5570-5574 (2020).

33. El-Shennawy, L., et al. Circulating ACE2-expressing extracellular vesicles block broad strains of SARS-CoV-2. *Nat Commun* **13**, 405 (2022).

34. Rao, L., et al. Decoy nanoparticles protect against COVID-19 by concurrently adsorbing viruses and inflammatory cytokines. *Proc Natl Acad Sci U S A* **117**, 27141-27147 (2020).

REVIEWERS' COMMENTS

Reviewer #2 (Remarks to the Author):

Major concerns:

1. Good.
2. Good.
3. It is highly beneficial for authors to seek professional editing services to improve their English. However, readability is subjective, and I will defer to you (the editor) on this matter.

Other concerns:

I am uncertain whether the 'other concerns' mentioned by the original reviewer #1 pertain solely to editing the manuscript. Please note that my following comments are based solely on the editing request and not the experimental requests. If experimental requests, there should be additional work need to be done. I will defer to you (the editor) on this matter.

1. Good.
2. Good.
3. Good.
4. Good.
5. Good.
6. Not addressed. I fully agree with the original reviewer #1 that the authors should not overclaim. Even if they claim that 'had the potential' in the revised manuscript, it is highly speculating. The revision is not satisfactory.
7. Good.
8. Not addressed. The revision could be further improved to reflect the limitation of <12 hr half-life of the treatment. Simply reapplying the gel is potentially problematic and impractical in high-risk environments, for example, the old gel could have accumulated high viral titers.

RESPONSES TO REVIEWERS

(Note: All the responses here are in blue. All the changes have been highlighted in yellow in the main text of revised manuscript.)

Reviewer #2 (Remarks to the Author):

Major concerns:

1. Good.
2. Good.
3. It is highly beneficial for authors to seek professional editing services to improve their English. However, readability is subjective, and I will defer to you (the editor) on this matter.

Response: For better readability of our manuscript, we had asked American Journal Experts (AJE) for language improvement (code: F998-D162-8E95-5985-BB56) during the first revision (Fig. R1). Corresponding certificate was listed below.

Fig. R1 Language improvement certificate of English editing in AJE.

Other concerns:

I am uncertain whether the 'other concerns' mentioned by the original reviewer #1 pertain solely to editing the manuscript. Please note that my following comments are based solely on the editing request and not the experimental requests. If experimental requests, there should be additional work need to be done. I will defer to you (the editor) on this matter.

1. Good.
2. Good.
3. Good.

4. Good.

5. Good.

6. Not addressed. I fully agree with the original reviewer #1 that the authors should not overclaim. Even if they claim that 'had the potential' in the revised manuscript, it is highly speculating. The revision is not satisfactory.

Response: According to the reviewer's suggestion, we have now revised the description to avoid overclaim as follow.

Main text:

Page 9, line 261-263: In another experiment in which mice were challenged with SPV variant (B.1.1.529, Omicron) aerosols, corresponding GFP expression analysis in the nasal cavity and lung also showed the similar protective effect of aMV@GEL (Fig. 4I, Fig. S24), ~~indicating that the protective effect of aMV@GEL was independent of the variants.~~

Page 10, line 287-289: Furthermore, sMV also achieved a similar protective effect against the H1N1-PR8 virus (Fig. S27), indicating the applicability of sMV@GEL for another typical H1N1 serotype.

Page 11, line 309-311: Similar protective effects of sMV@GEL were also observed in mice challenged with H1N1-PR8, demonstrating that the protective effect of sMV@GEL could be extended to another typical H1N1 serotype (Fig. S30).

7. Good.

8. Not addressed. The revision could be further improved to reflect the limitation of <12 hr half-life of the treatment. Simply reapplying the gel is potentially problematic and impractical in high-risk environments, for example, the old gel could have accumulated high viral titers.

Response: According to the reviewer's suggestion, we have now added the discussion about the intranasal retention time of our intranasal mask and provided the upgraded suggestion as follow.

Main text:

Page 17, line 477-483: For example, peptide-based hydrogels, protein-based hydrogels, and hydrogels from blends can be sprayed into the nasal cavity under shear stress and immediately recover their mechanical properties^{66,67}, thus remaining fixed at the inner wall of the nasal cavity. In addition, considering the 8 h intranasal retention time of our hydrogel, effort can be made to upgrade the gel component, such as gellan gum with higher mucoadhesion^{68,69}, for individuals with long-term exposure in the high-risk environment.

Reference in main text:

68. Agrawal, M., et al. Stimuli-responsive in situ gelling system for nose-to-brain drug delivery. *J Control Release* **327**, 235-265 (2020).

69. Bakr, M.M., Shukr, M.H. & ElMeshad, A.N. In situ hexosomal gel as a promising tool to ameliorate the transnasal brain delivery of vinpocetine: central composite optimization and in vivo biodistribution. *J Pharm Sci-U.S* **109**, 2213-2223 (2020).